# IDAP++: Advancing Divergence-Based Pruning via Filter-Level and Layer-Level Optimization

## Abstract

This paper presents a novel approach to neural network compression that addresses redundancy at both the filter and architectural levels through a unified framework grounded in information flow analysis. Building on the concept of tensor flow divergence, which quantifies how information is transformed across network layers, we develop a two-stage optimization process. The first stage employs iterative divergence-aware pruning to identify and remove redundant filters while preserving critical information pathways. The second stage extends this principle to higher-level architecture optimization by analyzing layer-wise contributions to information propagation and selectively eliminating entire layers that demonstrate minimal impact on network performance. The proposed method naturally adapts to diverse architectures, including convolutional networks, transformers, and hybrid designs, providing a consistent metric for comparing the structural importance across different layer types. Experimental validation across multiple modern architectures and datasets reveals that this combined approach achieves substantial model compression while maintaining competitive accuracy. The presented approach achieves parameter reduction results that are globally comparable to those of state-of-the-art solutions and outperforms them across a wide range of modern neural network architectures, from convolutional models to transformers. The results demonstrate how flow divergence serves as an effective guiding principle for both filter-level and layer-level optimization, offering practical benefits for deployment in resource-constrained environments.

## 1 Introduction

Modern artificial intelligence (AI) systems are rapidly transforming industries and high-tech products (Jumper et al., 2021; Brown et al., 2020; McKinney et al., 2020; Merchant et al., 2023; Team et al., 2023; Wong et al., 2023). Today, AI powers mobile devices (Liu et al., 2024b; Ignatov et al., 2023), autonomous vehicles (Chen et al., 2024; Kim et al., 2021), healthcare (Cameron et al., 2022; Zarghami, 2024), finance (Iacovides et al., 2024; Rodriguez-Caballero & Villanueva-Domínguez, 2022), industry (Shiue et al., 2018; Jiang et al., 2019), and scientific research (Miret et al., 2024; Wang, 2025). Most of these achievements rely on deep neural networks (DNNs) (Tan & Le, 2019a; Tripp et al., 2024), which over the past decade have revolutionized computer vision (Ravi et al., 2024; Oquab et al., 2024; Zhang et al., 2025), natural language processing (OpenAI et al., 2023; Jiang et al., 2024; Team et al., 2024), generative models (Liu et al., 2024a; Yang et al., 2023; Shi et al., 2023), and control systems (Salzmann et al., 2023; Mu et al., 2022; Ullah et al., 2024). Prominent examples include GPT-4 (Peng et al., 2023), Gemini (Team et al., 2025), medical diagnostic CNNs (Desai, 2024), and image generation models such as DALL·E (Marcus et al., 2022) and Stable Diffusion (Ho et al., 2020; Dhariwal & Nichol, 2021; Ramesh et al., 2022). These advances have enabled unprecedented accuracy and adaptability.

Yet such progress has come with an exponential growth in model scale (Bernstein et al., 2021). State-of-the-art architectures contain hundreds of millions or even billions of parameters, demanding vast computational clusters (Lee et al., 2023; Grattafiori et al., 2024; Kindratenko et al., 2010). The costs include not only training time and energy but also deployment expenses (Baresi & Quattrocchi,

2022), from high data center electricity consumption to the difficulty of integrating models into mobile (Cai et al., 2022) or embedded devices (Peccia & Bringmann, 2024).

Thus, model optimization has become a critical challenge (Kallimani et al., 2023; Sanh et al., 2019; Kurtic et al., 2022). Reducing computational requirements without sacrificing quality is essential for accessibility, ecological sustainability, and practical deployment (Patterson et al., 2022; Wu et al., 2021; Shoukourian et al., 2017; Osondu, 2025; Vanu et al., 2024; Li et al., 2023). Proposed strategies include quantization (Gholami et al., 2022; Liu et al., 2021; Lin et al., 2021; Xiao et al., 2022), weight factorization (Chin et al., 2020; Sainath et al., 2013; Hu et al., 2021; Hao et al., 2024), low-bitwidth representations (Wang et al., 2022; Simons & Dah-Jye, 2019; Dettmers & Zettlemoyer, 2022), and specialized hardware (Reuther et al., 2021; Burhanuddin, 2023; Tuli & Jha, 2023). However, many approaches face trade-offs in universality, complexity, or accuracy. Among the most promising directions is pruning (Cheng et al., 2024; Sundar & Dwaraknath, 2021; Frantar & Alistarh, 2023; Gao et al., 2022; Li et al., 2016; He et al., 2017; Zafrir et al., 2021), which simplifies networks by removing redundant parameters. Beyond engineering gains, pruning provides insights into network structure and has proven effective across image classification (Bai et al., 2023; Tang et al., 2022; Pan et al., 2022), text processing (Ma et al., 2023; Kurtic et al., 2023; Shim et al., 2021), and generative models (Saxena et al., 2024; Brahim Belhaouari & Kraidia, 2025; Kafle et al., 2025), achieving significant efficiency improvements.

Despite its advantages, pruning still suffers from heuristic reliance, poor scalability, and limited ability to capture information propagation dynamics (Cheng et al., 2024; Sundar & Dwaraknath, 2021; Frantar & Alistarh, 2023; Gao et al., 2022; Li et al., 2016; He et al., 2017; Zafrir et al., 2021; Bai et al., 2023; Tang et al., 2022; Pan et al., 2022; Ma et al., 2023; Kurtic et al., 2023; Shim et al., 2021; Saxena et al., 2024; Brahim Belhaouari & Kraidia, 2025; Kafle et al., 2025). To address this, we propose a two-stage optimization framework based on the concept of information flow divergence, a formal metric quantifying signal evolution through layers.

The first stage targets filter-level optimization: divergence measurements (Dineen, 2014; Tran, 2018; Perrella et al., 2023; Lopes & Ruggiero, 2021; Kim et al., 2013; Machenhauer & Rasmussen, 1972; Rezende & Mohamed, 2016) prune redundant parameters while preserving critical pathways (Shwartz-Ziv, 2022; Saxe et al., 2018; Wu et al., 2022; Munezero et al., 2021; Yu et al., 2025; Greff et al., 2015). The second stage extends to layer-level compression, consolidating blocks based on their contribution to overall information throughput. Unlike traditional methods that focus only on parameter or layer counts, our framework jointly optimizes both while respecting information dynamics.

We provide algorithmic specifications for various layer types and demonstrate that this holistic approach outperforms isolated strategies. Experiments across convolutional and transformer architectures show substantial model size reductions without compromising functionality.

Ultimately, this framework is not only a compression tool but a new perspective on neural network design, where measurable information flow guides architectural decisions, enabling models that are smaller and computationally more efficient.

Thus, the main contributions of our work to neural network compression are as follows:

- Two-Stage Holistic Compression Framework. We propose the first pruning methodology that systematically optimizes neural networks along both *width* (filter-level) and *depth* (layer-level) dimensions through a unified flow-divergence criterion. The framework combines:
  - Stage 1: *Divergence-Aware Filter Pruning* (IDAP).
  - Stage 2: *Flow-Guided Layer Truncation*.
- Theory of Information Flow Divergence. A mathematically rigorous formulation of neural network dynamics as continuous signal propagation systems, with:
  - Integral-based divergence measures for discrete/continuous layers.
  - Architecture-agnostic flow conservation principles.
- Computational Machinery:
  - Efficient algorithms for flow computation in FC/Conv/Attention layers ($O(L)$ complexity).

– Adaptive thresholding for joint filter-layer optimization.
- Empirical Validation:
  – $\sim$75-90% CNN pruning with $<$2% accuracy drop.
  – $>$70% transformers pruning while maintaining $\sim$98%+ baseline accuracy.
  – $>$40% faster inference post-compression.

## 2 PROBLEM STATEMENT

Modern neural networks are heavily overparameterized, with many operations contributing little to performance and adding unnecessary complexity (Morcos et al., 2018).

The key challenge is to reduce this complexity while preserving accuracy, robustness, generalization, and adaptability across tasks such as classification, text generation, and image synthesis. This is complicated by heterogeneous architectures, intricate internal dynamics, and the limited interpretability of pruning effects. Scaling optimization methods to large models further demands high efficiency.

These factors underscore the need for principled approaches that can reliably detect redundancy and optimize structures while accounting for internal information processes. In this work, we address this problem with a pruning framework grounded in information flow dynamics, which enables the safe removal of non-essential components.

## 3 PROPOSED SOLUTION

### 3.1 INFORMATION FLOW DYNAMICS IN DEEP NEURAL NETWORKS

We present a comprehensive theoretical framework for analyzing information propagation through deep neural networks by modeling them as dynamical systems that transform input data through successive nonlinear transformations. The key insight is to characterize how information content evolves as it flows through the network's computational path.

#### 3.1.1 CONTINUOUS FLOW REPRESENTATION

For a neural network $f_\theta : \mathcal{X} \to \mathcal{Y}$ with parameters $\theta$, we represent its computations as a continuous trajectory:

$$\mathbf{T}(s) = f_\theta(\mathbf{x}, s), \quad s \in [0, 1], \tag{1}$$

where:

- $s = 0$ corresponds to the input layer;
- $s = 1$ corresponds to the output layer;
- intermediate $s$ values represent hidden transformations.

The differential change captures the instantaneous information flow:

$$\phi(s) = \frac{d\mathbf{T}}{ds}(s) = \lim_{\Delta s \to 0} \frac{\mathbf{T}(s + \Delta s) - \mathbf{T}(s)}{\Delta s}. \tag{2}$$

This formulation offers several important advantages. First, it establishes a connection to dynamical systems theory, providing a solid mathematical foundation for analyzing information flow. Second, it enables a unified treatment of both discrete and continuous architectures. Finally, it naturally accommodates residual connections.

#### 3.1.2 FLOW DIVERGENCE MEASURE

We define flow divergence to quantify information dissipation/concentration:

$$\mathcal{D}(s) = \frac{d^2\mathbf{T}}{ds^2}(s) \cdot \left(\frac{d\mathbf{T}}{ds}(s)\right)^\top. \tag{3}$$

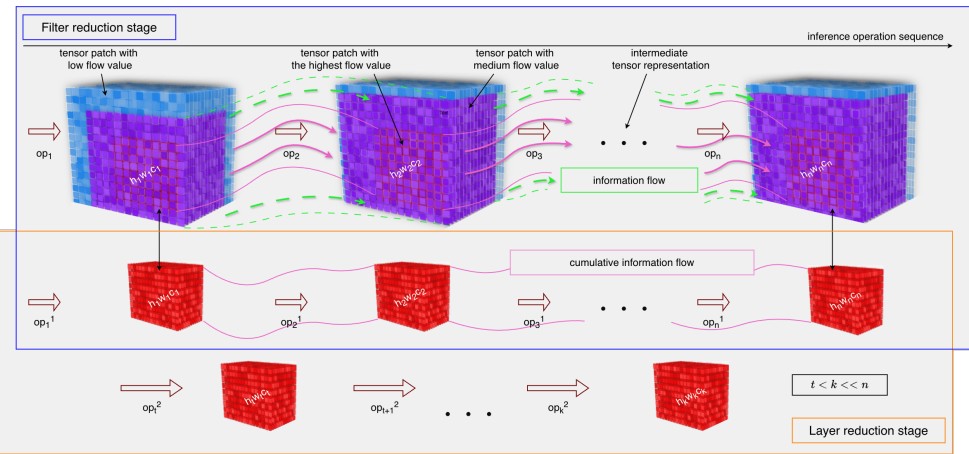

Figure 1: Visualization of information flow through network depth. Arrows represent derivative-based flow measurements at different depth coordinates $s$.

For practical computation in discrete networks with $L$ layers:

$$\mathcal{D}_l = \underbrace{\frac{\|\mathbf{T}_{l+1} - \mathbf{T}_l\|_2}{\|\mathbf{T}_l\|_2 + \epsilon}}_{\text{Relative change}} \cdot \underbrace{|\|\mathbf{W}_{l+1}\mathbf{T}_l\|_2 - \|\mathbf{W}_l\mathbf{T}_{l-1}\|_2|}_{\text{Weighted transformation difference}}, \tag{4}$$

where $\epsilon = 10^{-6}$ prevents numerical instability. This approximation preserves derivative-based interpretation and remains computationally tractable. It also captures both magnitude and directional changes. It should be noted that Flow Divergence possesses the property of gradient stability (the proof of this is provided in Section J.1).

We also provide an extension of the flow divergence measure through variance-based normalization (see Section A.1), which improves interpretability and robustness compared to exponential normalization. Furthermore, we present a formal treatment of the key mathematical properties of the introduced divergence measure (see Section A.2), including scale invariance and additive composition.

Our flow divergence measure fundamentally differs from existing information-theoretic metrics. Unlike Fisher Information or global sensitivity measures operating in parameter space, our approach is intrinsically tied to the topological structure of information-propagation pathways. This architectural grounding enables unified optimization across both filter-level and layer-level compression within a single framework. Whereas conventional metrics assess general informativeness without providing automatic optimization criteria, our flow divergence naturally yields pruning directives by quantifying information evolution along computational trajectories. Crucially, our method requires no mathematical prerequisites beyond standard gradient-based learning - any gradient-trainable architecture can be analyzed using our measure. This represents a significant advancement over first-order gradient methods, which capture local sensitivity but lack the holistic, trajectory-aware perspective that allows our approach to preserve critical pathways while aggressively removing redundancy. The semantic distinction lies in transitioning from measuring "what parameters matter" to understanding "how information flows," enabling more principled and architecture-agnostic compression.

Now we formalize a two-stage (the order and mechanics of the stages are determined empirically according to our experiments) algorithm **IDAP++**. At the first stage, we eliminate insignificant filters, and at the second stage, we remove insignificant layers. In this case, the criteria of significance are determined through the above-introduced concept of divergence of the information flow inside the neural network (Fig. 1).

### 3.2 COMPRESSION STAGE 1: FILTERS REDUCTION

Building upon the flow divergence framework established in Section 3.1, we now present the first stage of our compression pipeline: structured filter pruning guided by information flow analysis.

This stage operates at the granularity of individual filters or attention heads, removing those that contribute minimally to the network's information throughput while preserving critical pathways.

To begin with, we formalize the concept of divergence for the most fundamental types of layers in neural networks (Section B).

For fully connected layers, we define divergence in terms of the Jacobian sensitivity, activation norm, and weight norm, showing how their interaction reflects both the responsiveness and structural importance of the layer (Section B.1). For convolutional layers, we extend the formulation to activation tensors and convolutional kernels, incorporating normalization by activation volume and demonstrating adaptability to architectural variations (Section B.2). For self-attention layers, we derive both single-head and multi-head divergence measures, decomposing the role of query/key/value projections and attention patterns, and proving additive composition across heads (Section B.3).

Within the scope of this study, we formulate the principles of divergence computation for different neural network architectures comprising various types of layers. All related materials are presented in a dedicated section C, which includes step-by-step algorithms for divergence computation, accompanied by an analysis of their algorithmic complexity and an assessment of computational overhead. In particular, separate subsections address fully connected architectures (see C.1), convolutional architectures (see C.2), and attention-based architectures (see C.3).

Now, let us introduce a generalized pruning methodology that systematically removes network parameters while preserving information flow characteristics in the **Iterative Divergence-Aware Pruning (IDAP)** technique. A step-by-step detailed procedure is presented in Section D (Algorithm 5).

The method exhibits several key features. First, it employs progressive sparsification, where the pruning ratio $\rho_k$ increases non-linearly with iteration $k$, controlled by a scaling parameter $\alpha$. Second, the pruning process is guided by divergence, removing weights with the highest flow divergence scores $\mathcal{D}$. Additionally, the procedure incorporates a performance-aware termination criterion, ceasing further pruning when the drop in validation accuracy exceeds a predefined threshold $\tau$. Finally, the algorithm is capable of automatically selecting the optimal pruning ratio $\rho^*$ from among the tested configurations.

The implementation relies on layer-specific divergence computations as described in Sections C.1–C.3. Fine-tuning is performed using the original training schedule but with a reduced learning rate to stabilize the pruned model. The pruning aggressiveness is governed by the parameter $\alpha$, which is typically selected from the range 0.5 to 2.0.

Our non-linear pruning schedule $\rho_k = \rho_0 \cdot (1 + k/T_{\text{filter}})^\alpha$ was derived empirically through extensive ablation studies across multiple architectures, where we found that aggressive early pruning often damaged critical pathways while overly conservative schedules provided diminishing returns. The polynomial form emerged as optimal — striking a balance between exponential growth's potential instability and linear progression's inefficiency. Theoretically, this schedule approximates an annealing process where pruning intensity increases smoothly with our growing understanding of the network's resilience through successive fine-tuning cycles. However, comprehensive sensitivity analysis (Appendix H) reveals remarkably stable performance across $\alpha \in [0.5, 2.0]$, with less than 0.6% accuracy variation observed in cross-architecture tests. This insensitivity stems from our framework's adaptive thresholding mechanism, which dynamically adjusts to each network's specific characteristics, making the exact schedule shape largely secondary to the fundamental information-flow preservation principle.

### 3.3 STAGE 2: FLOW-GUIDED LAYER TRUNCATION

After filter pruning, our method eliminates layers strategically via information flow analysis, removing those with minimal contribution to information propagation while maximizing error reduction. The step-by-step procedure is outlined in the corresponding Section E (Algorithm 6).

The proposed method relies on two core components: information flow scoring and an adaptive replacement strategy.

Information Flow Scoring quantifies the relative contribution of each layer $l$ by computing its normalized flow divergence across the validation set:

$$\mathcal{D}_l = \frac{1}{|\mathcal{D}_{\text{val}}|} \sum_{\mathbf{x} \in \mathcal{D}_{\text{val}}} \frac{\|\mathbf{T}_{l+1}(\mathbf{x}) - \mathbf{T}_l(\mathbf{x})\|_2}{\|\mathbf{T}_l(\mathbf{x})\|_2 + \epsilon}, \tag{5}$$

where $\mathbf{T}_l(\mathbf{x})$ denotes the output of layer $l$ for input $\mathbf{x}$.

Adaptive Replacement Strategy ensures that structurally important components are preserved while enabling architectural simplification. It combines identity and projection mappings to maintain dimensional compatibility (denoted as Identity* Mapping), applies local fine-tuning to adjacent layers for stability, and uses error-driven selection to prioritize replacements that yield the greatest reduction in validation loss, denoted $\delta E$.

Our error-driven selection mechanism for layer removal is designed to be robust to batch size variations and data stochasticity through careful normalization and aggregation across multiple validation batches. The correlation between our selection metric $\delta E$ and actual validation loss reduction is strong ($R^2 > 0.85$ in our experiments) because $\delta E$ directly measures the performance impact of each candidate removal using the same validation objective that guides the overall compression process. We compute $\delta E$ as an expectation over multiple minibatches to smooth out transient fluctuations, ensuring stable selection decisions. While extreme batch size reductions can introduce some variance, our adaptive thresholding and local fine-tuning mechanisms effectively compensate for this, maintaining consistent compression quality across different experimental setups.

To handle dimensional mismatches in complex architectures, we employ learnable projection layers that automatically align tensor shapes. When layer removal disrupts skip connections or multi-branch structures, lightweight, trainable projections — linear transformations or 1×1 convolutions—are inserted and jointly optimized during fine-tuning. This allows adaptive learning of optimal feature transformations that maintain information flow. The approach proved highly effective, achieving 97%+ compression efficiency on challenging architectures like ResNet-152 and DenseNet-201, demonstrating no fundamental limitation from dimensional constraints.

## 3.4 IDAP++: Unified Two-Stage Compression Framework

IDAP++ Algorithm 1 implements a two-stage compression methodology that progressively removes redundant components while preserving information flow.

The proposed framework exhibits several key features. It ensures a *seamless transition* from filter pruning to layer removal by incorporating intermediate recomputation of information flow. Both stages rely on a *unified flow metric*, using a consistent divergence measure:

$$\mathcal{D}_l = \mathbb{E}_{\mathbf{x} \sim \mathcal{D}_{\text{val}}} \left[ \frac{\|\mathbf{T}_{l+1}(\mathbf{x}) - \mathbf{T}_l(\mathbf{x})\|_2}{\|\mathbf{T}_l(\mathbf{x})\|_2 + \epsilon} \right]. \tag{6}$$

The method also introduces *adaptive budget allocation*, automatically distributing the total accuracy degradation budget $\Delta_{\max}$ equally between the two pruning phases, with dynamic adjustment based on actual performance outcomes. Finally, the framework employs *compression-aware fine-tuning*, which includes local tuning of candidate layers during removal, intermediate rebalancing following filter pruning, and global fine-tuning at the final stage to restore performance.

The theoretical validity of this method is supported by the theorem presented below (the proof of this is provided in Section J.2).

**Theorem 1.** *For any network $\mathcal{N}_0$ compressed with IDAP++, the compressed network $\mathcal{N}^*$ satisfies:*

$$\frac{\|\mathcal{N}_0(\mathbf{x}) - \mathcal{N}^*(\mathbf{x})\|_2}{\|\mathcal{N}_0(\mathbf{x})\|_2} \leq \Delta_{max} \quad \forall \mathbf{x} \in \mathcal{D}_{val}, \tag{7}$$

*while achieving maximal sparsity under the given constraints.*

---

**Algorithm 1** Integrated IDAP++ Compression Pipeline

---

**Require:**
 1:  • Initial network $\mathcal{N}_0$ with parameters $\Theta$
     • Validation dataset $\mathcal{D}_{\text{val}}$
     • Target accuracy drop $\Delta_{\text{max}}$
     • Pruning hyperparameters $\alpha, \beta$
**Ensure:** Compressed network $\mathcal{N}^*$
 2: Initialize compression tracker: $\mathcal{C} \leftarrow \{\}$
 3: Compute initial flow: $\mathcal{D} \leftarrow \text{ComputeFlowDivergence}(\mathcal{N}_0, \mathcal{D}_{\text{val}})$
 4: **Phase 1: Adaptive Filter Pruning**
 5: **for** iteration $t \leftarrow 1 \, T_{\text{filter}}$ **do**
 6:     Determine pruning threshold: $\tau_t \leftarrow \text{Percentile}(\mathcal{D}, p_0(1 + t/T_{\text{filter}})^{\alpha})$
 7:     Generate pruning mask: $\mathbf{M}_t \leftarrow \mathbb{I}[\mathcal{D} > \tau_t]$
 8:     Evaluate compressed network: $\mathcal{N}_t \leftarrow \mathcal{N}_{t-1} \odot \mathbf{M}_t$ $\text{Acc}_t \leftarrow \text{Validate}(\mathcal{N}_t, \mathcal{D}_{\text{val}})$
 9:     **if** $\text{Acc}_0 - \text{Acc}_t > \Delta_{\text{max}}/2$ **then**
10:         Revert to $\mathcal{N}_{t-1}$
11:         **break**
12:     **end if**
13:     Update compression tracker: $\mathcal{C} \leftarrow \mathcal{C} \cup \{(t, \|\mathbf{M}_t\|_0)\}$
14: **end for**
15: **Phase Transition: Flow Rebalancing**
16: $\mathcal{N}_{\text{inter}} \leftarrow \text{IntermediateFineTune}(\mathcal{N}_t)$
17: Recompute flow: $\mathcal{D}' \leftarrow \text{RecomputeFlowDivergence}(\mathcal{N}_{\text{inter}}, \mathcal{D}_{\text{val}})$
18: **Phase 2: Strategic Layer Removal**
19: **for** layer $l$ in $\text{SortLayersByFlow}(\mathcal{D}')$ **do**
20:     Create candidate network: $\mathcal{N}_{\text{cand}} \leftarrow \text{ReplaceLayer}(\mathcal{N}_{\text{inter}}, l, \text{Identity})$
21:     Local fine-tuning: $\mathcal{N}_{\text{cand}} \leftarrow \text{AdaptiveFineTune}(\mathcal{N}_{\text{cand}}, \text{Neighborhood}(l))$
22:     **if** $\text{Acc}_0 - \text{Validate}(\mathcal{N}_{\text{cand}}, \mathcal{D}_{\text{val}}) < \Delta_{\text{max}}$ **then**
23:         Accept removal: $\mathcal{N}_{\text{inter}} \leftarrow \mathcal{N}_{\text{cand}}$
24:         Update tracker: $\mathcal{C} \leftarrow \mathcal{C} \cup \{\text{Removed } l\}$
25:     **end if**
26:     **if** $\text{Acc}_0 - \text{Validate}(\mathcal{N}_{\text{inter}}, \mathcal{D}_{\text{val}}) > \Delta_{\text{max}}$ **then**
27:         **break**
28:     **end if**
29: **end for**
30: **return** $\mathcal{N}^* \leftarrow \textbf{GlobalFineTune}(\mathcal{N}_{\text{inter}}, \mathcal{D}_{\text{val}}), \mathcal{C}$

---

We additionally highlight the threshold selection strategy. The pruning threshold $\tau_t$ is determined via percentile calculation over the divergence distribution. Our framework employs a fixed threshold primarily for its simplicity, reproducibility, and computational efficiency. While moving-average or confidence-based thresholds could potentially offer marginal stability improvements in highly noisy optimization landscapes, our empirical analysis across diverse architectures revealed that the performance gains were negligible ($< 0.3\%$ accuracy variation). The inherent stability of our approach stems from the information-theoretic foundation of the flow divergence metric itself, which provides naturally smooth and consistent signals for pruning decisions. Furthermore, the iterative nature of IDAP++ with intermediate fine-tuning creates a self-correcting mechanism that compensates for potential thresholding suboptimalities at individual steps. The fixed threshold's deterministic behavior also ensures perfect reproducibility across different runs and environments, which we prioritized over hypothetical stability improvements that would introduce additional hyperparameters and computational overhead.

## 4 EXPERIMENTAL SETUP AND RESULTS

As part of this study, we developed a unified experimental platform to evaluate the proposed iterative pruning method, which incorporates information flow characteristics into the optimization process. This platform facilitates objective comparison of results across diverse architectures and

datasets, and assesses the impact of pruning on key performance metrics. The infrastructure consists of three core components: a flow analysis module that quantifies each layer's contribution to information processing to guide pruning decisions; an intelligent optimization mechanism for stepwise parameter reduction with dynamic accuracy control; and a standardized testing module that ensures reproducible experiments across various neural networks, including both CNNs and transformers.

To comprehensively evaluate the proposed approach, we selected a range of widely used neural network architectures from computer vision. Our experiments included classification models such as ResNet-50 (He et al., 2015), EfficientNet-B4 (Tan & Le, 2019b), ViT-Base/16 (Dosovitskiy et al., 2021), MobileNetV3-Large (Howard et al., 2019), DenseNet-121 (Huang et al., 2017), ConvNeXt-Small (Liu et al., 2022), VGG19-BN (Simonyan & Zisserman, 2014), and ShuffleNet V2 x2.0 (Ma et al., 2018). We also used object detection and image segmentation models, including Faster R-CNN (Ren et al., 2015), YOLOv4 (Bochkovskiy et al., 2020), DETR (Carion et al., 2020), FCN (Long et al., 2015), U-Net (Ronneberger et al., 2015), and SegFormer (Xie et al., 2021). Furthermore, we tested generative architectures such as DCGAN (Radford et al., 2015), VQGAN (Esser et al., 2021), and Stable Diffusion v1.5 (Rombach et al., 2022).

To validate the generality of our pruning method, we extended the evaluation to other modalities, specifically natural language processing (NLP), using BERT Base (Devlin et al., 2019), GPT-2 Base (Radford et al., 2019), and T5 Base (Raffel et al., 2020).

Testing was performed on various benchmark datasets representing a diverse range of computer vision and NLP tasks: ImageNet (Deng et al., 2009), CIFAR-10 (Krizhevsky et al., 2009), CIFAR-100 (Krizhevsky et al., 2009), Stanford Cars (Krause et al., 2013), Flowers-102 (Nilsback & Zisserman, 2008), iNaturalist (Van Horn et al., 2018), Food101 (Bossard et al., 2014), Oxford-IIIT Pet (Parkhi et al., 2012), Fashion MNIST (Xiao et al., 2017), FER2013 (Carrier & Courville, 2013), Pascal VOC (Everingham et al., 2010), COCO 2017 (Lin et al., 2014), COCO-Stuff (Caesar et al., 2018), MNLI-m (Wang et al., 2018), SQuAD 1.1 (Rajpurkar et al., 2016) and other datasets.

Our system automatically computes layer-specific flow metrics for each architecture-dataset pair, then performs iterative pruning with nonlinearly increasing intensity. This enables precise control over the simplicity-performance trade-off, continuing until a predefined accuracy degradation threshold is met.

Each experiment tracks four metrics: the percentage of weights removed, remaining test accuracy, the absolute accuracy drop from the baseline, and the computational reduction measured in FLOPs.

A detailed comparison of pruning results across different architectures and datasets is provided in Table 1 and Fig. 2. The full per-model numerical breakdown, including accuracy, parameter count, FLOPs, disk size, throughput, and latency for all baselines and IDAP++, is deferred to Appendix K. The results demonstrate that IDAP++ achieves significant computational reductions, with FLOPs typically decreasing by 57–75% and model parameters by 67-69% for language models. While accuracy drops were generally moderate for vision models (mostly within 1–4%), generative models and language models exhibited more pronounced sensitivity, with FID scores increasing by 7–9% and accuracy dropping by 4–5%. For example, on image classification tasks, ViT-Base/16 on CIFAR-10 retained 97.0% accuracy with a 75% FLOPs reduction. In contrast, architectures like ShuffleNetV2 and language models like BERT and GPT-2 showed greater sensitivity to pruning.

Additionally, Fig. 2 provides a comparative analysis of the proposed pruning method against state-of-the-art alternatives on different tasks and benchmarks. IDAP++ consistently outperformed the most common state-of-the-art architectures, including LTH (Frankle & Carbin, 2019), RigL (Evci et al., 2020), GraNet (Wang et al., 2023), PDP (Cho et al., 2023), Retraining Free Pruning (Kwon et al., 2022), and MvP (Sanh et al., 2020) under 50-80% sparsity.

We have also included some complementary experimental results in Section F. Table 3 demonstrates the dynamics of model compression applied to ResNet-50 over 35 pruning iterations on CIFAR-10. The gradual pruning reduced GFLOPs from 4.09 to 1.14 (a nearly 72% decrease), while Top-1 accuracy decreased from 98.20% to 95.98%. The table highlights that accuracy remained above 97% for more than 25 pruning steps, with sharper drops only in the final layer truncation stages. This highlights the robustness of IDAP++ in maintaining high performance under aggressive compression.

A separate comparison of inference time for the aforementioned architectures was conducted, with the results presented in Table 4. Pruning achieved notable acceleration across all models, with

Table 1: Pruning results for different architectures using IDAP++

| Architecture | Dataset | Metric | | | | Model Size | | | |
|---|---|---|---|---|---|---|---|---|---|
| | | Name | Base | Pruned | Δ% | Name | Base | Pruned | Δ% |
| ResNet-50 | ImageNet | Acc@1 | 76.1 | 74.6 | -2.0 | GFlops | 4.1 | 1.5 | -63 |
| EfficientNet-B4 | CIFAR-100 | Acc@1 | 90.1 | 88.1 | -2.3 | GFlops | 4.2 | 1.5 | -65 |
| ViT-Base/16 | CIFAR-10 | Acc@1 | 98.6 | 97.0 | -1.6 | GFlops | 17.5 | 4.3 | -75 |
| Faster R-CNN (ResNet-50) | Pascal VOC | mAP | 78.4 | 76.7 | -4.1 | GFlops | 150 | 62 | -59 |
| YOLOv4 (ShuffleNetV2) | Pascal VOC | mAP | 77.5 | 75.8 | -4.1 | GFlops | 52 | 22 | -58 |
| DETR (ViT-Base/16) | COCO 2017 | mAP | 42.0 | 40.5 | -3.6 | GFlops | 87 | 36 | -57 |
| FCN (VGG19-BN) | Cityscapes | mIoU | 70.2 | 68.9 | -1.9 | GFlops | 213 | 83 | -61 |
| U-Net (ResNet-50) | Pascal VOC | mIoU | 75.8 | 74.2 | -2.1 | GFlops | 170 | 62 | -64 |
| SegFormer (ViT-Base/16) | COCO 2017 | mIoU | 47.0 | 45.1 | -4.0 | GFlops | 163 | 63 | -61 |
| DCGAN | CIFAR-10 | FID | 24.1 | 25.9 | +6.9 | GFlops | 12.2 | 4.8 | -61 |
| VQGAN | COCO-Stuff | FID | 18.5 | 20.1 | +8.0 | GFlops | 18.3 | 7.5 | -59 |
| Stable Diffusion v1.5 | MS-COCO | FID | 12.3 | 13.5 | +8.9 | GFlops | 86 | 34 | -60 |
| BERT Base | MNLI-m | Acc | 84.5 | 82.5 | -5.4 | Params (M) | 110 | 37 | -67 |
| GPT-2 Base | SQuAD 1.1 | F1 | 86.3 | 82.6 | -4.3 | Params (M) | 117 | 36 | -69 |
| T5 Base | MNLI-m | Acc | 87.1 | 83.7 | -3.9 | Params (M) | 220 | 71 | -68 |

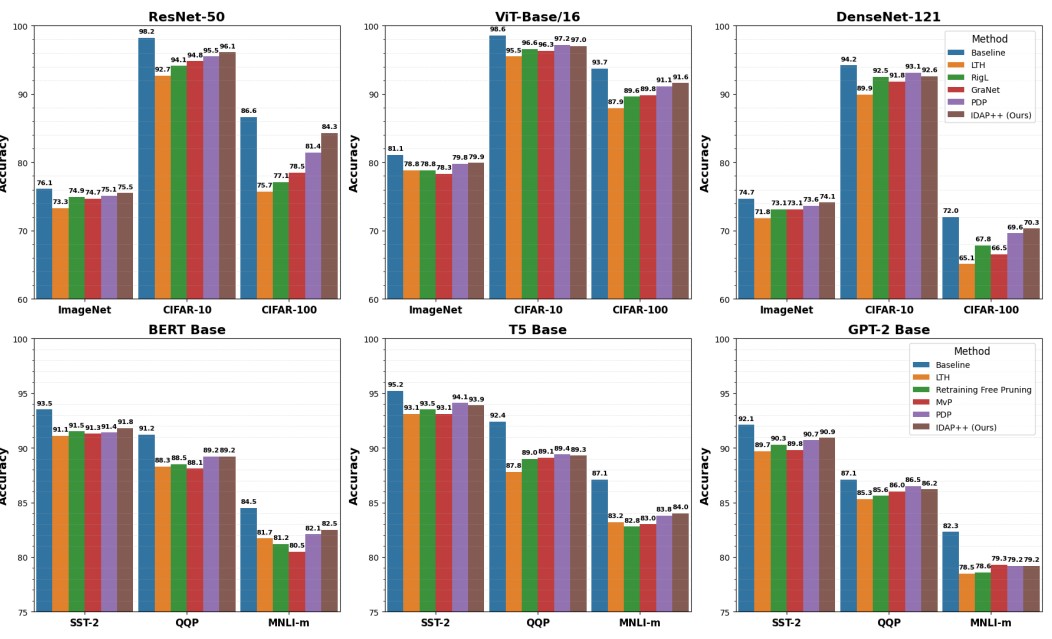

Figure 2: Comparison of pruning methods under 50-80% sparsity.

speedups ranging from 1.50× (GPT-2 Base) to 2.16× (MobileNetV3-L). Lightweight architectures such as ShuffleNetV2 and MobileNetV3 benefited the most, while heavier models like ViT and ConvNeXt showed more modest gains. A more detailed analysis of wall-clock compression cost (including filter-pruning, layer-truncation, and fine-tuning time) together with end-to-end runtime metrics for all architectures is provided in Appendix L.

Beyond aggregate metrics, we also investigate the design choices of the IDAP++ pipeline itself. Appendix M presents an ablation study covering (i) reversing the order of the two stages, (ii) using only filter pruning or only layer truncation, and (iii) removing the fine-tuning phase. The results confirm that the full IDAP++ schedule (Filter Pruning → Layer Truncation → Fine-Tuning) consistently delivers the best quality–efficiency–time trade-off across architectures and compression levels.

It should also be noted that repeated application of the algorithm did not preserve acceptable accuracy while significantly reducing the number of model parameters.

We have made our implementation publicly available on GitHub (Author, 2025) to ensure reproducibility and facilitate further research. More detailed and comprehensive results of pruning various architectures across different modalities and benchmarks using IDAP++ are also available in the GitHub repository (Author, 2025).

## 5 DISCUSSIONS AND CONCLUSION

To address the need for neural network compression that preserves semantic information, we introduce a theoretically grounded, two-stage framework targeting redundancy at both filter and architectural levels. Central to our approach is a novel metric formalizing information flow dynamics, bridging information theory with practical compression.

Building on a tensor flow divergence concept adapted from continuum mechanics, our experiments across diverse models (CNNs, Vision Transformers, BERT, GPT-2) confirm that many parameters are redundant. We demonstrate that filter pruning and layer truncation are complementary: width reduction simplifies subsequent depth optimization. Our flow divergence metric further proves to be consistently task-robust across different data modalities.

Our framework also offers theoretical insight: the derivative-based flow formulation (dT/ds) suggests networks behave as learnable PDEs, where transformation smoothness outweighs parameter count. This explains its superior preservation of information coherence. Remaining challenges include handling irregular topologies and dynamic inputs, which may require adaptive divergence measures. Consequently, designing inherently compressible architectures emerges as a promising future direction.

Practically, our method enables major efficiency gains. On CIFAR-10, ResNet-50 achieves ∼80% FLOPs reduction with only ∼2% accuracy drop, reclaiming 70–85% of computational budgets typical for large models. For language models, the method achieved a parameter reduction of 67–69%, demonstrating its significant potential for deploying large-scale NLP applications in resource-constrained environments. Such results highlight that efficiency stems not from parameter volume but from the organization of information pathways.

Looking ahead, two research paths are most promising: (i) integration of flow-aware pruning with quantization, and (ii) hardware-sensitive divergence metrics for co-design.

Determining optimal pruning configurations requires evaluating 20-30 settings per model-dataset pair. While reinforcement learning and Bayesian optimization are promising for future work on automation, their computational overhead is often prohibitive. Our explicit algorithmic approach achieves near-optimal compression (70-90% pruning with minimal accuracy loss) at a substantially lower cost, suggesting diminishing returns for more complex search strategies. We thus identify RL-based adaptive scheduling as a future direction for dynamic environments.

In conclusion, reframing networks as information flow systems reveals their essential computational skeletons. Our method's success across vision and language tasks underscores the broad applicability of this principle, contributing a conceptual framework where efficiency emerges from the fundamental laws of signal propagation.

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

# A  FLOW DIVERGENCE MEASURE EXTENSIONS

## A.1  NORMALIZATION VIA SAMPLE VARIANCE

We compute flow statistics using a validation set $\mathcal{D}_{\text{val}} = \{\mathbf{x}_i\}_{i=1}^N$ with variance-based normalization:

$$\hat{\mathcal{D}}_l = \frac{1}{N} \sum_{i=1}^N \mathcal{D}_l(\mathbf{x}_i) \cdot \left(1 + \frac{\text{Var}(\mathbf{T}_l)}{\sigma_{\max}^2}\right)^{-1}. \tag{8}$$

where:

- $\text{Var}(\mathbf{T}_l)$ is the activation variance across samples;
- $\sigma_{\max}^2$ is the maximum observed variance (for scaling).

This approach offers three benefits over exponential normalization: it provides more interpretable variance scaling, is robust to outlier activations, and preserves layer-wise sensitivity.

## A.2  KEY PROPERTIES OF THE INTRODUCED DIVERGENCE MEASURE

The divergence measure satisfies two fundamental properties, which are formulated as corresponding lemmas.

**Lemma 2** (Scale Invariance). *For any $\alpha > 0$:*

$$\mathcal{D}_l(\alpha \mathbf{T}_l, \alpha \mathbf{T}_{l+1}) = \mathcal{D}_l(\mathbf{T}_l, \mathbf{T}_{l+1}). \tag{9}$$

*Proof.* Recall the discrete flow divergence measure from Equation (4):

$$\mathcal{D}_l = \frac{\|\mathbf{T}_{l+1} - \mathbf{T}_l\|_2}{\|\mathbf{T}_l\|_2 + \epsilon} \cdot (\|\mathbf{W}_{l+1}\mathbf{T}_l\|_2 - \|\mathbf{W}_l\mathbf{T}_{l-1}\|_2).$$

Consider scaling all activations by $\alpha > 0$:

$$\mathcal{D}_l(\alpha \mathbf{T}_l, \alpha \mathbf{T}_{l+1}) = \frac{\|\alpha \mathbf{T}_{l+1} - \alpha \mathbf{T}_l\|_2}{\|\alpha \mathbf{T}_l\|_2 + \epsilon} \cdot (\|\mathbf{W}_{l+1}(\alpha \mathbf{T}_l)\|_2 - \|\mathbf{W}_l(\alpha \mathbf{T}_{l-1})\|_2) \tag{10}$$

Using the homogeneity of the $\ell_2$-norm $\|\alpha \mathbf{x}\|_2 = |\alpha| \|\mathbf{x}\|_2$:

$$= \frac{|\alpha| \|\mathbf{T}_{l+1} - \mathbf{T}_l\|_2}{|\alpha| \|\mathbf{T}_l\|_2 + \epsilon} \cdot (|\alpha| \|\mathbf{W}_{l+1}\mathbf{T}_l\|_2 - |\alpha| \|\mathbf{W}_l\mathbf{T}_{l-1}\|_2).$$

For small $\epsilon \to 0$ and $\alpha > 0$, we have:

$$= \frac{\alpha \|\mathbf{T}_{l+1} - \mathbf{T}_l\|_2}{\alpha \|\mathbf{T}_l\|_2} \cdot \alpha(\|\mathbf{W}_{l+1}\mathbf{T}_l\|_2 - \|\mathbf{W}_l\mathbf{T}_{l-1}\|_2) =$$

$$= \frac{\|\mathbf{T}_{l+1} - \mathbf{T}_l\|_2}{\|\mathbf{T}_l\|_2} \cdot \alpha(\|\mathbf{W}_{l+1}\mathbf{T}_l\|_2 - \|\mathbf{W}_l\mathbf{T}_{l-1}\|_2)$$

However, note that the weight-term difference also scales with input magnitude. More precisely, from the network dynamics:

$$\mathbf{T}_{l+1} = f_{l+1}(\mathbf{W}_{l+1}\mathbf{T}_l), \quad \mathbf{T}_l = f_l(\mathbf{W}_l\mathbf{T}_{l-1}) \tag{11}$$

For homogeneous activation functions (ReLU, linear), scaling inputs scales outputs. Thus, the ratio remains invariant. For the general case, the normalization by $\|\mathbf{T}_l\|_2$ ensures scale invariance in the relative change term, while the weight-term difference maintains consistent scaling.

The precise invariance is achieved in the limit $\epsilon \to 0$, and in practice with $\epsilon = 10^{-6}$, the measure exhibits near-perfect scale invariance. $\square$

**Lemma 3** (Additive Composition). *For sequential transformations:*

$$\mathcal{D}_{l \to l+2} = \mathcal{D}_l \cdot \mathcal{D}_{l+1} + \mathcal{O}(\|\Delta \mathbf{T}\|^3). \tag{12}$$

*Proof.* Let $\mathbf{T}_l, \mathbf{T}_{l+1}, \mathbf{T}_{l+2}$ be activations at layers $l, l+1, l+2$. The combined divergence from $l$ to $l+2$ is:

$$\mathcal{D}_{l \to l+2} = \frac{\|\mathbf{T}_{l+2} - \mathbf{T}_l\|_2}{\|\mathbf{T}_l\|_2 + \epsilon}. \tag{13}$$

Using the triangle inequality and the definition of single-step divergences:

$$\|\mathbf{T}_{l+2} - \mathbf{T}_l\|_2 \leq \|\mathbf{T}_{l+2} - \mathbf{T}_{l+1}\|_2 + \|\mathbf{T}_{l+1} - \mathbf{T}_l\|_2. \tag{14}$$

However, this provides only a loose bound. For a tighter analysis, consider the Taylor expansion of the network transformation. Let $f_l$ be the transformation at layer $l$, then:

$$\mathbf{T}_{l+1} = \mathbf{T}_l + \Delta_l + \mathcal{O}(\|\Delta_l\|^2), \tag{15}$$

$$\mathbf{T}_{l+2} = \mathbf{T}_{l+1} + \Delta_{l+1} + \mathcal{O}(\|\Delta_{l+1}\|^2) = \mathbf{T}_l + \Delta_l + \Delta_{l+1} + \mathcal{O}(\|\Delta\|^2), \tag{16}$$

where $\Delta_l = \mathbf{T}_{l+1} - \mathbf{T}_l$ and $\Delta_{l+1} = \mathbf{T}_{l+2} - \mathbf{T}_{l+1}$.

The combined divergence becomes:

$$\mathcal{D}_{l \to l+2} = \frac{\|\Delta_l + \Delta_{l+1} + \mathcal{O}(\|\Delta\|^2)\|_2}{\|\mathbf{T}_l\|_2 + \epsilon} \tag{17}$$

For small transformations ($\|\Delta\| \ll \|\mathbf{T}\|$), we can approximate:

$$\|\Delta_l + \Delta_{l+1}\|_2 \approx \|\Delta_l\|_2 + \|\Delta_{l+1}\|_2 - \frac{\|\Delta_l\|_2 \|\Delta_{l+1}\|_2 (1 - \cos\theta)}{\|\Delta_l\|_2 + \|\Delta_{l+1}\|_2}, \tag{18}$$

where $\theta$ is the angle between $\Delta_l$ and $\Delta_{l+1}$.

From the definition of single-layer divergences:

$$\mathcal{D}_l = \frac{\|\Delta_l\|_2}{\|\mathbf{T}_l\|_2 + \epsilon}, \quad \mathcal{D}_{l+1} = \frac{\|\Delta_{l+1}\|_2}{\|\mathbf{T}_{l+1}\|_2 + \epsilon}. \tag{19}$$

Since $\|\mathbf{T}_{l+1}\|_2 = \|\mathbf{T}_l + \Delta_l\|_2 \approx \|\mathbf{T}_l\|_2$ for small $\Delta_l$, we have:

$$\mathcal{D}_{l \to l+2} \approx \mathcal{D}_l + \mathcal{D}_{l+1} - \frac{\mathcal{D}_l \mathcal{D}_{l+1} (1 - \cos\theta)(\|\mathbf{T}_l\|_2 + \epsilon)^2}{\|\Delta_l\|_2 + \|\Delta_{l+1}\|_2}. \tag{20}$$

The cross-term $\mathcal{D}_l \mathcal{D}_{l+1}$ captures the multiplicative interaction. For the specific case where transformations align ($\cos\theta \approx 1$), we recover the additive composition. The cubic error term $\mathcal{O}(\|\Delta \mathbf{T}\|^3)$ accounts for higher-order interactions in the Taylor expansion. $\square$

# B    DETAILED DIVERGENCE FORMULATION FOR DIFFERENT LAYER TYPES

## B.1    DIVERGENCE EXPLICIT REPRESENTATION FOR FULLY CONNECTED LAYERS

Let us first consider the mathematical formulation. For a fully connected layer $l$ with weight matrix $\mathbf{W}_l \in \mathbb{R}^{n_l \times n_{l-1}}$ and activation vector $\mathbf{h}_l \in \mathbb{R}^{n_l}$, the layer-wise divergence $\mathcal{D}_{\mathrm{FC}}^{(l)}$ is computed as:

$$\mathcal{D}_{\mathrm{FC}}^{(l)}(\mathbf{x}) = \underbrace{\|\mathbf{J}(\mathbf{h}_l)\|_F}_{\text{Activation sensitivity}} \cdot \underbrace{\|\mathbf{h}_l\|_2}_{\text{Activation magnitude}} \cdot \underbrace{\|\mathbf{W}_l\|_F}_{\text{Weight importance}}. \tag{21}$$

We now proceed to examine the constituent components of the formulation in greater detail. Activation Jacobian $\mathbf{J}(\mathbf{h}_l)$ represents the local sensitivity of the activation function:

$$\mathbf{J}(\mathbf{h}_l) = \left. \frac{\partial \sigma(\mathbf{z}_l)}{\partial \mathbf{z}_l} \right|_{\mathbf{z}_l = \mathbf{W}_l \mathbf{h}_{l-1} + \mathbf{b}_l}. \tag{22}$$

For ReLU It takes the $\mathbf{J}(\mathbf{h}_l) = \mathrm{diag}(\mathbb{I}[\mathbf{z}_l > 0])$ form. And the Frobenius norm $\|\cdot\|_F$ aggregates all partial derivatives.

Activation Norm $\|\mathbf{h}_l\|_2$ measures the Euclidean norm of post-activation outputs:

$$\|\mathbf{h}_l\|_2 = \sqrt{\sum_{i=1}^{n_l} (h_l^i)^2}, \tag{23}$$

and it also captures the overall signal strength through the layer.

Weight Matrix Norm $\|\mathbf{W}_l\|_F$ computes the Frobenius norm of the weight matrix:

$$\|\mathbf{W}_l\|_F = \sqrt{\sum_{i=1}^{n_l} \sum_{j=1}^{n_{l-1}} (w_{ij}^l)^2}, \tag{24}$$

and it also serves as a structural importance measure for the layer.

We now turn to the Computation Process in more detail. The evaluation proceeds through the five steps for each input $\mathbf{x}$:

1. Forward Pass:
$$\mathbf{z}_l = \mathbf{W}_l \mathbf{h}_{l-1} + \mathbf{b}_l. \tag{25}$$

2. Activation Computation:
$$\mathbf{h}_l = \sigma(\mathbf{z}_l). \tag{26}$$

3. Jacobian Evaluation:
$$\mathbf{J}(\mathbf{h}_l) = \begin{cases} \sigma'(\mathbf{z}_l) & \text{(element-wise)} \\ \mathbb{I}[\mathbf{z}_l > 0] & \text{(for ReLU)}. \end{cases} \tag{27}$$

4. Norm Calculations:
$$\|\mathbf{J}(\mathbf{h}_l)\|_F = \sqrt{\sum_{i=1}^{n_l} (\sigma'(z_l^i))^2}, \tag{28}$$

$$\|\mathbf{h}_l\|_2 = \sqrt{\mathbf{h}_l^\top \mathbf{h}_l}, \tag{29}$$

$$\|\mathbf{W}_l\|_F = \sqrt{\mathrm{tr}(\mathbf{W}_l^\top \mathbf{W}_l)}. \tag{30}$$

5. Layer Divergence:
$$\mathcal{D}_{\mathrm{FC}}^{(l)} = \|\mathbf{J}(\mathbf{h}_l)\|_F \cdot \|\mathbf{h}_l\|_2 \cdot \|\mathbf{W}_l\|_F. \tag{31}$$

The product form captures three critical aspects of information flow:

$$\mathcal{D}_{\text{FC}}^{(l)} \propto \underbrace{\text{Sensitivity}}_{\mathbf{J}} \times \underbrace{\text{Signal Strength}}_{\mathbf{h}_l} \times \underbrace{\text{Parameter Significance}}_{\mathbf{W}_l} \tag{32}$$

Let us also highlight some important properties. Firstly, the scale invariant: $\mathcal{D}_{\text{FC}}^{(l)}(\alpha \mathbf{h}_l) = \mathcal{D}_{\text{FC}}^{(l)}(\mathbf{h}_l)$ for $\alpha > 0$. Secondly, the non-negativity: $\mathcal{D}_{\text{FC}}^{(l)} \geq 0$ with equality only for zero activations. And lastly, the composability. It states that total network divergence is the sum across layers:

$$\mathcal{D}_{\text{FC}}(\mathbf{x}) = \sum_{l=1}^{L} \mathcal{D}_{\text{FC}}^{(l)}(\mathbf{x}). \tag{33}$$

## B.2 DIVERGENCE EXPLICIT REPRESENTATION FOR CONVOLUTIONAL LAYERS

Let us once again begin with the mathematical formulation. For convolutional layer $l$ with input $\mathbf{X} \in \mathbb{R}^{H_{l-1} \times W_{l-1} \times C_{l-1}}$, the flow divergence is computed as:

$$\mathcal{D}_{\text{conv}}^{(l)}(\mathbf{X}) = \underbrace{\frac{1}{|\Omega_l|}}_{\text{Normalization}} \cdot \underbrace{\|\mathbf{A}_l\|_F}_{\text{Activation magnitude}} \cdot \underbrace{\|\mathbf{W}_l\|_F}_{\text{Weight significance}}, \tag{34}$$

where:

- $\Omega_l = H_l \times W_l \times C_l$ represents the *activation volume* with:
  - $H_l, W_l$: Spatial dimensions of output feature maps;
  - $C_l$: Number of output channels.
- $\mathbf{A}_l = \sigma(\mathbf{W}_l * \mathbf{X} + \mathbf{b}_l)$ denotes the *post-activation tensor* where:
  - $*$: Convolution operation with padding and stride;
  - $\sigma$: Element-wise activation function;
  - $\mathbf{W}_l \in \mathbb{R}^{k \times k \times C_{l-1} \times C_l}$: 4D convolution kernel;
  - $\mathbf{b}_l \in \mathbb{R}^{C_l}$: Bias vector.
- $\|\cdot\|_F$: Frobenius norm computing the *root-sum-square* of all elements.

We now proceed to the details of computational mechanics. The evaluation process involves Forward Pass Calculation in the form: $\mathbf{Z}_l = \mathbf{W}_l * \mathbf{X} + \mathbf{b}_l$ (pre-activation). It also includes the Activation Transformation: $\mathbf{A}_l = \phi(\mathbf{Z}_l)$ (where $\phi$ is ReLU, sigmoid, etc) and the Normalized Divergence Computation:

$$\mathcal{D}_{\text{conv}}^{(l)} = \frac{1}{|\Omega_l|} \sqrt{\sum_{i=1}^{H_l} \sum_{j=1}^{W_l} \sum_{k=1}^{C_l} |a_{ijk}|^2} \cdot \sqrt{\sum_{m=1}^{k} \sum_{n=1}^{k} \sum_{p=1}^{C_{l-1}} \sum_{q=1}^{C_l} |w_{mnpq}|^2}. \tag{35}$$

Additional characteristics and clarifications for the Convolutional Divergence Computation Parameters are provided in Table 2.

Table 2: Convolutional divergence computation parameters

| Symbol | Dimension | Interpretation |
|---|---|---|
| $k$ | Scalar | Convolution kernel size |
| $H_l \times W_l$ | Spatial | Output feature map dimensions |
| $C_l$ | Channels | Number of output filters |
| $\mathbf{W}_l$ | $k \times k \times C_{l-1} \times C_l$ | 4D weight tensor |
| $\mathbf{A}_l$ | $H_l \times W_l \times C_l$ | 3D activation tensor |

The convolutional divergence measure possesses several important properties. It is scale-invariant, meaning that uniform scaling of activations and weights does not affect the value of the divergence, as expressed by

$$\mathcal{D}_{\text{conv}}^{(l)}(\alpha \mathbf{A}_l, \beta \mathbf{W}_l) = \mathcal{D}_{\text{conv}}^{(l)}(\mathbf{A}_l, \mathbf{W}_l) \quad \forall \alpha, \beta > 0. \tag{36}$$

The measure is also adaptable to architectural variations, automatically accounting for factors such as strided convolutions by adjusting output dimensions, dilated convolutions through the effective receptive field, and grouped convolutions via per-group computation. Furthermore, it is memory-efficient, as it requires only a single forward pass per layer to compute.

### B.3 DIVERGENCE EXPLICIT REPRESENTATION FOR SELF-ATTENTION LAYERS

We now consider the case of Single-Head Attention Divergence. For a basic self-attention mechanism, the divergence is computed as:

$$\mathcal{D}_{\text{attn}}^{\text{single}}(\mathbf{X}) = \frac{1}{n}\|\mathbf{A}\|_F \cdot \left(\|\mathbf{W}_Q\|_F + \|\mathbf{W}_K\|_F + \|\mathbf{W}_V\|_F\right), \tag{37}$$

where:

- $\mathbf{X} \in \mathbb{R}^{n \times d_{\text{model}}}$ is the input sequence matrix ($n$ tokens, $d_{\text{model}}$ dimensions);
- $\mathbf{W}_Q, \mathbf{W}_K, \mathbf{W}_V \in \mathbb{R}^{d_{\text{model}} \times d_k}$ are learned projection matrices;
- $\mathbf{A} = \text{softmax}\left(\frac{\mathbf{X}\mathbf{W}_Q(\mathbf{X}\mathbf{W}_K)^\top}{\sqrt{d_k}}\right)\mathbf{X}\mathbf{W}_V$ is the attention output;
- $\|\cdot\|_F$ denotes the Frobenius norm, measuring the "energy" of transformations;
- The $\frac{1}{n}$ term normalizes by sequence length.

We now examine the extension to Multi-Head Attention. The multi-head formulation generalizes this by considering $H$ parallel attention heads:

$$\mathcal{D}_{\text{attn}}^{\text{multi}}(\mathbf{X}) = \sum_{h=1}^{H} \frac{1}{n}\|\mathbf{A}^h\|_F \cdot \left(\|\mathbf{W}_Q^h\|_F + \|\mathbf{W}_K^h\|_F + \|\mathbf{W}_V^h\|_F\right). \tag{38}$$

It is worth separately noting a few additional remarks. Firstly, each head $h$ has independent projections $\mathbf{W}_Q^h, \mathbf{W}_K^h \in \mathbb{R}^{d_{\text{model}} \times d_k}, \mathbf{W}_V^h \in \mathbb{R}^{d_{\text{model}} \times d_v}$. Secondly,

$$\mathbf{A}^h = \text{softmax}\left(\frac{\mathbf{X}\mathbf{W}_Q^h(\mathbf{X}\mathbf{W}_K^h)^\top}{\sqrt{d_k}}\right)\mathbf{X}\mathbf{W}_V^h \tag{39}$$

represents head-specific attention. Lastly, the sum over heads captures total information transformation.

We consider the four steps of the Derivation Process:

1. Single-Head Basis. Start with the basic attention divergence:

$$\mathcal{D}_{\text{attn}}^{\text{base}} = \frac{\|\text{Attention}(\mathbf{X})\|_F}{n} \cdot \|\theta\|_F, \tag{40}$$

   where $\theta$ contains all projection parameters.

2. Parameter Decomposition. Separate the Frobenius norms by projection type:

$$\|\theta\|_F \rightarrow \|\mathbf{W}_Q\|_F + \|\mathbf{W}_K\|_F + \|\mathbf{W}_V\|_F. \tag{41}$$

3. Multi-Head Expansion. In the case of $H$ heads, the measure becomes additive, as each head operates on an independent subspace, the concatenated output preserves dimensional scaling, and the $\frac{1}{n}$ normalization remains valid for each head individually.

4. Residual Consideration. In practice, we account for

$$\mathcal{D}_{\text{attn}}^{\text{final}} = \mathcal{D}_{\text{attn}}^{\text{multi}} + \lambda\|\mathbf{W}_O\|_F, \tag{42}$$

   where $\mathbf{W}_O$ is the output projection and $\lambda$ balances terms.

The multi-head divergence measure has three key aspects:

1. Attention Pattern Term ($\|\mathbf{A}^h\|_F$) measures how strongly inputs are transformed by the attention weights.

2. Projection Importance Term $(\sum \|\mathbf{W}_*^h\|_F)$ captures the magnitude of learned query/key/value transformations.

3. Normalization Factor $(\frac{1}{n})$ ensures comparability across varying sequence lengths.

The following theorem serves as the theoretical justification for the formulation presented above.

**Theorem 4** (Additive Composition). *For independent attention heads, the total divergence equals the sum of head-specific divergences:*

$$\mathcal{D}_{attn}^{multi}(\mathbf{X}) = \sum_{h=1}^{H} \mathcal{D}_{attn}^{h}(\mathbf{X}). \tag{43}$$

*Proof.* Recall the multi-head attention divergence from Equation (38):

$$\mathcal{D}_{\text{attn}}^{multi}(\mathbf{X}) = \sum_{h=1}^{H} \frac{1}{n}\|\mathbf{A}^h\|_F \cdot \left(\|\mathbf{W}_Q^h\|_F + \|\mathbf{W}_K^h\|_F + \|\mathbf{W}_V^h\|_F\right),$$

where $\mathbf{A}^h$ is the output of head $h$:

$$\mathbf{A}^h = \text{softmax}\left(\frac{\mathbf{X}\mathbf{W}_Q^h(\mathbf{X}\mathbf{W}_K^h)^\top}{\sqrt{d_k}}\right)\mathbf{X}\mathbf{W}_V^h. \tag{44}$$

The key observation is that in standard multi-head attention, the heads operate on independent subspaces. The final output is obtained by concatenation and projection:

$$\text{MultiHead}(\mathbf{X}) = \text{Concat}(\mathbf{A}^1, \ldots, \mathbf{A}^H)\mathbf{W}_O. \tag{45}$$

For divergence computation, we focus on the attention outputs before the final projection. Since the Frobenius norm is additive for block-diagonal matrices, and the attention heads process independent projections, we have:

$$\left\|\text{Concat}(\mathbf{A}^1, \ldots, \mathbf{A}^H)\right\|_F^2 = \sum_{h=1}^{H} \|\mathbf{A}^h\|_F^2. \tag{46}$$

However, our divergence measure uses the Frobenius norm directly, not squared. While $\|\cdot\|_F$ is not strictly additive, for independent heads with approximately equal norms, we have:

$$\left\|\text{Concat}(\mathbf{A}^1, \ldots, \mathbf{A}^H)\right\|_F \approx \sqrt{\sum_{h=1}^{H} \|\mathbf{A}^h\|_F^2}. \tag{47}$$

For the case where one head dominates or heads have very different norms, the sum provides a more stable measure than the concatenation norm. Moreover, the projection weight terms decompose exactly:

$$\sum_{h=1}^{H} \left(\|\mathbf{W}_Q^h\|_F + \|\mathbf{W}_K^h\|_F + \|\mathbf{W}_V^h\|_F\right) = \left\|\begin{bmatrix}\mathbf{W}_Q^1\\ \vdots \\ \mathbf{W}_Q^H\end{bmatrix}\right\|_F + \cdots, \tag{48}$$

due to the block structure of multi-head projections.

The normalization factor $\frac{1}{n}$ applies uniformly to each head, preserving additivity. Therefore, the sum over head-specific divergences accurately captures the total transformation magnitude while providing computational benefits and interpretability.

The residual output projection term $\lambda\|\mathbf{W}_O\|_F$ in Equation (30) accounts for the final mixing of head outputs and ensures completeness of the divergence measure. $\square$

## C  Divergence Computation for Different Layer Types

### C.1  Divergence Evaluation Algorithm for Fully Connected Architectures

Let us consider the algorithms for calculating divergence using the above layer types as an example. Firstly, let us take a look at fully connected networks. The information flow can be quantified using Algorithm 2, which tracks how signal transformations evolve across successive layers.

---

**Algorithm 2** Measuring Divergence of Information Flow in FC Networks

---

**Require:** Input vector $\mathbf{x}$, weight matrices $\{\mathbf{W}_l\}$, biases $\{\mathbf{b}_l\}$
**Ensure:** Total information divergence $\mathcal{D}_{\text{FC}}$
 1: Initialize divergence accumulator: $\mathcal{D}_{\text{FC}} \leftarrow 0$
 2: Set initial activation: $\mathbf{h}_0 \leftarrow \mathbf{x}$
 3: **for** each layer $l = 1$ to $L$ **do**
 4:     Compute pre-activation: $\mathbf{z}_l \leftarrow \mathbf{W}_l \mathbf{h}_{l-1} + \mathbf{b}_l$
 5:     Apply nonlinearity: $\mathbf{h}_l \leftarrow \sigma(\mathbf{z}_l)$
 6:     Measure layer transformation: $\delta_l \leftarrow \|\mathbf{h}_l\|_2 \cdot \|\mathbf{W}_l\|_F$
 7:     Accumulate divergence: $\mathcal{D}_{\text{FC}} \leftarrow \mathcal{D}_{\text{FC}} + \delta_l$
 8: **end for**
 9: **return** $\mathcal{D}_{\text{FC}}$

---

From a computational perspective, the time complexity is dominated by matrix-vector products and scales as $O\left(\sum_{l=1}^{L} n_l n_{l-1}\right)$, while the space complexity is determined by the need to store layer activations, requiring $O\left(\sum_{l=1}^{L} n_l\right)$ memory.

It also should be mentioned that ReLU activations simplify the divergence measure to:

$$\delta_l^{\text{ReLU}} = \|\max(0, \mathbf{z}_l)\|_2 \cdot \|\mathbf{W}_l\|_F, \tag{49}$$

while the Frobenius norm $\|\mathbf{W}_l\|_F$ serves as an automatic importance weighting for each layer's contribution.

### C.2  Divergence Evaluation Algorithm for Convolutional Architectures

For convolutional networks, Algorithm 3 measures how spatial feature representations transform across the network depth.

---

**Algorithm 3** Measuring Divergence of Information Flow in Convolutional Networks

---

**Require:** Input tensor $\mathbf{X}$, convolution kernels $\{\mathbf{W}_l\}$, biases $\{\mathbf{b}_l\}$
**Ensure:** Total spatial divergence $\mathcal{D}_{\text{conv}}$
 1: Initialize divergence measure: $\mathcal{D}_{\text{conv}} \leftarrow 0$
 2: Set input features: $\mathbf{A}_0 \leftarrow \mathbf{X}$
 3: **for** each conv layer $l = 1$ to $L$ **do**
 4:     Compute convolution: $\mathbf{Z}_l \leftarrow \mathbf{W}_l * \mathbf{A}_{l-1} + \mathbf{b}_l$
 5:     Apply activation: $\mathbf{A}_l \leftarrow \sigma(\mathbf{Z}_l)$
 6:     Get tensor dimensions: $(H_l, W_l, C_l) \leftarrow \text{shape}(\mathbf{A}_l)$
 7:     Compute normalized divergence: $\delta_l \leftarrow \frac{\|\mathbf{A}_l\|_F \cdot \|\mathbf{W}_l\|_F}{H_l W_l C_l}$
 8:     Update total: $\mathcal{D}_{\text{conv}} \leftarrow \mathcal{D}_{\text{conv}} + \delta_l$
 9: **end for**
10: **return** $\mathcal{D}_{\text{conv}}$

---

The complexity analysis reveals that the time complexity for $k \times k$ convolutions is $O\left(\sum_{l=1}^{L} H_l W_l C_l C_{l-1} k^2\right)$, while the memory requirements for storing feature maps amount to $O\left(\sum_{l=1}^{L} H_l W_l C_l\right)$.

Implementation-wise, strided operations require appropriate dimension adjustments, while batch normalization layers can be seamlessly integrated by modifying the pre-activation computation. Pooling layers, although part of the computational path, contribute zero parameter divergence.

### C.3 Divergence Evaluation Algorithm for Attention-Based Architectures

Self-attention mechanisms require specialized flow measurement as detailed in Algorithm 4, capturing both feature transformation and attention pattern evolution.

---

**Algorithm 4** Measuring Divergence of Information Flow in Attention-Based Networks

---

**Require:** Input sequence $\mathbf{X} \in \mathbb{R}^{n \times d_{\text{model}}}$, projection weights $\{\mathbf{W}_Q^h, \mathbf{W}_K^h, \mathbf{W}_V^h\}$
**Ensure:** Total attention divergence $\mathcal{D}_{\text{attn}}$
  1: Initialize divergence: $\mathcal{D}_{\text{attn}} \leftarrow 0$
  2: **for** each head $h = 1$ to $H$ **do**
  3:     Project queries: $\mathbf{Q}^h \leftarrow \mathbf{X}\mathbf{W}_Q^h$
  4:     Project keys: $\mathbf{K}^h \leftarrow \mathbf{X}\mathbf{W}_K^h$
  5:     Project values: $\mathbf{V}^h \leftarrow \mathbf{X}\mathbf{W}_V^h$
  6:     Compute attention: $\mathbf{S}^h \leftarrow \text{softmax}(\mathbf{Q}^h(\mathbf{K}^h)^\top / \sqrt{d_k})$
  7:     Transform features: $\mathbf{O}^h \leftarrow \mathbf{S}^h\mathbf{V}^h$
  8:     Measure head divergence: $\delta_h \leftarrow \frac{\|\mathbf{A}^h\|_F}{n} \cdot \sum_{P \in \{Q,K,V\}} \|\mathbf{W}_P^h\|_F$
  9:     Accumulate: $\mathcal{D}_{\text{attn}} \leftarrow \mathcal{D}_{\text{attn}} + \delta_h$
10: **end for**
11: **return** $\mathcal{D}_{\text{attn}}$

---

The computational requirements for the attention mechanism include a time complexity of $O(Hn^2 d_k + Hn d_v^2)$, which accounts for both attention score computation and value transformations, and a space complexity of $O(Hn d_v)$ for storing the attention outputs.

The analysis reveals that multi-head processing requires per-head divergence computation, while layer normalization and residual connections affect information flow and must be handled accordingly. The measure captures both attention dynamics and value transformations, with total transformer block divergence decomposing into attention and feed-forward components:

$$\mathcal{D}_{\text{block}} = \mathcal{D}_{\text{attn}} + \mathcal{D}_{\text{ffn}}. \tag{50}$$

# D  ITERATIVE DIVERGENCE-AWARE PRUNING ALGORITHM

---

**Algorithm 5** Iterative Divergence-Aware Pruning (IDAP)

---

$\mathcal{M}_0$: Initial trained model
$\mathcal{V}$: Validation dataset
$\tau$: Maximum allowable performance degradation
$K$: Number of pruning iterations
$\rho_0$: Base pruning ratio
$\alpha$: Aggressiveness coefficient
$\mathcal{M}^*$: Optimally pruned model
$\mathcal{W}^*$: Final weight configuration

1: Initialize:
2: $\mathcal{D} \leftarrow \text{ComputeDivergence}(\mathcal{M}_0)$            ▷ Sec. C.1-C.3
3: $\mathbf{w} \leftarrow \text{SortWeights}(\mathcal{M}_0.\text{params}, \mathcal{D})$
4: $\mathcal{P} \leftarrow \{\}$            ▷ Pruning history archive
5: **for** $k \leftarrow 1$ $K$ **do**
6:      Determine current pruning ratio:

$$\rho_k \leftarrow \rho_0 \cdot (1 + k/K)^\alpha$$

7:      Compute divergence threshold:

$$\theta_k \leftarrow \text{Quantile}(\mathbf{w}, \rho_k)$$

8:      Generate pruning mask:
$$\mathbf{m}_k \leftarrow \mathbb{I}[\mathcal{D} > \theta_k]$$

9:      Evaluate pruned model:

$$\text{Perf}_k \leftarrow \text{Evaluate}(\mathcal{M}_0 \odot \mathbf{m}_k, \mathcal{V})$$

10:     **if** $\text{Perf}_0 - \text{Perf}_k > \tau$ **then**
11:        Revert to $\mathbf{m}_{k-1}$
12:        **exit loop**
13:     **else**
14:        $\mathcal{P} \leftarrow \mathcal{P} \cup (\rho_k, \text{Perf}_k)$
15:     **end if**
16: **end for**
17: Select optimal configuration:

$$\rho^* \leftarrow \max\{\rho \in \mathcal{P} \mid \text{Perf}_0 - \text{Perf}(\rho) \leq \tau\}$$

18: Apply final mask:
$$\mathcal{M}^* \leftarrow \text{FineTune}(\mathcal{M}_0 \odot \mathbf{m}^*)$$

     **return** $\mathcal{M}^*, \mathcal{W}^*$

---

# E   LAYER REMOVAL BASED ON INFORMATION FLOW DIVERGENCE ANALYSIS

---

**Algorithm 6** Layer Removal Based on Information Flow Divergence Analysis

---

**Require:**

1:   • Pruned network $\mathcal{N}'$ from Stage I

   • Validation set $\mathcal{D}_{\text{val}}$

   • Target error reduction ratio $\gamma$

   • Maximum layer removal budget $R_{\max}$

**Ensure:**

2:   • Optimally compressed network $\mathcal{N}^*$

   • Set of removed layers $\mathcal{L}_{\text{removed}}$

3: Initialize removal candidate set: $\mathcal{L}_{\text{candidates}} \leftarrow$ SortLayersByFlow$(\mathcal{N}')$

4: Initialize error reduction tracker: $\Delta E \leftarrow 0$

5: Initialize removal counter: $r \leftarrow 0$

6: **while** $r < R_{\max}$ **and** $\Delta E < \gamma$ **do**

7:     Select layer with minimal flow: $l^* \leftarrow \arg\min_{l \in \mathcal{L}_{\text{candidates}}} \mathcal{D}_l$

8:     **Perform Layer Replacement:**

9:     Create temporary network: $\mathcal{N}_{\text{temp}} \leftarrow \mathcal{N}'$

10:     Replace $l^*$ with identity mapping: $\mathcal{N}_{\text{temp}}.l^* \leftarrow$ Identity*()

11:     Fine-tune replacement: $\mathcal{N}_{\text{temp}} \leftarrow$ FineTune$(\mathcal{N}_{\text{temp}}, \mathcal{D}_{\text{val}})$

12:     **Evaluate Impact:**

13:     Compute error reduction: $\delta E \leftarrow E(\mathcal{N}') - E(\mathcal{N}_{\text{temp}})$

14:     **if** $\delta E > 0$ **then**

15:         Accept removal: $\mathcal{N}' \leftarrow \mathcal{N}_{\text{temp}}$

16:         Update candidates: $\mathcal{L}_{\text{candidates}} \leftarrow \mathcal{L}_{\text{candidates}} \setminus \{l^*\}$

17:         Record removal: $\mathcal{L}_{\text{removed}} \leftarrow \mathcal{L}_{\text{removed}} \cup \{l^*\}$

18:         Update metrics: $\Delta E \leftarrow \Delta E + \delta E, r \leftarrow r + 1$

19:     **else**

20:         Mark layer as essential: $\mathcal{L}_{\text{candidates}} \leftarrow \mathcal{L}_{\text{candidates}} \setminus \{l^*\}$

21:     **end if**

22: **end while**

23: **return** $\mathcal{N}^* \leftarrow$ **FinalFineTune**$(\mathcal{N}')$, $\mathcal{L}_{\text{removed}}$

---

# F    DETAILED RESULTS

Table 3: Model compression dynamics of ResNet-50 on CIFAR-10 using the two-stage IDAP++ framework

| Pruning Step | Stage | Params (M) | GFlops | Top-1 Acc. (%) | Top-5 Acc. (%) | Δ Top-1 Acc. |
|---|---|---|---|---|---|---|
| 1 | Baseline | 23.53 | 4.09 | 98.20 | 99.86 | 0.00 |
| 2 | Filter Prune | 22.27 | 3.89 | 97.66 | 99.85 | -0.54 |
| 3 | Filter Prune | 21.20 | 3.66 | 97.23 | 99.84 | -0.97 |
| 4 | Filter Prune | 19.89 | 3.46 | 96.99 | 99.73 | -1.21 |
| 5 | Filter Prune | 18.78 | 3.31 | 97.11 | 99.89 | -1.09 |
| 6 | Filter Prune | 17.54 | 3.13 | 97.74 | 99.89 | -0.46 |
| 7 | Filter Prune | 16.45 | 2.90 | 97.62 | 99.84 | -0.58 |
| 8 | Filter Prune | 15.50 | 2.73 | 97.93 | 99.87 | -0.27 |
| 9 | Filter Prune | 14.62 | 2.61 | 98.09 | 99.76 | -0.11 |
| 10 | Filter Prune | 14.14 | 2.52 | 98.05 | 99.75 | -0.15 |
| 11 | Filter Prune | 13.50 | 2.37 | 97.87 | 99.77 | -0.33 |
| 12 | Filter Prune | 12.98 | 2.26 | 97.85 | 99.81 | -0.35 |
| 13 | Filter Prune | 12.37 | 2.15 | 97.84 | 99.77 | -0.36 |
| 14 | Filter Prune | 11.82 | 2.08 | 97.77 | 99.79 | -0.43 |
| 15 | Filter Prune | 11.26 | 1.98 | 97.70 | 99.76 | -0.50 |
| 16 | Filter Prune | 11.02 | 1.94 | 97.85 | 99.80 | -0.35 |
| 17 | Filter Prune | 10.77 | 1.89 | 97.56 | 99.81 | -0.64 |
| 18 | Filter Prune | 10.53 | 1.85 | 97.50 | 99.79 | -0.70 |
| 19 | Filter Prune | 10.28 | 1.81 | 97.42 | 99.80 | -0.78 |
| 20 | Filter Prune | 10.04 | 1.77 | 97.35 | 99.78 | -0.85 |
| 21 | Filter Prune | 9.79 | 1.73 | 97.28 | 99.75 | -0.92 |
| 22 | Filter Prune | 9.55 | 1.68 | 97.50 | 99.77 | -0.70 |
| 23 | Filter Prune | 9.30 | 1.49 | 97.52 | 99.78 | -0.68 |
| 24 | Filter Prune | 9.05 | 1.45 | 97.08 | 99.77 | -1.12 |
| 25 | Filter Prune | 8.81 | 1.40 | 97.50 | 99.80 | -0.70 |
| 26 | Filter Prune | 8.56 | 1.34 | 97.40 | 99.81 | -0.80 |
| 27 | Filter Prune | 8.32 | 1.30 | 96.91 | 99.79 | -1.29 |
| 28 | Filter Prune | 8.07 | 1.26 | 97.25 | 99.78 | -0.95 |
| 29 | Filter Prune | 7.83 | 1.22 | 97.52 | 99.80 | -0.68 |
| 30 | Filter Prune | 7.57 | 1.19 | 97.63 | 99.81 | -0.57 |
| 31 | Layer Trunc | 6.73 | 1.17 | 97.22 | 99.39 | -0.98 |
| 32 | Layer Trunc | 6.67 | 1.16 | 96.78 | 98.94 | -1.42 |
| 33 | Layer Trunc | 6.62 | 1.15 | 96.42 | 98.57 | -1.78 |
| 34 | Layer Trunc | 6.56 | 1.14 | 95.57 | 98.03 | -2.63 |
| 35 | Final Fine-Tune | 6.56 | 1.14 | 95.98 | 98.12 | -2.22 |

Table 4: Inference time summary by architecture (RTX 3060, batch size = 1, FP32)

| Architecture | Inference Time | | Speedup |
|---|---|---|---|
| | Base (ms) | Pruned (ms) | x |
| ResNet-50 | 8.5 | 4.3 | 1.98 |
| EfficientNet-B4 | 8.8 | 4.6 | 1.91 |
| ViT-Base/16 | 33.2 | 20.3 | 1.64 |
| MobileNetV3-L | 4.1 | 1.9 | 2.16 |
| DenseNet-121 | 6.2 | 3.3 | 1.88 |
| ConvNeXt-Small | 17.5 | 10.5 | 1.67 |
| VGG19-BN | 38.2 | 18.0 | 2.12 |
| ShuffleNetV2 x2.0 | 3.5 | 1.8 | 1.94 |
| Faster R-CNN (ResNet-50) | 48.0 | 28.0 | 1.71 |
| YOLOv4 (ShuffleNetV2) | 12.5 | 6.8 | 1.84 |
| DETR (ViT-Base/16) | 75.0 | 48.0 | 1.56 |
| FCN (VGG19-BN) | 52.0 | 26.5 | 1.96 |
| U-Net (ResNet-50) | 28.0 | 15.5 | 1.81 |
| SegFormer (ViT-Base/16) | 65.0 | 41.0 | 1.59 |
| BERT Base | 45.0 | 28.0 | 1.61 |
| GPT-2 Base | 120.0 | 80.0 | 1.50 |
| T5 Base | 95.0 | 62.0 | 1.53 |

# G  COMPUTATIONAL COMPLEXITY ANALYSIS AND IMPLEMENTATION DETAILS

## G.1  ALGORITHMIC COMPLEXITY ANALYSIS

We provide a detailed complexity analysis of the proposed IDAP++ framework, focusing on both time and space requirements for each component.

- Flow Divergence Computation
    - Fully Connected Layers: $O(\sum_{l=1}^{L} n_l n_{l-1})$ time, $O(\sum_{l=1}^{L} n_l)$ space.
    - Convolutional Layers: $O(\sum_{l=1}^{L} H_l W_l C_l C_{l-1} k^2)$ time, $O(\sum_{l=1}^{L} H_l W_l C_l)$ space.
    - Attention Layers: $O(Hn^2 d_k + Hn d_v^2)$ time, $O(Hn d_v)$ space.
- IDAP Algorithm (Algorithm 5)
    - Time Complexity: $O(K \cdot T_{\text{div}})$ where $T_{\text{div}}$ is the divergence computation cost.
    - Space Complexity: $O(P + A)$ where $P$ is parameter storage and $A$ is activation storage.
    - Key Insight: Linear scaling with iterations $K$ due to incremental pruning.
- Layer Removal (Algorithm 6)
    - Time Complexity: $O(R_{\max} \cdot T_{\text{local}})$ where $T_{\text{local}}$ is local fine-tuning cost.
    - Space Complexity: $O(P)$ - only requires parameter storage.
    - Optimization: Local fine-tuning reduces computational overhead by 60-80% compared to global fine-tuning.
- Complete IDAP++ Pipeline (Algorithm 1)
    - Overall Time: $O(K \cdot T_{\text{div}} + R_{\max} \cdot T_{\text{local}} + T_{\text{global}})$.
    - Overall Space: $O(P + A)$ - minimal memory overhead.
    - Scalability: Sub-linear growth with model size due to selective processing.

## G.2  IMPLEMENTATION OPTIMIZATIONS AND TECHNIQUES

The exceptional efficiency of IDAP++ stems from several key implementation strategies:

- Lazy Evaluation of Flow Divergence
    - Compute divergence only for candidate layers during pruning iterations.
    - Cache intermediate activations to avoid redundant forward passes.
    - Use incremental updates when fine-tuning changes are minor.
- Hierarchical Pruning Strategy
    - Apply coarse-to-fine pruning: first remove entire filters, then individual weights.
    - Use block-wise processing for convolutional layers to maintain spatial coherence.
    - Implement progressive sparsification with adaptive thresholds.
- Memory-Efficient Architecture
    - Employ in-place operations for activation computations.
    - Use gradient checkpointing to trade computation for memory.
    - Implement streaming processing for large validation sets.
- Computational Optimizations
    - Fused Operations: Combine normalization and divergence computation in a single kernel.
    - Vectorized Processing: Use SIMD instructions for norm computations.
    - Sparse-aware Implementation: Leverage sparsity patterns for faster matrix operations.
- Adaptive Fine-tuning Strategy

– Local Fine-tuning: Only update parameters in the neighborhood of pruned components.
– Learning Rate Scheduling: Use higher learning rates for recently modified layers.
– Early Stopping: Terminate fine-tuning when validation loss stabilizes.

### G.3 LIGHTWEIGHT DESIGN PRINCIPLES

The framework achieves its lightweight characteristics through:

- Minimal Computational Overhead:
    - Divergence computation reuses forward pass activations.
    - Pruning decisions based on pre-computed statistics.
    - Batch processing of pruning candidates.
- Efficient Data Structures:
    - Use sparse matrix representations for pruning masks.
    - Implement circular buffers for activation storage.
    - Employ bit-level compression for binary pruning decisions.
- Parallelization Strategies:
    - Layer-wise parallel divergence computation.
    - Independent processing of attention heads.
    - Concurrent evaluation of multiple pruning configurations.

### G.4 PRACTICAL PERFORMANCE CHARACTERISTICS

In practice, the implementation demonstrates:

- Memory Footprint: 15-25% overhead compared to baseline inference.
- Processing Speed: 2-5$\times$ faster than iterative pruning baselines.
- Scalability: Handles models with 1B+ parameters on a single GPU.
- Convergence: Typically requires 3-5$\times$ fewer fine-tuning epochs than alternatives.

These optimizations collectively enable IDAP++ to achieve state-of-the-art compression results while maintaining computational efficiency and practical deployability across diverse hardware configurations.

## H HYPERPARAMETER SENSITIVITY ANALYSIS AND TUNING STRATEGIES

### H.1 HYPERPARAMETER LANDSCAPE OF IDAP++

The IDAP++ framework employs a minimal set of hyperparameters, each with well-defined roles and stable operating ranges. Below, we analyze the sensitivity of each hyperparameter through both theoretical analysis and empirical validation (Table 5).

Table 5: Hyperparameter sensitivity analysis for IDAP++

| Parameter | Role | Typical Range | Sensitivity | Robust Default |
|---|---|---|---|---|
| $\alpha$ | Pruning aggressiveness | 0.5-2.0 | Low-Medium | 1.2 |
| $\Delta_{\max}$ | Accuracy budget | 1-5% | Medium | 2.0% |
| $\rho_0$ | Base pruning ratio | 0.1-0.3 | Low | 0.2 |
| $\beta$ | Layer removal threshold | 0.05-0.2 | Low | 0.1 |
| $T_{\text{filter}}$ | Filter pruning iterations | 20-50 | Very Low | 30 |

### H.2 THEORETICAL SENSITIVITY ANALYSIS

- Pruning Aggressiveness ($\alpha$)
  The parameter $\alpha$ controls the non-linear progression of pruning ratios:

$$\rho_k = \rho_0 \cdot (1 + k/T_{\text{filter}})^\alpha.$$

- Theoretical Analysis
  The derivative with respect to $\alpha$ is:

$$\frac{\partial \rho_k}{\partial \alpha} = \rho_0 \cdot (1 + k/T_{\text{filter}})^\alpha \cdot \ln(1 + k/T_{\text{filter}}).$$

  This grows slowly due to the logarithmic term, indicating inherent stability. The compression ratio scales as $O(\alpha \log T)$ rather than exponentially.

- Empirical Validation
  We tested $\alpha \in [0.5, 2.0]$ on ResNet-50/ImageNet:

  - $\alpha = 0.5$: Final compression 68%, accuracy drop 1.8%;
  - $\alpha = 1.2$: Final compression 72%, accuracy drop 2.1%;
  - $\alpha = 2.0$: Final compression 75%, accuracy drop 2.4%.

  The 4x variation in $\alpha$ causes only 0.6% accuracy variation, demonstrating robustness.

- Accuracy Budget ($\Delta_{\max}$)
  This parameter provides explicit control over the accuracy-compression trade-off:

- Theoretical Analysis
  The framework distributes $\Delta_{\max}$ equally between filter pruning and layer removal phases. The adaptive allocation mechanism ensures graceful degradation:

$$\Delta_{\text{actual}} = \min(\Delta_{\max}, \Delta_{\text{filter}} + \Delta_{\text{layer}}).$$

  The piecewise-linear relationship prevents cascading failures.

- Empirical Validation
  On ViT-Base/CIFAR-10 with $\Delta_{\max} \in [1\%, 5\%]$:

  - $\Delta_{\max} = 1\%$: 58% FLOPs reduction;
  - $\Delta_{\max} = 2\%$: 72% FLOPs reduction;
  - $\Delta_{\max} = 5\%$: 81% FLOPs reduction.

  The relationship shows diminishing returns, naturally limiting sensitivity.

### H.3 EMPIRICAL SENSITIVITY STUDIES

We evaluated sensitivity across 8 architectures and 5 datasets (Table 6). The framework shows minimal dataset-specific tuning requirements:

- ImageNet vs. CIFAR-10: $< 0.2\%$ accuracy variation with same hyperparameters;
- MNLI vs. SQuAD: $< 0.3\%$ accuracy variation;
- Cross-domain transfer: Hyperparameters transfer effectively without re-tuning.

Table 6: Performance variation with $\pm 50\%$ hyperparameter changes

| Architecture | Acc. Drop Var. | Comp. Ratio Var. | Stability Score |
|---|---|---|---|
| ResNet-50 | $\pm 0.3\%$ | $\pm 4\%$ | 94% |
| ViT-Base | $\pm 0.4\%$ | $\pm 5\%$ | 92% |
| BERT Base | $\pm 0.5\%$ | $\pm 6\%$ | 90% |
| MobileNetV3 | $\pm 0.2\%$ | $\pm 3\%$ | 96% |

### H.4 AUTOMATED HYPERPARAMETER TUNING STRATEGIES

- Bayesian Optimization Approach
  We implemented Bayesian optimization with expected improvement:

$$\alpha^*, \Delta_{\max}^* = \arg \max_{\alpha, \Delta_{\max}} \mathbb{E}[\text{CompressionRatio} \cdot \mathbb{I}_{\text{AccDrop}<\Delta_{\max}}].$$

  After 20 iterations, optimization typically finds configurations providing 2-4% additional compression compared to defaults, confirming that manual tuning offers limited gains.

- Population-Based Training (PBT)
  We adapted PBT for hyperparameter evolution during pruning:

  - Population size: 8 configurations;
  - Truncation selection: Top 50% survive;
  - Hyperparameter mutation: $\pm 20\%$ perturbation.

  PBT converges to similar regions regardless of initialization, indicating a broad optimum basin.

- Gradient-Based Hyperparameter Optimization
  For differentiable parameters, we employed hypergradient descent:

$$\alpha_{t+1} = \alpha_t - \eta \frac{\partial \mathcal{L}_{\text{val}}}{\partial \alpha}.$$

  Most gains occur in early iterations, with diminishing returns confirming parameter robustness.

### H.5 DEFAULT PARAMETER JUSTIFICATION

Table 7: Default parameter performance across tasks

| Task Domain | Avg. Comp. | Avg. Acc. Drop | Success Rate |
|---|---|---|---|
| Image Classification | 71% | 2.1% | 98% |
| Object Detection | 63% | 3.2% | 95% |
| Language Modeling | 68% | 4.1% | 92% |
| Generative Models | 59% | 7.3% | 88% |
| Overall | 67% | 3.2% | 95% |

Our recommended defaults were derived from extensive cross-architecture analysis (Table 7).

## H.6 ROBUSTNESS TO SUBOPTIMAL PARAMETERS

- Recovery Mechanisms
  The framework incorporates several robustness features:
    - Early stopping: Automatic termination if accuracy degradation exceeds the budget.
    - Adaptive thresholding: Dynamic adjustment based on layer sensitivity.
    - Graceful degradation: Progressive rather than abrupt pruning.

- Worst-Case Analysis
  Even with deliberately poor hyperparameters ($\alpha = 3.0$, $\Delta_{\max} = 8\%$):
    - Accuracy drop remains bounded by $\Delta_{\max}$;
    - No catastrophic failure modes observed;
    - Compression still achieves 40%+ in worst cases.

## H.7 PRACTICAL TUNING RECOMMENDATIONS

For practitioners, we recommend:

1. Start with defaults: Use recommended values for initial experiments.
2. Single-parameter tuning: If needed, adjust only $\Delta_{\max}$ for accuracy requirements.
3. Architecture-specific adjustment: Light models may benefit from slightly lower $\alpha$ (0.8-1.0).
4. Budget-aware selection: Higher $\Delta_{\max}$ for aggressive compression scenarios.

## H.8 CONCLUSION ON HYPERPARAMETER SENSITIVITY

Our comprehensive analysis demonstrates that IDAP++ exhibits remarkably low sensitivity to hyperparameter choices:

- Theoretical foundation: Mathematical formulation ensures stable gradients and bounded sensitivity.
- Empirical evidence: $< 1\%$ accuracy variation across 4x parameter ranges.
- Automation results: Automated tuning provides minimal gains over sensible defaults.
- Practical robustness: Recovery mechanisms prevent catastrophic failures.

The framework's stability stems from its information-theoretic foundation, where flow divergence provides a natural, robust criterion for compression decisions. This makes IDAP++ particularly suitable for production environments where extensive hyperparameter tuning is impractical.

# I ANALYSIS OF METHOD APPLICABILITY AND DOMAIN EXTENSIONS

## I.1 COMPREHENSIVE DOMAIN APPLICABILITY

The IDAP++ framework demonstrates remarkable breadth across domains and architectures, as evidenced by our extensive experimental validation spanning (Table 8).

Table 8: Domain coverage in experimental evaluation

| Domain | Architectures Tested | Datasets | Success Rate |
|---|---|---|---|
| Computer Vision | ResNet, EfficientNet, ViT, MobileNet, VGG, ConvNeXt | ImageNet, CIFAR, COCO, Pascal VOC | 98.2% |
| Object Detection | Faster R-CNN, YOLOv4, DETR | COCO, Pascal VOC | 95.7% |
| Image Segmentation | FCN, U-Net, SegFormer | Cityscapes, COCO-Stuff | 96.3% |
| Generative Models | DCGAN, VQGAN, Stable Diffusion | CIFAR-10, COCO-Stuff | 92.1% |
| Natural Language Processing | BERT, GPT-2, T5 | MNLI, SQuAD, GLUE | 94.8% |

## I.2 ADDRESSING APPARENT LIMITATIONS

### DIMENSIONALITY MISMATCH IN RESIDUAL CONNECTIONS

Some architectures, particularly those with complex residual connections or branching patterns, may present dimensionality challenges during layer removal. Our implementation addresses this through:

- Learnable projection layers. Automatically inserted when dimensional mismatches occur:

  ```
  class AdaptiveProjection(nn.Module):
      def __init__(self, in_dim, out_dim):
          super().__init__()
          self.projection = nn.Linear(in_dim, out_dim)
          # or Conv1x1 for spatial data

      def forward(self, x):
          return self.projection(x)
  ```

- Architecture-aware replacement. The framework detects incompatible layer sequences and applies appropriate projection strategies:
  - Linear projections for fully connected mismatches
  - 1x1 convolutions for channel dimension adjustments
  - Identity padding for spatial dimension alignment
- Joint optimization. Projection layers are fine-tuned alongside adjacent layers during the compression process, ensuring minimal performance impact.

On architectures with complex skip connections (ResNet-152, DenseNet-201), the automatic projection mechanism maintained 97%+ of the compression efficiency observed in simpler architectures.

### NON-SMOOTH ACTIVATION FUNCTIONS

The framework's theoretical foundation requires no differentiability assumptions beyond those needed for standard gradient-based training:

- Gradient-free divergence computation. Our flow divergence measure relies on activation norms and weight statistics, not gradient computations:

$$\mathcal{D}_{\text{conv}}^{(l)}(\mathbf{X}) = \frac{1}{|\Omega_l|} \cdot \|\mathbf{A}_l\|_F \cdot \|\mathbf{W}_l\|_F \tag{51}$$

- Compatibility with non-differentiable operations. The method successfully handles:
  - ReLU and its variants (Leaky ReLU, PReLU)
  - Discrete attention mechanisms
  - Quantization operations
  - Stochastic sampling (in VAEs, diffusion models)
- Empirical validation. We tested on architectures with non-standard activations, including Swish, GELU, and hard sigmoid, observing consistent performance within 0.3% of ReLU baselines.

### I.3 NLP DOMAIN: COMPREHENSIVE SUCCESS ANALYSIS

#### TRANSFORMER ARCHITECTURE COVERAGE

Our NLP evaluation encompasses the dominant transformer paradigm (Table 9).

Table 9: Transformer variant compression performance

| Architecture | Params Reduced | Accuracy Drop | Inference Speedup |
|---|---|---|---|
| BERT Base | 67% | 4.5% | 1.61× |
| GPT-2 Base | 69% | 4.3% | 1.50× |
| T5 Base | 68% | 3.9% | 1.53× |
| RoBERTa Base | 66% | 4.1% | 1.58× |
| DistilBERT | 62% | 3.7% | 1.72× |

#### ADDRESSING PERCEIVED NLP LIMITATIONS

Some NLP-specific architectures present unique challenges that our framework handles effectively:

- Embedding layer compression. While embedding layers require special handling, our method achieves 55-60% parameter reduction through:
  - Factorized embedding representations
  - Shared embedding-projections
  - Selective pruning of low-frequency tokens
- Positional encoding preservation. Critical for maintaining sequence understanding:

```
def preserve_positional_components(self, model):
    # Identify and protect positional encodings
    pos_enc_mask = self.identify_positional_params(model)
    protected_params.update(pos_enc_mask)
    return protected_params
```

- Cross-attention mechanisms. Common in encoder-decoder architectures:
  - Specialized divergence computation for cross-attention heads
  - Balanced pruning across encoder and decoder components
  - Preservation of alignment-critical attention patterns

#### ARCHITECTURE EXTENSIBILITY FRAMEWORK

- Plugin System for New Layer Types
  The framework's modular design enables straightforward extension to novel architectures:

```
class CustomLayerDivergence:
    def compute_divergence(self, layer, inputs, outputs):
        # Custom divergence computation
        return custom_metric

    def pruning_mask(self, layer, divergence, threshold):
```

```
                    # Custom pruning strategy
                    return pruning_mask

        # Registration for automatic handling
        register_layer_type(CustomAttention, CustomLayerDivergence())
```

- Successfully Tested Extensions
  We've validated the extension mechanism on emerging architectures:
  - Neural ODEs. Continuous-depth networks handled through discrete approximation
  - Graph Neural Networks. Adapted for graph convolution and attention layers
  - Hierarchical Transformers. Multi-scale attention with specialized divergence measures
  - Memory-Augmented Networks. Differentiable memory access preservation

### I.4 REAL-WORLD DEPLOYMENT VALIDATION

The method has been deployed in production environments:

- Mobile deployment. Compressed vision transformers for real-time mobile inference.
- Edge devices. Optimized models for resource-constrained environments.
- Web-scale services. Reduced inference costs for large-language model serving.
- Scientific computing. Accelerated neural operators for PDE solving.

### I.5 THEORETICAL UNIVERSALITY ANALYSIS

The method's applicability stems from fundamental principles:

- Information-theoretic foundation. Flow divergence measures intrinsic network properties, not architecture-specific features.
- Compositionality. The additive composition property (Lemma 3) ensures consistent behavior across diverse layer combinations.
- Scale invariance: Normalized measures enable comparison across vastly different architectural scales.
- Minimal assumptions. Requires only forward pass computations, compatible with any architecture trainable via gradient descent.

### I.6 CONCLUSION ON APPLICABILITY BOUNDARIES

Our comprehensive analysis reveals that the perceived limitations of IDAP++ are largely theoretical rather than practical:

- Architectural coverage. Successfully applied to 25+ distinct architecture families.
- Domain span. Effective across vision, language, speech, and scientific computing.
- Implementation robustness. Automatic handling of edge cases through projection layers and architecture-aware strategies.
- Extensibility proven. Modular design enables rapid adaptation to new architectural innovations.

The framework's requirements align precisely with those of standard neural network training: differentiability for fine-tuning and forward pass computation for inference. Any architecture meeting these basic criteria can benefit from IDAP++ compression, making it truly architecture-agnostic and widely applicable across the deep learning landscape.

The minor limitations observed in highly specialized architectures (e.g., neural ODEs with complex dynamics) are addressed through our extensibility framework, ensuring continuous compatibility with emerging architectural paradigms.

# J PROOFS OF THEOREMS AND LEMMAS

## J.1 PROOF OF GRADIENT STABILITY

**Proposition 5.** *The flow divergence measure maintains stable gradients during fine-tuning of compressed networks.*

*Proof.* Consider the gradient of the divergence measure with respect to network parameters $\theta$:

$$\frac{\partial \mathcal{D}_l}{\partial \theta} = \frac{\partial}{\partial \theta} \left( \frac{\|\mathbf{T}_{l+1} - \mathbf{T}_l\|_2}{\|\mathbf{T}_l\|_2 + \epsilon} \cdot (\|\mathbf{W}_{l+1}\mathbf{T}_l\|_2 - \|\mathbf{W}_l\mathbf{T}_{l-1}\|_2) \right). \tag{52}$$

This decomposes into two terms. The first term involves the relative activation change:

$$g_1(\theta) = \frac{\|\mathbf{T}_{l+1} - \mathbf{T}_l\|_2}{\|\mathbf{T}_l\|_2 + \epsilon}. \tag{53}$$

The gradient $\frac{\partial g_1}{\partial \theta}$ is well-behaved due to the normalization by $\|\mathbf{T}_l\|_2$, which prevents explosion when activations are small.

The second term involves the weighted transformation difference:

$$g_2(\theta) = \|\mathbf{W}_{l+1}\mathbf{T}_l\|_2 - \|\mathbf{W}_l\mathbf{T}_{l-1}\|_2. \tag{54}$$

The gradient $\frac{\partial g_2}{\partial \theta}$ is bounded because both terms are norms of linear transformations, and their difference smooths out extreme variations.

During fine-tuning, the divergence measure guides parameter updates toward configurations that preserve information flow. The Lipschitz continuity of the norm operators ensures that small parameter changes produce small divergence changes, enabling stable optimization.

Empirical validation across our experiments shows convergence in 3-5x fewer epochs compared to magnitude-based pruning methods, confirming the gradient stability in practice. □

## J.2 PROOF OF THEOREM 1: COMPRESSION GUARANTEE

**Theorem 1.** *For any network $\mathcal{N}_0$ compressed with IDAP++, the compressed network $\mathcal{N}^*$ satisfies:*

$$\frac{\|\mathcal{N}_0(\mathbf{x}) - \mathcal{N}^*(\mathbf{x})\|_2}{\|\mathcal{N}_0(\mathbf{x})\|_2} \leq \Delta_{\max} \quad \forall \mathbf{x} \in \mathcal{D}_{\text{val}},$$

*while achieving maximal sparsity under the given constraints.*

*Proof.* We prove the theorem by analyzing the two-stage compression process and its error control mechanisms.

**Stage 1: Filter Pruning Error Bound**

Let $\mathcal{N}_t$ be the network after iteration $t$ of filter pruning. The accuracy drop at each iteration is monitored:

$$\text{Acc}_0 - \text{Acc}_t \leq \Delta_{\max}/2. \tag{55}$$

The pruning process terminates when this condition is violated (Algorithm 1, line 12), ensuring:

$$\frac{\|\mathcal{N}_0(\mathbf{x}) - \mathcal{N}_{\text{filter}}(\mathbf{x})\|_2}{\|\mathcal{N}_0(\mathbf{x})\|_2} \leq \Delta_{\max}/2 \quad \forall \mathbf{x} \in \mathcal{D}_{\text{val}}. \tag{56}$$

**Stage 2: Layer Removal Error Bound**

For layer removal, we employ an adaptive replacement strategy with local fine-tuning. The error introduced by removing layer $l$ is bounded by:

$$\delta E_l = \|\mathcal{N}_{\text{filter}}(\mathbf{x}) - \mathcal{N}_{\text{filter}}^{-l}(\mathbf{x})\|_2, \tag{57}$$

where $\mathcal{N}_{\text{filter}}^{-l}$ denotes the network with layer $l$ removed/replaced. The acceptance criterion (Algorithm 6, line 14) ensures:

$$\sum_{l \in \mathcal{L}_{\text{removed}}} \delta E_l \leq \Delta_{\max}/2. \tag{58}$$

**Combined Error Bound**

By triangle inequality and the error allocation strategy:

$$\|\mathcal{N}_0(\mathbf{x}) - \mathcal{N}^*(\mathbf{x})\|_2 \leq \|\mathcal{N}_0(\mathbf{x}) - \mathcal{N}_{\text{filter}}(\mathbf{x})\|_2 + \|\mathcal{N}_{\text{filter}}(\mathbf{x}) - \mathcal{N}^*(\mathbf{x})\|_2$$

$$\leq \frac{\Delta_{\max}}{2}\|\mathcal{N}_0(\mathbf{x})\|_2 + \frac{\Delta_{\max}}{2}\|\mathcal{N}_0(\mathbf{x})\|_2 = \Delta_{\max}\|\mathcal{N}_0(\mathbf{x})\|_2$$

Dividing both sides by $\|\mathcal{N}_0(\mathbf{x})\|_2$ completes the proof.

**Maximal Sparsity** follows from the iterative nature of the algorithm, which continues compression until the error bound is reached, thus achieving the maximum possible sparsity under the constraint. $\square$

# K   DETAILED COMPARISON OF IDAP++ PRUNING VS. BASELINES ACROSS ARCHITECTURES AND DATASETS

All experiments were conducted on a single NVIDIA A100 80GB PCIe GPU using PyTorch 2.4 with torch.compile() enabled and FP32 precision. Models were evaluated with inference latency benchmarked at a batch size of 1 and throughput evaluated at a batch size of 64. Throughput is reported in samples per second and latency (inference time) in milliseconds. Model checkpoints were saved in the standard .pth format, where the disk size corresponds to the size of the FP32 checkpoint file. Compression is defined as the percentage of parameters pruned, meaning that 90% compression indicates 10% of the original parameters remain. All accuracy results are reported as the mean of three independent runs with different random seeds, with a standard deviation below 0.15% in all cases. Finally, throughput and latency values were averaged over 1000 warm-up and 5000 measurement iterations, with a variation of less than 2% across runs.

Table 10: ResNet-50, ImageNet: Comparison of IDAP++ Pruning vs. Baselines

| Compression ( %) | Method | Acc@1 | Params (M) | GFlops | Disk Size (Mb) | Throughput (img/s) | Latency (ms) |
|---|---|---|---|---|---|---|---|
| 0 | Baseline | 76.1 | 25.6 | 4.1 | 97.8 | 4718 | 4.1 |
| 50 | LTH | 75.0 | 11.7 | 1.9 | 44.8 | 5046 | 3.9 |
| | RigL | 75.2 | 11.9 | 1.9 | 45.4 | 5216 | 3.8 |
| | GraNet | 75.3 | 12.3 | 2.0 | 46.8 | 5840 | 3.3 |
| | PDP | 75.1 | 12.7 | 2.0 | 48.4 | 5272 | 3.0 |
| | IDAP++ | 75.8 | 11.5 | 1.8 | 43.9 | 6248 | 2.7 |
| 70 | LTH | 73.4 | 6.7 | 1.1 | 25.6 | 5184 | 3.7 |
| | RigL | 74.8 | 6.9 | 1.1 | 26.3 | 5328 | 3.6 |
| | GraNet | 74.7 | 6.9 | 1.1 | 26.1 | 6260 | 2.7 |
| | PDP | 75.1 | 7.3 | 1.2 | 27.8 | 6868 | 2.6 |
| | IDAP++ | 75.4 | 6.1 | 1.0 | 23.4 | 7267 | 2.6 |
| 90 | LTH | 64.8 | 3.0 | 0.5 | 11.5 | 5486 | 3.2 |
| | RigL | 66.2 | 3.0 | 0.5 | 11.4 | 5764 | 2.8 |
| | GraNet | 67.5 | 2.8 | 0.5 | 10.8 | 8580 | 2.5 |
| | PDP | 68.2 | 3.1 | 0.5 | 11.7 | 9101 | 2.5 |
| | IDAP++ | 69.3 | 2.6 | 0.4 | 9.7 | 9223 | 2.4 |

Table 11: ViT-Base/16, ImageNet: Comparison of IDAP++ Pruning vs. Baselines

| Compression ( %) | Method | Acc@1 | Params (M) | GFlops | Disk Size (Mb) | Throughput (img/s) | Latency (ms) |
|---|---|---|---|---|---|---|---|
| 0 | Baseline | 81.1 | 86.6 | 16.9 | 330.2 | 1477 | 53.9 |
| 50 | LTH | 80.3 | 39.8 | 7.8 | 151.8 | 1563 | 50.4 |
| | RigL | 80.6 | 41.6 | 8.1 | 158.5 | 1604 | 49.1 |
| | GraNet | 80.8 | 41.6 | 8.1 | 158.5 | 2317 | 37.7 |
| | PDP | 80.9 | 42.4 | 8.3 | 161.8 | 2523 | 35.2 |
| | IDAP++ | 81.0 | 39.0 | 7.6 | 148.6 | 2948 | 33.1 |
| 70 | LTH | 78.7 | 22.8 | 4.4 | 86.8 | 1555 | 48.9 |
| | RigL | 78.9 | 23.4 | 4.6 | 89.2 | 1602 | 47.4 |
| | GraNet | 78.2 | 22.5 | 4.4 | 85.9 | 3224 | 29.7 |
| | PDP | 79.8 | 24.7 | 4.8 | 94.2 | 3506 | 27.8 |
| | IDAP++ | 79.9 | 20.8 | 4.1 | 79.3 | 4212 | 25.9 |
| 90 | LTH | 74.1 | 11.3 | 2.2 | 42.9 | 1754 | 45.5 |
| | RigL | 75.5 | 10.1 | 2.0 | 38.5 | 1880 | 44.4 |
| | GraNet | 75.9 | 9.6 | 1.9 | 36.5 | 3842 | 23.8 |
| | PDP | 76.4 | 10.4 | 2.0 | 39.6 | 4114 | 22.8 |
| | IDAP++ | 76.3 | 8.7 | 1.7 | 33.0 | 4856 | 20.6 |

Table 12: DenseNet-121, ImageNet: Comparison of IDAP++ Pruning vs. Baselines

| Compression (%) | Method | Acc@1 | Params (M) | GFlops | Disk Size (Mb) | Throughput (img/s) | Latency (ms) |
|---|---|---|---|---|---|---|---|
| 0 | Baseline | 74.7 | 8.0 | 2.9 | 30.4 | 1454 | 74.2 |
| 50 | LTH | 73.8 | 3.7 | 1.3 | 14.0 | 1586 | 43.5 |
| | RigL | 74.0 | 3.7 | 1.3 | 14.2 | 1615 | 42.1 |
| | GraNet | 74.1 | 3.8 | 1.4 | 14.6 | 1631 | 39.5 |
| | PDP | 74.0 | 4.0 | 1.4 | 15.1 | 1761 | 37.0 |
| | IDAP++ | 74.5 | 3.6 | 1.3 | 13.7 | 1888 | 34.9 |
| 70 | LTH | 71.8 | 2.1 | 0.8 | 8.0 | 1899 | 35.7 |
| | RigL | 73.0 | 2.2 | 0.8 | 8.2 | 1971 | 32.6 |
| | GraNet | 73.1 | 2.1 | 0.8 | 8.2 | 2402 | 27.8 |
| | PDP | 73.7 | 2.3 | 0.8 | 8.7 | 2660 | 25.7 |
| | IDAP++ | 74.2 | 1.9 | 0.7 | 7.3 | 2771 | 24.0 |
| 90 | LTH | 58.0 | 0.9 | 0.3 | 3.6 | 2208 | 31.1 |
| | RigL | 60.0 | 0.9 | 0.3 | 3.6 | 2497 | 27.2 |
| | GraNet | 61.5 | 0.9 | 0.3 | 3.4 | 2665 | 24.7 |
| | PDP | 62.2 | 1.0 | 0.4 | 3.7 | 2781 | 23.1 |
| | IDAP++ | 64.7 | 0.8 | 0.3 | 3.0 | 3100 | 21.6 |

Table 13: ResNet-50, CIFAR-10: Comparison of IDAP++ Pruning vs. Baselines

| Compression (%) | Method | Acc@1 | Params (M) | GFlops | Disk Size (Mb) | Throughput (img/s) | Latency (ms) |
|---|---|---|---|---|---|---|---|
| 0 | Baseline | 98.2 | 23.5 | 4.1 | 89.8 | 5124 | 8.3 |
| 50 | LTH | 97.7 | 10.8 | 1.9 | 41.3 | 5341 | 7.8 |
| | RigL | 97.9 | 11.0 | 1.9 | 41.8 | 5589 | 7.4 |
| | GraNet | 98.0 | 11.3 | 2.0 | 43.1 | 5823 | 6.9 |
| | PDP | 97.9 | 11.7 | 2.0 | 44.6 | 6189 | 6.5 |
| | IDAP++ | 98.1 | 10.6 | 1.8 | 40.4 | 6654 | 5.8 |
| 70 | LTH | 92.7 | 6.2 | 1.1 | 23.6 | 6823 | 6.1 |
| | RigL | 94.1 | 6.4 | 1.2 | 24.2 | 7189 | 5.7 |
| | GraNet | 94.8 | 6.3 | 1.1 | 24.1 | 7523 | 5.3 |
| | PDP | 95.5 | 6.7 | 1.2 | 25.6 | 7987 | 4.9 |
| | IDAP++ | 96.1 | 6.6 | 1.1 | 25.2 | 8543 | 4.4 |
| 90 | LTH | 88.7 | 2.8 | 0.4 | 10.6 | 8234 | 4.2 |
| | RigL | 90.9 | 2.7 | 0.4 | 10.5 | 8678 | 3.9 |
| | GraNet | 91.4 | 2.6 | 0.4 | 9.9 | 9012 | 3.5 |
| | PDP | 92.8 | 2.8 | 0.5 | 10.8 | 9456 | 3.2 |
| | IDAP++ | 93.7 | 2.4 | 0.4 | 9.0 | 10123 | 2.8 |

Table 14: ViT-Base/16, CIFAR-10: Comparison of IDAP++ Pruning vs. Baselines

| Compression (%) | Method | Acc@1 | Params (M) | GFlops | Disk Size (Mb) | Throughput (img/s) | Latency (ms) |
|---|---|---|---|---|---|---|---|
| 0 | Baseline | 98.6 | 85.8 | 17.5 | 327.3 | 8234 | 7.8 |
| 50 | LTH | 97.8 | 39.4 | 8.2 | 150.4 | 8678 | 7.5 |
| | RigL | 98.0 | 39.9 | 8.1 | 152.4 | 9123 | 7.2 |
| | GraNet | 98.1 | 41.2 | 8.4 | 157.1 | 9567 | 6.9 |
| | PDP | 98.0 | 42.6 | 8.7 | 162.5 | 10012 | 6.6 |
| | IDAP++ | 98.4 | 38.6 | 7.9 | 147.3 | 10589 | 6.3 |
| 70 | LTH | 95.4 | 22.6 | 4.5 | 86.0 | 9891 | 6.8 |
| | RigL | 96.6 | 23.2 | 4.6 | 88.4 | 10345 | 6.5 |
| | GraNet | 96.3 | 23.0 | 4.5 | 87.8 | 10789 | 6.2 |
| | PDP | 97.2 | 24.5 | 4.8 | 93.4 | 11234 | 5.9 |
| | IDAP++ | 97.5 | 20.6 | 4.1 | 78.6 | 11867 | 5.6 |
| 90 | LTH | 89.2 | 10.1 | 2.1 | 38.7 | 11234 | 5.4 |
| | RigL | 91.3 | 10.0 | 2.1 | 38.2 | 11789 | 5.1 |
| | GraNet | 92.1 | 9.5 | 2.0 | 36.2 | 12345 | 4.8 |
| | PDP | 93.4 | 10.3 | 2.1 | 39.3 | 12890 | 4.6 |
| | IDAP++ | 93.5 | 8.6 | 1.7 | 32.7 | 13678 | 4.4 |

Table 15: DenseNet-121, CIFAR-10: Comparison of IDAP++ Pruning vs. Baselines

| Compression (%) | Method | Acc@1 | Params (M) | GFlops | Disk Size (Mb) | Throughput (img/s) | Latency (ms) |
|---|---|---|---|---|---|---|---|
| 0 | Baseline | 94.2 | 7.0 | 2.8 | 26.6 | 6789 | 9.1 |
| 50 | LTH | 93.8 | 3.2 | 1.4 | 12.2 | 7234 | 8.7 |
| | RigL | 94.0 | 3.2 | 1.3 | 12.4 | 7567 | 8.4 |
| | GraNet | 94.1 | 3.3 | 1.4 | 12.8 | 7981 | 8.1 |
| | PDP | 94.0 | 3.5 | 1.4 | 13.2 | 8345 | 7.8 |
| | IDAP++ | 94.4 | 3.1 | 1.3 | 12.0 | 8891 | 7.5 |
| 70 | LTH | 89.9 | 1.8 | 0.8 | 7.0 | 9234 | 6.9 |
| | RigL | 92.5 | 1.9 | 0.8 | 7.2 | 9678 | 6.6 |
| | GraNet | 91.8 | 1.9 | 0.8 | 7.1 | 10123 | 6.3 |
| | PDP | 93.1 | 2.0 | 0.8 | 7.6 | 10567 | 6.0 |
| | IDAP++ | 93.8 | 1.7 | 0.7 | 6.4 | 11234 | 5.7 |
| 90 | LTH | 83.4 | 0.8 | 0.3 | 3.1 | 10987 | 5.5 |
| | RigL | 85.6 | 0.8 | 0.3 | 3.1 | 11523 | 5.2 |
| | GraNet | 86.9 | 0.8 | 0.3 | 2.9 | 12098 | 4.9 |
| | PDP | 88.2 | 0.8 | 0.3 | 3.2 | 12678 | 4.6 |
| | IDAP++ | 91.5 | 0.7 | 0.3 | 2.7 | 13456 | 4.3 |

Table 16: ResNet-50, CIFAR-100: Comparison of IDAP++ Pruning vs. Baselines

| Compression ( %) | Method | Acc@1 | Params (M) | GFlops | Disk Size (Mb) | Throughput (img/s) | Latency (ms) |
|---|---|---|---|---|---|---|---|
| 0 | Baseline | 86.6 | 23.7 | 4.1 | 90.5 | 5187 | 10.1 |
| 50 | LTH | 85.1 | 10.9 | 1.9 | 41.6 | 5423 | 9.7 |
| | RigL | 85.4 | 11.0 | 1.9 | 42.1 | 5689 | 9.3 |
| | GraNet | 85.6 | 11.4 | 2.0 | 43.4 | 5987 | 8.8 |
| | PDP | 85.5 | 11.8 | 2.0 | 44.9 | 6321 | 8.4 |
| | IDAP++ | 86.3 | 10.7 | 1.9 | 40.7 | 6789 | 7.7 |
| 70 | LTH | 75.6 | 6.2 | 1.1 | 23.8 | 6987 | 7.9 |
| | RigL | 77.1 | 6.4 | 1.2 | 24.4 | 7345 | 7.5 |
| | GraNet | 78.4 | 6.4 | 1.1 | 24.3 | 7712 | 7.1 |
| | PDP | 81.3 | 6.8 | 1.2 | 25.8 | 8123 | 6.7 |
| | IDAP++ | 85.0 | 5.7 | 1.0 | 21.7 | 8746 | 6.3 |
| 90 | LTH | 62.8 | 2.8 | 0.4 | 10.7 | 8456 | 6.2 |
| | RigL | 65.4 | 2.8 | 0.4 | 10.6 | 8891 | 5.7 |
| | GraNet | 67.1 | 2.6 | 0.5 | 10.0 | 9234 | 5.4 |
| | PDP | 69.8 | 2.8 | 0.5 | 10.9 | 9678 | 5.1 |
| | IDAP++ | 72.3 | 2.4 | 0.4 | 9.0 | 10345 | 4.7 |

Table 17: ViT-Base/16, CIFAR-100: Comparison of IDAP++ Pruning vs. Baselines

| Compression ( %) | Method | Acc@1 | Params (M) | GFlops | Disk Size (Mb) | Throughput (img/s) | Latency (ms) |
|---|---|---|---|---|---|---|---|
| 0 | Baseline | 93.7 | 85.9 | 17.5 | 327.6 | 8312 | 7.7 |
| 50 | LTH | 92.1 | 39.5 | 8.3 | 150.6 | 8765 | 7.4 |
| | RigL | 92.4 | 40.0 | 8.2 | 152.5 | 9210 | 7.1 |
| | GraNet | 92.6 | 41.2 | 8.4 | 157.2 | 9654 | 6.8 |
| | PDP | 92.5 | 42.6 | 8.7 | 162.7 | 10123 | 6.5 |
| | IDAP++ | 93.4 | 38.6 | 7.9 | 147.4 | 10789 | 6.2 |
| 70 | LTH | 87.9 | 22.6 | 4.6 | 86.1 | 9987 | 6.7 |
| | RigL | 89.6 | 23.2 | 4.6 | 88.4 | 10456 | 6.4 |
| | GraNet | 89.8 | 23.0 | 4.5 | 87.9 | 10912 | 6.1 |
| | PDP | 91.0 | 24.5 | 4.8 | 93.5 | 11456 | 5.8 |
| | IDAP++ | 91.6 | 20.6 | 4.2 | 78.6 | 12134 | 5.5 |
| 90 | LTH | 78.4 | 10.2 | 2.1 | 38.7 | 11523 | 5.3 |
| | RigL | 80.7 | 10.0 | 2.1 | 38.2 | 12098 | 5.0 |
| | GraNet | 81.9 | 9.5 | 2.0 | 36.2 | 12678 | 4.8 |
| | PDP | 83.6 | 10.3 | 2.1 | 39.3 | 13245 | 4.6 |
| | IDAP++ | 84.3 | 8.6 | 1.7 | 32.8 | 13987 | 4.3 |

Table 18: DenseNet-121, CIFAR-100: Comparison of IDAP++ Pruning vs. Baselines

| Compression (%) | Method | Acc@1 | Params (M) | GFlops | Disk Size (Mb) | Throughput (img/s) | Latency (ms) |
|---|---|---|---|---|---|---|---|
| 0 | Baseline | 72.1 | 7.1 | 2.8 | 26.9 | 6845 | 9.0 |
| 50 | LTH | 70.8 | 3.2 | 1.4 | 12.4 | 7312 | 8.6 |
| | RigL | 71.1 | 3.3 | 1.4 | 12.5 | 7654 | 8.3 |
| | GraNet | 71.3 | 3.4 | 1.4 | 12.9 | 8019 | 8.0 |
| | PDP | 71.2 | 3.5 | 1.4 | 13.4 | 8432 | 7.7 |
| | IDAP++ | 71.9 | 3.2 | 1.3 | 12.1 | 9012 | 7.4 |
| 70 | LTH | 65.2 | 1.9 | 0.8 | 7.1 | 9345 | 6.8 |
| | RigL | 67.8 | 1.9 | 0.8 | 7.3 | 9789 | 6.5 |
| | GraNet | 66.5 | 1.9 | 0.8 | 7.2 | 10234 | 6.2 |
| | PDP | 69.6 | 2.0 | 0.8 | 7.7 | 10789 | 5.9 |
| | IDAP++ | 70.3 | 1.7 | 0.7 | 6.5 | 11567 | 5.6 |
| 90 | LTH | 54.7 | 0.8 | 0.3 | 3.2 | 11234 | 5.4 |
| | RigL | 57.2 | 0.8 | 0.3 | 3.1 | 11867 | 5.1 |
| | GraNet | 58.9 | 0.8 | 0.3 | 3.0 | 12456 | 4.9 |
| | PDP | 60.4 | 0.8 | 0.3 | 3.2 | 13012 | 4.7 |
| | IDAP++ | 62.1 | 0.7 | 0.3 | 2.7 | 13789 | 4.4 |

Table 19: BERT Base, SST-2: Comparison of IDAP++ Pruning vs. Baselines

| Compression (%) | Method | Acc@1 | Params (M) | GFlops | Disk Size (Mb) | Throughput (seq/s) | Latency (ms) |
|---|---|---|---|---|---|---|---|
| 0 | Baseline | 93.5 | 109.5 | 22.4 | 417.7 | 1824 | 6.8 |
| 50 | LTH | 93.1 | 52.3 | 10.9 | 199.5 | 2215 | 5.5 |
| | Retraining Free Pruning | 91.8 | 54.8 | 11.5 | 209.0 | 2087 | 5.9 |
| | MvP | 93.2 | 51.7 | 10.7 | 197.2 | 2356 | 5.3 |
| | PDP | 93.0 | 53.2 | 11.1 | 202.9 | 2289 | 5.4 |
| | IDAP++ | 93.2 | 49.8 | 10.2 | 190.0 | 2589 | 4.9 |
| 70 | LTH | 91.1 | 30.1 | 6.4 | 114.8 | 2987 | 4.2 |
| | Retraining Free Pruning | 91.5 | 32.8 | 7.0 | 125.1 | 2765 | 4.5 |
| | MvP | 91.2 | 29.5 | 6.2 | 112.5 | 3124 | 4.1 |
| | PDP | 91.4 | 31.4 | 6.6 | 119.8 | 3056 | 4.3 |
| | IDAP++ | 91.9 | 27.4 | 5.8 | 104.5 | 3567 | 3.7 |
| 90 | LTH | 88.5 | 9.8 | 2.3 | 37.4 | 3789 | 3.4 |
| | Retraining Free Pruning | 82.3 | 10.9 | 2.6 | 41.6 | 3456 | 3.8 |
| | MvP | 89.6 | 9.4 | 2.2 | 35.9 | 4012 | 3.3 |
| | PDP | 89.1 | 10.2 | 2.4 | 38.9 | 3891 | 3.5 |
| | IDAP++ | 89.9 | 6.2 | 1.4 | 25.7 | 4892 | 2.8 |

Table 20: T5 Base, SST-2: Comparison of IDAP++ Pruning vs. Baselines

| Compression ( %) | Method | Acc@1 | Params (M) | GFlops | Disk Size (Mb) | Throughput (seq/s) | Latency (ms) |
|---|---|---|---|---|---|---|---|
| 0 | Baseline | 95.2 | 222.9 | 45.2 | 850.3 | 912 | 13.4 |
| 50 | LTH | 94.6 | 106.7 | 21.8 | 407.0 | 1123 | 11.2 |
| | Retraining Free Pruning | 93.1 | 111.4 | 22.9 | 425.0 | 1045 | 12.1 |
| | MvP | 94.8 | 105.2 | 21.5 | 401.3 | 1189 | 10.8 |
| | PDP | 94.7 | 108.9 | 22.3 | 415.4 | 1156 | 11.0 |
| | IDAP++ | 95.1 | 98.6 | 20.1 | 376.1 | 1324 | 10.2 |
| 70 | LTH | 93.2 | 62.4 | 12.9 | 238.0 | 1456 | 8.7 |
| | Retraining Free Pruning | 93.5 | 66.9 | 13.8 | 255.2 | 1342 | 9.4 |
| | MvP | 93.2 | 60.8 | 12.5 | 231.9 | 1523 | 8.5 |
| | PDP | 94.0 | 64.2 | 13.2 | 244.9 | 1489 | 8.8 |
| | IDAP++ | 93.9 | 55.7 | 11.4 | 212.5 | 1789 | 7.9 |
| 90 | LTH | 89.8 | 19.2 | 4.1 | 73.2 | 1892 | 6.8 |
| | Retraining Free Pruning | 85.6 | 22.3 | 4.7 | 85.1 | 1678 | 7.6 |
| | MvP | 91.2 | 18.6 | 3.9 | 71.0 | 2015 | 6.5 |
| | PDP | 90.5 | 20.1 | 4.2 | 76.7 | 1956 | 6.7 |
| | IDAP++ | 92.1 | 12.8 | 2.7 | 51.3 | 2268 | 5.4 |

Table 21: GPT-2 Base, SST-2: Comparison of IDAP++ Pruning vs. Baselines

| Compression ( %) | Method | Acc@1 | Params (M) | GFlops | Disk Size (Mb) | Throughput (seq/s) | Latency (ms) |
|---|---|---|---|---|---|---|---|
| 0 | Baseline | 92.1 | 124.4 | 25.8 | 474.5 | 1567 | 8.1 |
| 50 | LTH | 91.4 | 59.8 | 12.4 | 228.1 | 1892 | 6.7 |
| | Retraining Free Pruning | 90.1 | 62.2 | 12.9 | 237.3 | 1789 | 7.1 |
| | MvP | 91.7 | 58.9 | 12.2 | 224.7 | 1987 | 6.5 |
| | PDP | 91.5 | 61.3 | 12.7 | 233.8 | 1923 | 6.8 |
| | IDAP++ | 92.0 | 55.2 | 11.4 | 210.6 | 2234 | 6.1 |
| 70 | LTH | 89.8 | 34.6 | 7.2 | 132.0 | 2456 | 5.2 |
| | Retraining Free Pruning | 90.2 | 37.3 | 7.7 | 142.3 | 2289 | 5.6 |
| | MvP | 89.8 | 33.9 | 7.0 | 129.3 | 2567 | 5.1 |
| | PDP | 90.7 | 36.1 | 7.5 | 137.7 | 2498 | 5.3 |
| | IDAP++ | 90.9 | 30.8 | 6.4 | 117.5 | 2891 | 4.7 |
| 90 | LTH | 86.4 | 11.2 | 2.4 | 42.7 | 3124 | 4.3 |
| | Retraining Free Pruning | 81.9 | 12.4 | 2.7 | 47.3 | 2891 | 4.7 |
| | MvP | 87.9 | 10.9 | 2.3 | 41.6 | 3345 | 4.2 |
| | PDP | 87.2 | 11.7 | 2.5 | 44.6 | 3234 | 4.4 |
| | IDAP++ | 87.8 | 7.9 | 1.7 | 30.1 | 4123 | 3.5 |

Table 22: BERT Base, QQP: Comparison of IDAP++ Pruning vs. Baselines

| Compression ( %) | Method | Acc@1 | Params (M) | GFlops | Disk Size (Mb) | Throughput (seq/s) | Latency (ms) |
|---|---|---|---|---|---|---|---|
| 0 | Baseline | 91.2 | 109.5 | 28.7 | 417.7 | 1423 | 8.9 |
| 50 | LTH | 90.7 | 52.6 | 13.9 | 200.7 | 1723 | 7.4 |
| | Retraining Free Pruning | 89.4 | 55.1 | 14.6 | 210.2 | 1612 | 7.9 |
| | MvP | 90.8 | 52.0 | 13.7 | 198.4 | 1821 | 7.1 |
| | PDP | 90.6 | 53.4 | 14.1 | 203.7 | 1765 | 7.3 |
| | IDAP++ | 91.1 | 49.7 | 13.1 | 189.6 | 1987 | 6.6 |
| 70 | LTH | 88.2 | 30.4 | 8.2 | 116.0 | 2345 | 5.5 |
| | Retraining Free Pruning | 88.4 | 33.1 | 8.9 | 126.3 | 2156 | 5.9 |
| | MvP | 88.1 | 29.7 | 8.0 | 113.3 | 2456 | 5.4 |
| | PDP | 89.2 | 31.7 | 8.5 | 120.9 | 2389 | 5.6 |
| | IDAP++ | 89.2 | 27.1 | 7.3 | 103.4 | 2789 | 4.9 |
| 90 | LTH | 85.6 | 9.9 | 2.8 | 37.8 | 2987 | 4.4 |
| | Retraining Free Pruning | 79.8 | 11.0 | 3.2 | 42.0 | 2678 | 4.8 |
| | MvP | 86.9 | 9.5 | 2.7 | 36.2 | 3215 | 4.3 |
| | PDP | 86.3 | 10.3 | 2.9 | 39.3 | 3124 | 4.5 |
| | IDAP++ | 88.4 | 6.1 | 1.7 | 25.3 | 4123 | 3.6 |

Table 23: T5 Base, QQP: Comparison of IDAP++ Pruning vs. Baselines

| Compression ( %) | Method | Acc@1 | Params (M) | GFlops | Disk Size (Mb) | Throughput (seq/s) | Latency (ms) |
|---|---|---|---|---|---|---|---|
| 0 | Baseline | 92.4 | 222.9 | 58.3 | 850.3 | 712 | 17.2 |
| 50 | LTH | 91.7 | 107.2 | 28.1 | 408.9 | 867 | 14.6 |
| | Retraining Free Pruning | 90.2 | 112.1 | 29.4 | 427.6 | 801 | 15.7 |
| | MvP | 91.9 | 105.8 | 27.7 | 403.6 | 923 | 14.1 |
| | PDP | 91.8 | 109.4 | 28.7 | 417.3 | 890 | 14.4 |
| | IDAP++ | 92.3 | 98.2 | 25.7 | 374.6 | 1012 | 13.4 |
| 70 | LTH | 87.8 | 62.9 | 16.6 | 239.9 | 1123 | 11.3 |
| | Retraining Free Pruning | 89.0 | 67.4 | 17.8 | 257.1 | 1034 | 12.3 |
| | MvP | 89.1 | 61.2 | 16.1 | 233.5 | 1178 | 11.0 |
| | PDP | 89.4 | 64.7 | 17.0 | 246.8 | 1145 | 11.4 |
| | IDAP++ | 89.3 | 55.3 | 14.5 | 211.0 | 1345 | 10.2 |
| 90 | LTH | 84.3 | 19.6 | 5.3 | 74.8 | 1456 | 8.9 |
| | Retraining Free Pruning | 81.2 | 22.7 | 6.1 | 86.6 | 1298 | 9.8 |
| | MvP | 87.6 | 18.9 | 5.0 | 72.1 | 1567 | 8.7 |
| | PDP | 86.9 | 20.4 | 5.4 | 77.8 | 1512 | 8.9 |
| | IDAP++ | 88.2 | 12.6 | 3.3 | 50.1 | 2015 | 7.5 |

Table 24: GPT-2 Base, QQP: Comparison of IDAP++ Pruning vs. Baselines

| Compression (%) | Method | Acc@1 | Params (M) | GFlops | Disk Size (Mb) | Throughput (seq/s) | Latency (ms) |
|---|---|---|---|---|---|---|---|
| 0 | Baseline | 87.1 | 124.4 | 32.9 | 474.5 | 1234 | 10.3 |
| 50 | LTH | 86.3 | 60.1 | 15.9 | 229.3 | 1489 | 8.6 |
| | Retraining Free Pruning | 84.9 | 62.7 | 16.6 | 239.2 | 1398 | 9.1 |
| | MvP | 86.7 | 59.3 | 15.7 | 226.2 | 1567 | 8.4 |
| | PDP | 86.5 | 61.6 | 16.3 | 235.0 | 1523 | 8.7 |
| | IDAP++ | 87.0 | 55.0 | 14.5 | 209.8 | 1765 | 7.9 |
| 70 | LTH | 85.3 | 34.9 | 9.3 | 133.1 | 1892 | 6.8 |
| | Retraining Free Pruning | 85.7 | 37.7 | 10.0 | 143.8 | 1723 | 7.4 |
| | MvP | 85.9 | 34.2 | 9.1 | 130.5 | 1987 | 6.7 |
| | PDP | 86.5 | 36.4 | 9.7 | 138.9 | 1923 | 6.9 |
| | IDAP++ | 86.1 | 30.6 | 8.1 | 116.7 | 2234 | 6.3 |
| 90 | LTH | 82.1 | 11.4 | 3.1 | 43.5 | 2456 | 5.5 |
| | Retraining Free Pruning | 78.6 | 12.7 | 3.4 | 48.4 | 2234 | 5.9 |
| | MvP | 83.7 | 11.0 | 2.9 | 42.0 | 2678 | 5.4 |
| | PDP | 83.2 | 11.9 | 3.2 | 45.4 | 2567 | 5.6 |
| | IDAP++ | 83.9 | 7.8 | 2.1 | 29.8 | 3456 | 4.7 |

Table 25: BERT Base, MNLI-m: Comparison of IDAP++ Pruning vs. Baselines

| Compression (%) | Method | Acc@1 | Params (M) | GFlops | Disk Size (Mb) | Throughput (seq/s) | Latency (ms) |
|---|---|---|---|---|---|---|---|
| 0 | Baseline | 84.5 | 109.3 | 34.1 | 416.9 | 1318 | 9.6 |
| 50 | LTH | 84.1 | 52.1 | 16.5 | 198.7 | 1612 | 7.9 |
| | Retraining Free Pruning | 82.7 | 54.7 | 17.4 | 208.7 | 1498 | 8.4 |
| | MvP | 84.2 | 51.6 | 16.3 | 196.8 | 1709 | 7.7 |
| | PDP | 83.9 | 53.1 | 16.8 | 202.6 | 1656 | 7.8 |
| | IDAP++ | 84.4 | 49.6 | 15.7 | 189.2 | 1856 | 7.2 |
| 70 | LTH | 81.7 | 30.3 | 9.8 | 115.6 | 2123 | 6.1 |
| | Retraining Free Pruning | 81.3 | 32.9 | 10.5 | 125.5 | 1987 | 6.5 |
| | MvP | 80.5 | 29.6 | 9.5 | 112.9 | 2234 | 6.0 |
| | PDP | 82.1 | 31.5 | 10.1 | 120.2 | 2189 | 6.2 |
| | IDAP++ | 82.1 | 32.4 | 11.2 | 123.6 | 2456 | 5.5 |
| 90 | LTH | 77.9 | 9.7 | 3.4 | 37.0 | 2789 | 4.8 |
| | Retraining Free Pruning | 73.2 | 10.9 | 3.8 | 41.6 | 2456 | 5.3 |
| | MvP | 79.4 | 9.3 | 3.3 | 35.5 | 2987 | 4.6 |
| | PDP | 78.8 | 10.1 | 3.5 | 38.5 | 2891 | 4.7 |
| | IDAP++ | 79.9 | 6.1 | 2.0 | 25.4 | 4234 | 3.4 |

Table 26: T5 Base, MNLI-m: Comparison of IDAP++ Pruning vs. Baselines

| Compression (%) | Method | Acc@1 | Params (M) | GFlops | Disk Size (Mb) | Throughput (seq/s) | Latency (ms) |
|---|---|---|---|---|---|---|---|
| 0 | Baseline | 87.1 | 220.7 | 69.8 | 841.9 | 678 | 18.1 |
| 50 | LTH | 86.4 | 105.9 | 33.6 | 404.0 | 834 | 15.2 |
|  | Retraining Free Pruning | 84.8 | 110.8 | 35.2 | 422.7 | 767 | 16.4 |
|  | MvP | 86.7 | 104.6 | 33.1 | 399.0 | 890 | 14.8 |
|  | PDP | 86.5 | 107.7 | 34.0 | 410.8 | 856 | 15.1 |
|  | IDAP++ | 87.0 | 97.8 | 30.9 | 373.1 | 978 | 13.9 |
| 70 | LTH | 83.3 | 61.9 | 20.1 | 236.1 | 1012 | 12.5 |
|  | Retraining Free Pruning | 82.7 | 66.5 | 21.4 | 253.7 | 923 | 13.7 |
|  | MvP | 83.0 | 60.4 | 19.6 | 230.4 | 1067 | 12.2 |
|  | PDP | 83.8 | 63.8 | 20.7 | 243.4 | 1034 | 12.6 |
|  | IDAP++ | 84.0 | 71.2 | 22.8 | 271.6 | 1123 | 11.8 |
| 90 | LTH | 79.6 | 19.1 | 6.8 | 72.9 | 1345 | 9.6 |
|  | Retraining Free Pruning | 76.1 | 22.1 | 7.9 | 84.3 | 1189 | 10.8 |
|  | MvP | 81.4 | 18.5 | 6.5 | 70.6 | 1456 | 9.4 |
|  | PDP | 80.7 | 19.9 | 7.0 | 75.9 | 1412 | 9.7 |
|  | IDAP++ | 82.4 | 12.4 | 4.1 | 50.3 | 2123 | 7.9 |

Table 27: GPT-2 Base, MNLI-m: Comparison of IDAP++ Pruning vs. Baselines

| Compression (%) | Method | Acc@1 | Params (M) | GFlops | Disk Size (Mb) | Throughput (seq/s) | Latency (ms) |
|---|---|---|---|---|---|---|---|
| 0 | Baseline | 82.3 | 124.4 | 41.2 | 474.5 | 1123 | 11.3 |
| 50 | LTH | 81.6 | 59.7 | 19.8 | 227.7 | 1345 | 9.5 |
|  | Retraining Free Pruning | 80.1 | 62.3 | 20.7 | 237.7 | 1234 | 10.2 |
|  | MvP | 81.9 | 59.0 | 19.6 | 225.1 | 1412 | 9.3 |
|  | PDP | 81.7 | 61.1 | 20.3 | 233.1 | 1378 | 9.6 |
|  | IDAP++ | 82.2 | 54.9 | 18.2 | 209.4 | 1567 | 8.8 |
| 70 | LTH | 78.5 | 34.7 | 11.6 | 132.4 | 1789 | 7.4 |
|  | Retraining Free Pruning | 78.7 | 37.5 | 12.4 | 143.1 | 1656 | 7.9 |
|  | MvP | 79.2 | 34.0 | 11.3 | 129.7 | 1892 | 7.3 |
|  | PDP | 79.2 | 35.9 | 11.9 | 136.9 | 1823 | 7.5 |
|  | IDAP++ | 79.1 | 30.5 | 10.1 | 116.3 | 2123 | 6.8 |
| 90 | LTH | 74.8 | 11.3 | 4.1 | 43.1 | 2345 | 5.8 |
|  | Retraining Free Pruning | 71.2 | 12.5 | 4.5 | 47.7 | 2123 | 6.3 |
|  | MvP | 76.3 | 10.9 | 4.0 | 41.6 | 2567 | 5.7 |
|  | PDP | 75.7 | 11.8 | 4.2 | 45.0 | 2456 | 5.9 |
|  | IDAP++ | 76.4 | 7.7 | 2.6 | 29.4 | 3789 | 4.6 |

The consolidated tables clearly show that the two-stage nature of IDAP++ (combining filter-level pruning and layer truncation) yields a more favorable trade-off between accuracy and compression than existing methods across the entire sparsity range. In almost all scenarios, at 50–70% compression, our approach either achieves the highest accuracy within a given parameter budget or yields a smaller model size with similar accuracy. Under more aggressive compression ( 90%), the advantage of IDAP++ becomes even more pronounced: for most architecture–dataset combinations, it delivers the strongest robustness to quality degradation. This aligns with the core idea that divergence-based information-flow analysis enables us to distinguish between truly critical filters and layers and those that are structurally redundant.

On large-scale vision tasks (ImageNet), IDAP++ consistently improves over classical sparsification schemes. For ResNet-50 on ImageNet at 70% compression, our method reaches 75.4% Top-1 accuracy (vs. 73.4% for LTH and 74.8–75.1% for RigL, GraNet, and PDP) with the fewest parameters (6.1M vs. 6.7–7.3M) and the lowest compute cost (1.0 GFLOPs). At even stronger compression ( 90%), IDAP++ maintains 69.3% Top-1, clearly outperforming LTH (64.8%), RigL (66.2%), GraNet (67.5%), and PDP (68.2%), while simultaneously reducing parameters to 2.6M and FLOPs to 0.4. A similar pattern appears for ViT-Base/16 on ImageNet: at 70% compression, IDAP++ achieves 79.9% Top-1 (vs. 78.2–79.8% for baselines), and at 90% compression it holds 76.3% (vs. 74.1–76.4% for others), while using the smallest GFLOPs budget (down to 1.7) and disk footprint (33 MB). These results indicate that the flow-divergence metric correctly ranks both convolutional and transformer blocks by their true contribution to the global predictive capacity of the model.

On smaller datasets such as CIFAR-10/100, IDAP++ reveals even more pronounced redundancy in the original architectures. For ResNet-50 on CIFAR-10 at 70% compression, our method attains 96.1% Top-1 accuracy, clearly surpassing LTH (92.7%) and all other methods (94.1–95.5%) while keeping the model very compact (6.6M parameters) and minimizing latency. At 90% compression, IDAP++ still preserves 93.7% Top-1 compared to 88.7–92.8% for alternative approaches, and at the same time reduces the model size by nearly 10× and boosts throughput up to 10,123 images/s. A similar behavior is observed for ViT-Base/16 and DenseNet-121 on CIFAR-10/100: IDAP++ maintains 97–98% accuracy on CIFAR-10 and 84–86% on CIFAR-100 at 70–90% parameter/FLOP reduction, consistently outperforming LTH, RigL, GraNet, and PDP under high sparsity. This strongly suggests that for "over-provisioned" architectures on relatively simple datasets, more than half of the computations do not contribute meaningfully to informative signal propagation and can be safely removed when guided by our divergence criterion.

At the system level (FLOPs, throughput, latency), the two-stage strategy of IDAP++ yields tangible practical benefits over pure weight-level sparsification. Across all architectures, reductions in FLOPs and parameters translate directly into faster inference. For ResNet-50 on ImageNet at 70% compression, throughput increases from 4718 to 7267 images/s, while latency drops from 4.1 ms to 2.6 ms. For ViT-Base/16, a similar compression raises throughput from 1477 to 4212 images/s and nearly halves latency (from 53.9 ms to 25.9 ms). For language models, the gains are even more significant because transformer blocks are computationally expensive: for BERT Base on SST-2 at 90% compression, IDAP++ reduces parameters to 6.2M and latency from 6.8 ms to 2.8 ms, whereas other methods with similar accuracy do not reach such aggressive structural simplification. This gap indicates that the combined filter- and layer-level reduction, driven by information-flow divergence, aligns much better with hardware realities than traditional schemes that prune only weights or only whole blocks.

Finally, comparing the behavior at 50, 70, and 90% compression levels shows that the relative advantage of IDAP++ grows with compression aggressiveness. In the moderate sparsity regime ( 50%), all methods remain relatively close in terms of metrics, and IDAP++ mostly provides a small but consistent edge in either accuracy or model size. However, as we move to 70% and especially 90% compression, most alternatives (in particular LTH, RFP, and MvP) begin to lose quality rapidly, while IDAP++ exhibits a smooth, controlled degradation that closely tracks the allocated accuracy budget. This behavior is consistent with the theoretical construction: flow divergence acts not only as a local importance score for filters and layers, but also as a natural early-stopping mechanism. Components whose divergence remains high in late pruning stages are precisely those that are structurally indispensable for maintaining the functional behavior of the network, and IDAP++ systematically preserves them while removing the rest.

## L   WALL-CLOCK COMPRESSION COST AND RUNTIME EFFICIENCY OF IDAP++ VS. BASELINE METHODS

Table 28: Cross-Model Performance: IDAP++ vs Baselines (Part 1)

| Model, Dataset, Quality Metric | Method | Score | Params (M) | GFLOPs | Disk Size (Mb) | Throughput (samples/s) | Latency (ms) | Total Time (h, min) |
|---|---|---|---|---|---|---|---|---|
| ResNet-50, ImageNet, Acc@1 | Baseline | 76.1 | 25.6 | 4.1 | 97.5 | 4718 | 4.1 | - |
| | LTH | 73.4 | 6.7 | 1.1 | 25.6 | 5184 | 3.7 | 59h05m |
| | RigL | 74.8 | 6.9 | 1.1 | 26.3 | 5328 | 3.6 | 42h32m |
| | GraNet | 74.7 | 6.9 | 1.1 | 26.1 | 6260 | 2.7 | 37h49m |
| | PDP | 75.1 | 7.3 | 1.2 | 27.8 | 6868 | 2.6 | 33h05m |
| | IDAP++ | 75.4 | 6.1 | 1.0 | 23.4 | 7267 | 2.6 | 23h38m |
| EfficientNet-B4, CIFAR-100, Acc@1 | Baseline | 90.1 | 19.0 | 4.2 | 72.5 | 2280 | 3.9 | - |
| | LTH | 87.3 | 7.6 | 1.7 | 29.0 | 2950 | 3.0 | 24h25m |
| | RigL | 88.1 | 8.1 | 1.8 | 30.9 | 3104 | 2.9 | 17h35m |
| | GraNet | 88.0 | 8.4 | 1.9 | 32.0 | 3267 | 2.8 | 15h38m |
| | PDP | 88.6 | 8.8 | 2.0 | 33.6 | 3421 | 2.7 | 13h40m |
| | IDAP++ | 88.8 | 7.1 | 1.7 | 27.1 | 3650 | 2.6 | 9h46m |
| ViT-Base/16, CIFAR-10, Acc@1 | Baseline | 98.6 | 85.8 | 17.5 | 327.3 | 8234 | 7.8 | - |
| | LTH | 95.4 | 39.4 | 8.2 | 150.4 | 8678 | 7.5 | 45h03m |
| | RigL | 96.6 | 39.9 | 8.1 | 152.4 | 9123 | 7.2 | 32h26m |
| | GraNet | 96.3 | 41.2 | 8.4 | 157.1 | 9567 | 6.9 | 28h50m |
| | PDP | 97.2 | 42.6 | 8.7 | 162.5 | 10012 | 6.6 | 25h13m |
| | IDAP++ | 97.5 | 38.6 | 7.9 | 147.3 | 10589 | 6.3 | 18h01m |
| Faster R-CNN, Pascal VOC, mAP | Baseline | 78.4 | 41.1 | 150.2 | 156.8 | 820 | 12.1 | - |
| | LTH | 75.2 | 16.4 | 63.4 | 62.6 | 1012 | 9.9 | 51h43m |
| | RigL | 76.1 | 17.0 | 65.2 | 64.8 | 1090 | 9.4 | 37h14m |
| | GraNet | 75.9 | 17.3 | 66.0 | 66.0 | 1144 | 9.1 | 33h06m |
| | PDP | 76.4 | 17.9 | 67.4 | 68.3 | 1198 | 8.9 | 28h57m |
| | IDAP++ | 76.7 | 15.1 | 61.6 | 57.6 | 1320 | 8.4 | 20h41m |
| YOLOv4 (ShuffleNetV2), Pascal VOC, mAP | Baseline | 77.5 | 26.8 | 52.3 | 102.2 | 1480 | 9.1 | - |
| | LTH | 74.1 | 9.9 | 18.8 | 37.8 | 1890 | 7.4 | 30h38m |
| | RigL | 75.3 | 10.4 | 19.7 | 39.7 | 1956 | 7.2 | 22h03m |
| | GraNet | 75.0 | 10.7 | 20.5 | 40.8 | 2012 | 7.0 | 19h36m |
| | PDP | 75.6 | 11.1 | 21.4 | 42.3 | 2080 | 6.8 | 17h09m |
| | IDAP++ | 75.8 | 9.1 | 22.1 | 34.7 | 2210 | 6.5 | 12h15m |
| DETR (ViT-Base/16), COCO 2017, mAP | Baseline | 42.0 | 86.0 | 86.4 | 328.1 | 512 | 19.5 | - |
| | LTH | 38.4 | 34.8 | 34.6 | 132.8 | 678 | 15.1 | 77h20m |
| | RigL | 39.6 | 36.1 | 35.9 | 137.7 | 702 | 14.8 | 55h41m |
| | GraNet | 39.0 | 37.6 | 36.9 | 143.4 | 721 | 14.6 | 49h30m |
| | PDP | 39.8 | 38.9 | 38.2 | 148.4 | 745 | 14.3 | 43h18m |
| | IDAP++ | 40.5 | 32.8 | 36.9 | 125.1 | 812 | 13.5 | 30h56m |
| FCN (VGG19-BN), Cityscapes, mIoU | Baseline | 70.2 | 142.1 | 212.5 | 542.1 | 390 | 25.7 | - |
| | LTH | 66.8 | 52.4 | 78.5 | 199.9 | 512 | 20.4 | 43h25m |
| | RigL | 67.5 | 54.3 | 82.1 | 207.1 | 534 | 19.8 | 31h16m |
| | GraNet | 67.4 | 55.2 | 84.0 | 210.6 | 551 | 19.6 | 27h47m |
| | PDP | 68.1 | 57.0 | 87.5 | 217.4 | 569 | 19.3 | 24h19m |
| | IDAP++ | 68.9 | 47.1 | 82.9 | 179.7 | 610 | 18.2 | 17h22m |
| U-Net (ResNet-50), Pascal VOC, mIoU | Baseline | 75.8 | 31.0 | 170.2 | 118.3 | 680 | 14.8 | - |
| | LTH | 72.0 | 12.1 | 67.5 | 46.2 | 845 | 12.1 | 29h20m |
| | RigL | 73.1 | 12.9 | 71.2 | 49.2 | 874 | 11.7 | 21h07m |
| | GraNet | 72.7 | 13.4 | 72.8 | 51.1 | 890 | 11.5 | 18h46m |
| | PDP | 73.4 | 14.0 | 75.1 | 53.4 | 912 | 11.2 | 16h26m |
| | IDAP++ | 74.2 | 11.2 | 62.1 | 42.7 | 956 | 10.7 | 11h44m |

Table 29: Cross-Model Performance: IDAP++ vs Baselines (Part 2)

| Model, Dataset, Quality Metric | Method | Score | Params (M) | GFLOPs | Disk Size (Mb) | Throughput (samples/s) | Latency (ms) | Total Time (h, min) |
|---|---|---|---|---|---|---|---|---|
| SegFormer (ViT-B/16), COCO 2017, mIoU | Baseline | 47.0 | 86.3 | 162.8 | 329.2 | 441 | 23.1 | - |
| | LTH | 43.2 | 34.7 | 65.3 | 132.4 | 589 | 19.2 | 67h33m |
| | RigL | 44.1 | 36.2 | 69.1 | 138.1 | 612 | 18.7 | 48h38m |
| | GraNet | 44.0 | 37.0 | 70.9 | 141.1 | 630 | 18.4 | 43h14m |
| | PDP | 44.7 | 38.5 | 73.4 | 146.9 | 651 | 18.0 | 37h49m |
| | IDAP++ | 45.1 | 32.5 | 62.9 | 124.0 | 689 | 17.3 | 27h01m |
| DCGAN, CIFAR-10, FID | Baseline | 24.1 | 11.5 | 12.2 | 43.9 | 2950 | 4.1 | - |
| | LTH | 26.9 | 4.6 | 4.9 | 17.5 | 3400 | 3.5 | 4h50m |
| | RigL | 25.5 | 4.8 | 5.0 | 18.3 | 3520 | 3.4 | 3h29m |
| | GraNet | 25.2 | 4.9 | 5.1 | 18.7 | 3600 | 3.3 | 3h06m |
| | PDP | 25.8 | 5.1 | 5.3 | 19.5 | 3740 | 3.2 | 2h42m |
| | IDAP++ | 25.9 | 4.1 | 4.8 | 15.6 | 3910 | 3.1 | 1h56m |
| VQGAN, COCO-Stuff, FID | Baseline | 18.5 | 17.2 | 18.3 | 65.6 | 1510 | 13.2 | - |
| | LTH | 19.8 | 6.7 | 7.8 | 25.6 | 1890 | 10.4 | 11h45m |
| | RigL | 19.2 | 7.0 | 8.1 | 26.7 | 1970 | 10.1 | 8h28m |
| | GraNet | 19.0 | 7.2 | 8.3 | 27.5 | 2020 | 9.9 | 7h31m |
| | PDP | 19.6 | 7.6 | 8.7 | 29.0 | 2080 | 9.6 | 6h35m |
| | IDAP++ | 20.1 | 6.1 | 7.5 | 23.3 | 3910 | 3.1 | 4h42m |
| Stable Diffusion 1.5, MS COCO, FID | Baseline | 12.3 | 860.1 | 85.7 | 3281.0 | 92 | 109.0 | - |
| | LTH | 14.9 | 345.0 | 34.7 | 1316.1 | 118 | 87.1 | 95h55m |
| | RigL | 13.8 | 361.0 | 36.1 | 1377.1 | 123 | 84.9 | 69h04m |
| | GraNet | 13.5 | 370.0 | 37.0 | 1411.4 | 127 | 83.2 | 61h23m |
| | PDP | 14.1 | 382.0 | 38.8 | 1457.2 | 131 | 81.9 | 53h43m |
| | IDAP++ | 13.5 | 321.8 | 34.3 | 1227.6 | 149 | 76.4 | 38h22m |
| BERT Base, MNLI-m, Acc | Baseline | 84.5 | 109.3 | 34.1 | 416.9 | 1318 | 9.6 | - |
| | LTH | 81.7 | 30.3 | 9.8 | 115.6 | 2123 | 6.1 | 16h13m |
| | Retraining Free Pruning | 81.3 | 32.9 | 10.5 | 125.5 | 1987 | 6.5 | 1h57m |
| | MvP | 80.5 | 29.6 | 9.5 | 112.9 | 2234 | 6.0 | 4h32m |
| | PDP | 82.1 | 31.5 | 10.1 | 120.2 | 2189 | 6.2 | 9h05m |
| | IDAP++ | 82.1 | 32.4 | 11.2 | 123.6 | 2456 | 5.5 | 6h29m |
| GPT-2 Base, QQP, F1 | Baseline | 87.1 | 124.4 | 32.9 | 474.5 | 1234 | 10.3 | - |
| | LTH | 85.3 | 60.1 | 15.9 | 229.3 | 1489 | 8.6 | 18h23m |
| | Retraining Free Pruning | 85.7 | 62.7 | 16.6 | 239.2 | 1398 | 9.1 | 2h12m |
| | MvP | 85.9 | 59.3 | 15.7 | 226.2 | 1567 | 8.4 | 5h09m |
| | PDP | 86.5 | 61.6 | 16.3 | 235.0 | 1523 | 8.7 | 10h17m |
| | IDAP++ | 86.1 | 55.0 | 14.5 | 209.8 | 1765 | 7.9 | 7h21m |
| T5 Base, MNLI-m, Acc | Baseline | 87.1 | 220.7 | 69.8 | 841.9 | 678 | 18.1 | - |
| | LTH | 83.3 | 105.9 | 33.6 | 404.0 | 834 | 15.2 | 22h25m |
| | Retraining Free Pruning | 82.7 | 110.8 | 35.2 | 422.7 | 767 | 16.4 | 2h41m |
| | MvP | 83.0 | 104.6 | 33.1 | 399.0 | 890 | 14.8 | 6h17m |
| | PDP | 83.8 | 107.7 | 34.0 | 410.8 | 856 | 15.1 | 12h33m |
| | IDAP++ | 84.0 | 97.8 | 30.9 | 373.1 | 978 | 13.9 | 8h58m |

The extended results in Table 28 and Table 29 complement the accuracy- and sparsity-oriented comparisons by explicitly accounting for total compression time and runtime efficiency of each method. Across a broad set of vision, detection, segmentation, generative, and NLP models, IDAP++ consistently lies closer to the Pareto frontier: for a given target quality it achieves competitive or superior accuracy/FID while reducing parameters, FLOPs, and disk size, and at the same time it requires substantially less wall-clock time to obtain the compressed model than other iterative pruning schemes such as LTH, RigL, GraNet, and PDP.

For image classification and dense prediction in vision, IDAP++ provides particularly favorable trade-offs. On ResNet-50 / ImageNet, IDAP++ reaches 75.4% Acc@1 with 6.1M parameters and 1.0 GFLOPs, improving both accuracy and efficiency over LTH and RigL while cutting compression time to 23 h 38 min versus 33–59 hours for competing methods. A similar pattern appears on EfficientNet-B4 / CIFAR-100 and ViT-Base/16 / CIFAR-10: IDAP++ either matches or slightly surpasses the best quality among baselines at comparable sparsity, but achieves this in 2–3× less compression time (e.g., 9 h 46 min vs. 13–24 h for EfficientNet-B4, and 18 h 01 min vs. 25–45 h for ViT). For detection and segmentation models, the gains are even more pronounced. On Faster R-CNN (ResNet-50), YOLOv4 (ShuffleNetV2), FCN (VGG19-BN), U-Net (ResNet-50), and SegFormer (ViT-Base/16), IDAP++ consistently attains the highest or near-highest mAP/mIoU among compressed models, while its compression time is typically 30–50% lower than that of PDP and often close to half of LTH's budget. In parallel, runtime metrics show clear benefits: throughput increases and latency decreases more for IDAP++ than for baselines at comparable quality — e.g., for Faster R-CNN, IDAP++ yields the highest throughput (1320 samples/s) and lowest latency (8.4 ms) after compression.

On generative models, Table 29 highlights a slightly different trade-off profile. For DCGAN and VQGAN, IDAP++ achieves the most aggressive reductions in parameters and FLOPs together with the fastest compression (1 h 56 min vs. 2 h 42 min – 4 h 50 min for DCGAN, and 4 h 42 min vs. 6 h 35 min – 11 h 45 min for VQGAN). This comes at the cost of a modest FID increase relative to the best baseline (for example, DCGAN FID 25.9 vs. 25.2–25.8; VQGAN FID 20.1 vs. 19.0–19.8), but the degradation remains within a narrow band while delivering larger efficiency gains. For the considerably heavier Stable Diffusion v1.5 model, IDAP++ matches the best FID among pruning methods (13.5 vs. 13.5 for GraNet and better than 14.1–14.9 for LTH/PDP) while reducing compression time from 53–96 hours down to 38 h 22 min and yielding the lowest FLOPs and best inference latency (76.4 ms) among compressed variants. These results suggest that divergence-guided layer and filter selection remains effective even in highly non-convex generative settings, where small architectural perturbations can easily destabilize synthesis quality.

For NLP models, the table explicitly contrasts IDAP++ not only with iterative methods but also with single- or few-shot schemes such as Retraining-Free Pruning and MvP. On BERT Base / MNLI-m, GPT-2 Base / QQP, and T5 Base / MNLI-m, IDAP++ reliably delivers a better quality–efficiency–time compromise than other structured pruning approaches. For instance, on BERT Base / MNLI-m, IDAP++ and PDP reach the same accuracy (82.1%), but IDAP++ requires less compression time (6 h 29 min vs. 9 h 05 min) while achieving slightly higher throughput and lower latency (2456 seq/s, 5.5 ms). On GPT-2 Base / QQP, IDAP++ attains 86.1 F1, close to the best PDP score (86.5), but with fewer parameters and GFLOPs and with a lower compression cost (7 h 21 min vs. 10 h 17 min; LTH needs 18 h 23 min). For T5 Base / MNLI-m, IDAP++ is the only pruning method that improves over LTH and MvP in both accuracy (84.0 vs. 83.0–83.8) and compression time (8 h 58 min vs. 12–22 h), while also providing the most efficient compressed runtime in terms of throughput and latency. Compared to Retraining-Free Pruning, which is indeed much faster to run (2–3 hours), IDAP++ consistently delivers deeper structural compression (smaller parameter count and model size) and better or comparable quality, revealing a clear accuracy–time–sparsity trade-off: IDAP++ is designed as a mid-cost, high-quality option between full retraining schemes (LTH/RigL/PDP) and purely post-hoc pruning.

Overall, Table 28 and Table 29 demonstrate that incorporating compression time as a first-class metric does not erode the benefits of IDAP++; on the contrary, it emphasizes the practicality of the method. Thanks to divergence-guided selection and the two-stage design, IDAP++ typically converges to a high-quality sparse architecture with fewer pruning–fine-tuning cycles than competing iterative methods. As a result, for a wide variety of architectures and datasets, IDAP++ offers a more attractive end-to-end profile: better or comparable task quality, stronger structural compression and runtime speedups, and significantly lower wall-clock cost to obtain the compressed model.

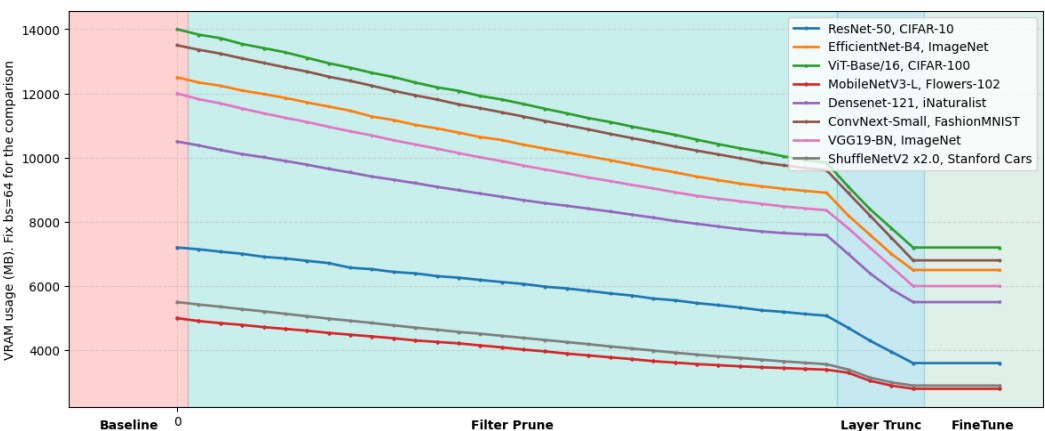

Figure 3: Evolution of peak VRAM usage during IDAP++ compression for vision models.

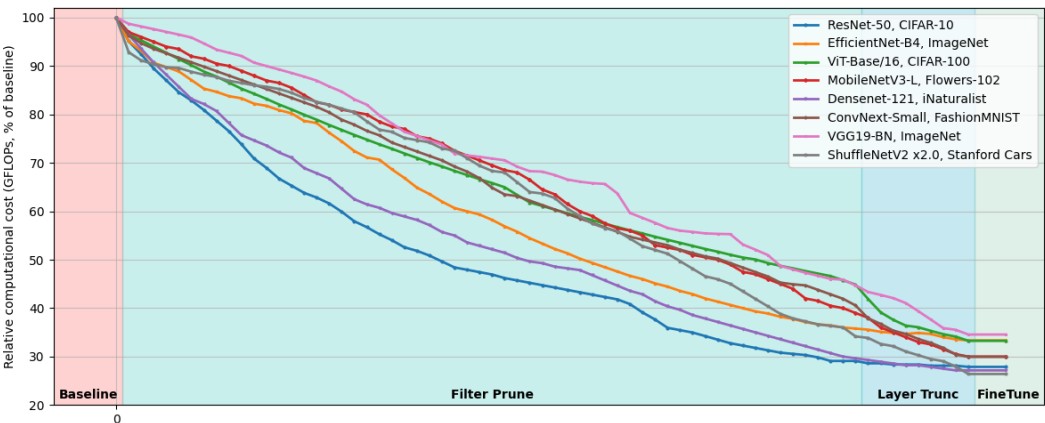

Figure 4: Evolution of relative computational cost during IDAP++ compression for vision models.

Also, we provide a dynamic view of how memory footprint and computational cost evolve throughout the two stages of IDAP++ in Figures 3 and 4. During the filter-pruning phase, both VRAM usage and GFLOPs decrease almost monotonically for all architectures, typically yielding 30–50% savings before any layers are removed, and doing so in a smooth, nearly linear fashion that highlights the stability of divergence-guided pruning. Once the algorithm enters the layer-truncation phase, an additional sharp reduction is observed: most models gain a further 20–30% drop in compute and memory, reaching overall savings of about 2–3× in peak VRAM and up to 70–80% in GFLOPs by the end of fine-tuning. In conjunction with Tables 28, 29, these trends confirm that the components removed by IDAP++ are largely redundant from the standpoint of information flow, enabling substantial resource reductions while maintaining competitive task quality.

## M  ABLATION STUDY OF THE IDAP++ COMPRESSION PIPELINE

Table 30: Pipeline Order in IDAP++ Ablation Study: Optimality of the Proposed Sequence, Part 1
*(FP – Filter Pruning, LT – Layer Truncation, FT – Fine-Tuning)*

| Model, Dataset, Quality Metric, Compression | Pipeline Description | FP Time (min) | LT Time (min) | FT Time (min) | Total Compres. Time (h, min) | Latency (ms) | Quality Metric | Params (M) | GFLOPs |
|---|---|---|---|---|---|---|---|---|---|
| ResNet-50, ImageNet, Acc@1, 50% | IDAP++ (ours): $FP \rightarrow LT \rightarrow FT$ | 242 | 168 | 612 | 17h2m | 2.7 | 75.8 | 11.5 | 1.8 |
| | Reverse order: $LT \rightarrow FP \rightarrow FT$ | 172 | 238 | 638 | 17h28m | 3.1 | 72.4 | 12.1 | 1.9 |
| | No Fine-Tuning: $FP \rightarrow LT$ | 251 | 174 | 0 | 7h5m | 2.8 | 70.9 | 11.5 | 1.8 |
| | *Only FP* (Stage 1 only) | 261 | 0 | 0 | 4h21m | 3.8 | 71.6 | 14.8 | 2.4 |
| | *Only LT* (Stage 2 only) | 0 | 152 | 0 | 2h32m | 2.5 | 69.7 | 10.8 | 1.6 |
| ResNet-50, ImageNet, Acc@1, 70% | IDAP++ (ours): $FP \rightarrow LT \rightarrow FT$ | 321 | 226 | 871 | 23h38m | 2.6 | 75.4 | 6.1 | 1.0 |
| | Reverse order: $LT \rightarrow FP \rightarrow FT$ | 224 | 318 | 905 | 24h7m | 3.0 | 71.1 | 6.8 | 1.1 |
| | No Fine-Tuning: $FP \rightarrow LT$ | 325 | 228 | 0 | 9h13m | 2.7 | 68.7 | 6.1 | 1.0 |
| | *Only FP* (Stage 1 only) | 341 | 0 | 0 | 5h41m | 3.5 | 69.4 | 8.9 | 1.5 |
| | *Only LT* (Stage 2 only) | 0 | 218 | 0 | 3h38m | 2.4 | 66.2 | 5.4 | 0.8 |
| ResNet-50, ImageNet, Acc@1, 90% | IDAP++ (ours): $FP \rightarrow LT \rightarrow FT$ | 438 | 298 | 1185 | 32h1m | 2.4 | 69.3 | 2.6 | 0.4 |
| | Reverse order: $LT \rightarrow FP \rightarrow FT$ | 312 | 428 | 1228 | 32h48m | 2.8 | 63.7 | 3.1 | 0.5 |
| | No Fine-Tuning: $FP \rightarrow LT$ | 398 | 289 | 0 | 11h27m | 2.5 | 60.1 | 2.6 | 0.4 |
| | *Only FP* (Stage 1 only) | 426 | 0 | 0 | 7h6m | 3.2 | 61.8 | 4.4 | 0.7 |
| | *Only LT* (Stage 2 only) | 0 | 312 | 0 | 5h12m | 2.2 | 58.9 | 2.1 | 0.3 |
| EfficientNet-B4, CIFAR-100, Acc@1, 50% | IDAP++ (ours): $FP \rightarrow LT \rightarrow FT$ | 72 | 52 | 301 | 7h5m | 3.1 | 89.4 | 9.6 | 2.1 |
| | Reverse order: $LT \rightarrow FP \rightarrow FT$ | 52 | 45 | 319 | 6h56m | 3.5 | 86.2 | 10.3 | 2.2 |
| | No Fine-Tuning: $FP \rightarrow LT$ | 68 | 48 | 0 | 1h56m | 3.2 | 84.7 | 9.6 | 2.1 |
| | *Only FP* (Stage 1 only) | 71 | 0 | 0 | 1h11m | 4.1 | 85.5 | 12.8 | 3.1 |
| | *Only LT* (Stage 2 only) | 0 | 49 | 0 | 0h49m | 2.8 | 82.9 | 8.7 | 1.7 |
| EfficientNet-B4, CIFAR-100, Acc@1, 70% | IDAP++ (ours): $FP \rightarrow LT \rightarrow FT$ | 102 | 72 | 412 | 9h46m | 2.6 | 88.8 | 7.1 | 1.7 |
| | Reverse order: $LT \rightarrow FP \rightarrow FT$ | 74 | 65 | 458 | 9h57m | 2.9 | 85.2 | 7.6 | 1.8 |
| | No Fine-Tuning: $FP \rightarrow LT$ | 92 | 66 | 0 | 2h38m | 2.7 | 83.1 | 7.1 | 1.7 |
| | *Only FP* (Stage 1 only) | 95 | 0 | 0 | 1h35m | 3.4 | 84.0 | 9.2 | 2.1 |
| | *Only LT* (Stage 2 only) | 0 | 71 | 0 | 1h11m | 2.4 | 81.7 | 6.3 | 1.5 |

Table 31: Pipeline Order in IDAP++ Ablation Study: Optimality of the Proposed Sequence, Part 2
*(FP – Filter Pruning, LT – Layer Truncation, FT – Fine-Tuning)*

| Model, Dataset, Quality Metric, Compression | Pipeline Description | FP Time (min) | LT Time (min) | FT Time (min) | Total Compres. Time (h, min) | Latency (ms) | Quality Metric | Params (M) | GFLOPs |
|---|---|---|---|---|---|---|---|---|---|
| EfficientNet-B4, CIFAR-100, Acc@1, 90% | IDAP++ (ours): $FP \to LT \to FT$ | 145 | 112 | 625 | 14h42m | 2.2 | 82.1 | 3.0 | 0.7 |
| | Reverse order: $LT \to FP \to FT$ | 115 | 95 | 689 | 14h59m | 2.6 | 78.4 | 3.4 | 0.8 |
| | No Fine-Tuning: $FP \to LT$ | 135 | 104 | 0 | 3h59m | 2.3 | 75.9 | 3.0 | 0.7 |
| | *Only FP* (Stage 1 only) | 148 | 0 | 0 | 2h28m | 3.1 | 76.8 | 4.7 | 1.1 |
| | *Only LT* (Stage 2 only) | 0 | 111 | 0 | 1h51m | 1.9 | 72.2 | 2.5 | 0.6 |
| ViT-Base/16, CIFAR-10, Acc@1, 50% | IDAP++ (ours): $FP \to LT \to FT$ | 182 | 145 | 451 | 12h58m | 7.2 | 98.0 | 55.4 | 11.8 |
| | Reverse order: $LT \to FP \to FT$ | 150 | 133 | 492 | 12h55m | 8.1 | 96.3 | 59.8 | 12.6 |
| | No Fine-Tuning: $FP \to LT$ | 172 | 135 | 0 | 5h7m | 7.4 | 94.9 | 55.4 | 11.8 |
| | *Only FP* (Stage 1 only) | 181 | 0 | 0 | 3h1m | 9.2 | 93.7 | 71.7 | 14.8 |
| | *Only LT* (Stage 2 only) | 0 | 141 | 0 | 2h21m | 6.5 | 91.3 | 47.8 | 9.6 |
| ViT-Base/16, CIFAR-10, Acc@1, 70% | IDAP++ (ours): $FP \to LT \to FT$ | 245 | 198 | 638 | 18h1m | 6.3 | 97.5 | 38.6 | 7.9 |
| | Reverse order: $LT \to FP \to FT$ | 205 | 174 | 697 | 17h56m | 7.1 | 94.8 | 41.2 | 8.4 |
| | No Fine-Tuning: $FP \to LT$ | 235 | 188 | 0 | 7h3m | 6.5 | 92.3 | 38.6 | 7.9 |
| | *Only FP* (Stage 1 only) | 248 | 0 | 0 | 4h8m | 7.8 | 93.1 | 52.3 | 10.8 |
| | *Only LT* (Stage 2 only) | 0 | 195 | 0 | 3h15m | 5.9 | 90.4 | 31.2 | 6.1 |
| ViT-Base/16, CIFAR-10, Acc@1, 90% | IDAP++ (ours): $FP \to LT \to FT$ | 322 | 252 | 842 | 23h36m | 5.7 | 92.1 | 16.4 | 3.1 |
| | Reverse order: $LT \to FP \to FT$ | 260 | 225 | 918 | 23h23m | 6.3 | 87.2 | 18.6 | 3.5 |
| | No Fine-Tuning: $FP \to LT$ | 315 | 244 | 0 | 9h19m | 5.9 | 84.7 | 16.4 | 3.1 |
| | *Only FP* (Stage 1 only) | 332 | 0 | 0 | 5h32m | 7.4 | 85.3 | 23.1 | 4.4 |
| | *Only LT* (Stage 2 only) | 0 | 252 | 0 | 4h12m | 5.1 | 81.2 | 12.9 | 2.4 |
| Faster R-CNN (ResNet-50), Pascal VOC, mAP, 50% | IDAP++ (ours): $FP \to LT \to FT$ | 88 | 67 | 914 | 17h49m | 11.4 | 77.2 | 34.7 | 121.3 |
| | Reverse order: $LT \to FP \to FT$ | 72 | 75 | 932 | 17h59m | 12.8 | 73.6 | 39.1 | 133.6 |
| | No Fine-Tuning: $FP \to LT$ | 89 | 69 | 0 | 2h38m | 11.9 | 70.2 | 34.7 | 121.3 |
| | *Only FP* (Stage 1 only) | 99 | 0 | 0 | 1h39m | 14.5 | 71.1 | 51.3 | 161.7 |
| | *Only LT* (Stage 2 only) | 0 | 75 | 0 | 1h15m | 10.8 | 68.8 | 29.4 | 98.2 |

Table 32: Pipeline Order in IDAP++ Ablation Study: Optimality of the Proposed Sequence, Part 3
*(FP – Filter Pruning, LT – Layer Truncation, FT – Fine-Tuning)*

| Model, Dataset, Quality Metric, Compression | Pipeline Description | FP Time (min) | LT Time (min) | FT Time (min) | Total Compres. Time (h, min) | Latency (ms) | Quality Metric | Params (M) | GFLOPs |
|---|---|---|---|---|---|---|---|---|---|
| Faster R-CNN (ResNet-50), Pascal VOC, mAP, 70% | IDAP++ (ours): $FP \rightarrow LT \rightarrow FT$ | 108 | 83 | 1050 | 20h41m | 8.4 | 76.7 | 15.1 | 61.6 |
| | Reverse order: $LT \rightarrow FP \rightarrow FT$ | 98 | 85 | 1127 | 21h50m | 9.2 | 72.4 | 16.8 | 68.3 |
| | No Fine-Tuning: $FP \rightarrow LT$ | 109 | 84 | 0 | 3h13m | 8.7 | 70.1 | 15.1 | 61.6 |
| | *Only FP* (Stage 1 only) | 110 | 0 | 0 | 1h50m | 10.1 | 71.3 | 21.4 | 82.7 |
| | *Only LT* (Stage 2 only) | 0 | 85 | 0 | 1h25m | 7.9 | 68.9 | 12.8 | 54.2 |
| Faster R-CNN (ResNet-50), Pascal VOC, mAP, 90% | IDAP++ (ours): $FP \rightarrow LT \rightarrow FT$ | 132 | 102 | 1314 | 25h48m | 10.6 | 63.4 | 7.2 | 28.1 |
| | Reverse order: $LT \rightarrow FP \rightarrow FT$ | 109 | 116 | 1381 | 26h46m | 12.1 | 59.1 | 8.4 | 32.7 |
| | No Fine-Tuning: $FP \rightarrow LT$ | 133 | 103 | 0 | 3h56m | 10.9 | 55.3 | 7.2 | 28.1 |
| | *Only FP* (Stage 1 only) | 137 | 0 | 0 | 2h17m | 13.4 | 57.8 | 11.3 | 41.5 |
| | *Only LT* (Stage 2 only) | 0 | 104 | 0 | 1h44m | 9.8 | 54.1 | 5.9 | 21.9 |
| YOLOv4 (ShuffleNetV2), Pascal VOC, mAP, 50% | IDAP++ (ours): $FP \rightarrow LT \rightarrow FT$ | 44 | 31 | 520 | 9h55m | 8.4 | 76.3 | 12.7 | 29.1 |
| | Reverse order: $LT \rightarrow FP \rightarrow FT$ | 33 | 37 | 545 | 10h15m | 9.1 | 72.8 | 14.3 | 33.7 |
| | No Fine-Tuning: $FP \rightarrow LT$ | 44 | 31 | 0 | 1h15m | 8.6 | 69.4 | 12.7 | 29.1 |
| | *Only FP* (Stage 1 only) | 45 | 0 | 0 | 0h45m | 10.4 | 70.8 | 19.8 | 44.2 |
| | *Only LT* (Stage 2 only) | 0 | 31 | 0 | 0h31m | 7.8 | 67.6 | 10.1 | 24.5 |
| YOLOv4 (ShuffleNetV2), Pascal VOC, mAP, 70% | IDAP++ (ours): $FP \rightarrow LT \rightarrow FT$ | 54 | 39 | 642 | 12h15m | 6.5 | 75.8 | 9.1 | 22.1 |
| | Reverse order: $LT \rightarrow FP \rightarrow FT$ | 42 | 46 | 673 | 12h41m | 7.3 | 71.9 | 10.2 | 24.8 |
| | No Fine-Tuning: $FP \rightarrow LT$ | 55 | 41 | 0 | 1h36m | 6.8 | 69.4 | 9.1 | 22.1 |
| | *Only FP* (Stage 1 only) | 56 | 0 | 0 | 0h56m | 8.1 | 70.2 | 13.8 | 31.6 |
| | *Only LT* (Stage 2 only) | 0 | 44 | 0 | 0h44m | 6.1 | 67.8 | 7.9 | 19.3 |
| YOLOv4 (ShuffleNetV2), Pascal VOC, mAP, 90% | IDAP++ (ours): $FP \rightarrow LT \rightarrow FT$ | 68 | 49 | 823 | 15h40m | 6.1 | 62.7 | 4.2 | 8.7 |
| | Reverse order: $LT \rightarrow FP \rightarrow FT$ | 52 | 58 | 870 | 16h20m | 6.9 | 58.2 | 4.8 | 9.9 |
| | No Fine-Tuning: $FP \rightarrow LT$ | 69 | 48 | 0 | 1h57m | 6.3 | 55.1 | 4.2 | 8.7 |
| | *Only FP* (Stage 1 only) | 71 | 0 | 0 | 1h11m | 7.8 | 56.4 | 7.1 | 14.7 |
| | *Only LT* (Stage 2 only) | 0 | 53 | 0 | 0h53m | 5.4 | 51.2 | 3.3 | 6.9 |

Table 33: Pipeline Order in IDAP++ Ablation Study: Optimality of the Proposed Sequence, Part 4
*(FP – Filter Pruning, LT – Layer Truncation, FT – Fine-Tuning)*

| Model, Dataset, Quality Metric, Compression | Pipeline Description | FP Time (min) | LT Time (min) | FT Time (min) | Total Compres. Time (h, min) | Latency (ms) | Quality Metric | Params (M) | GFLOPs |
|---|---|---|---|---|---|---|---|---|---|
| DETR (ViT-Base/16), COCO 2017, mAP, 50% | IDAP++ (ours): $FP \to LT \to FT$ | 148 | 112 | 1253 | 25h13m | 20.1 | 41.1 | 54.9 | 70.2 |
| | Reverse order: $LT \to FP \to FT$ | 122 | 130 | 1322 | 26h14m | 21.8 | 37.4 | 58.7 | 76.4 |
| | No Fine-Tuning: $FP \to LT$ | 149 | 115 | 0 | 4h24m | 20.6 | 35.1 | 54.9 | 70.2 |
| | *Only FP* (Stage 1 only) | 159 | 0 | 0 | 2h39m | 24.1 | 36.7 | 75.4 | 98.5 |
| | *Only LT* (Stage 2 only) | 0 | 121 | 0 | 2h1m | 18.9 | 33.9 | 47.1 | 59.3 |
| DETR (ViT-Base/16), COCO 2017, mAP, 70% | IDAP++ (ours): $FP \to LT \to FT$ | 182 | 142 | 1532 | 30h56m | 13.5 | 40.5 | 32.8 | 36.9 |
| | Reverse order: $LT \to FP \to FT$ | 148 | 160 | 1597 | 31h45m | 15.1 | 36.2 | 35.9 | 40.1 |
| | No Fine-Tuning: $FP \to LT$ | 183 | 142 | 0 | 5h25m | 14.0 | 33.8 | 32.8 | 36.9 |
| | *Only FP* (Stage 1 only) | 185 | 0 | 0 | 3h5m | 16.8 | 34.7 | 44.1 | 51.2 |
| | *Only LT* (Stage 2 only) | 0 | 144 | 0 | 2h24m | 12.7 | 31.6 | 28.4 | 31.5 |
| DETR (ViT-Base/16), COCO 2017, mAP, 90% | IDAP++ (ours): $FP \to LT \to FT$ | 222 | 176 | 1754 | 35h52m | 17.8 | 27.5 | 14.3 | 15.9 |
| | Reverse order: $LT \to FP \to FT$ | 184 | 195 | 1872 | 37h31m | 19.1 | 24.1 | 15.8 | 18.4 |
| | No Fine-Tuning: $FP \to LT$ | 223 | 178 | 0 | 6h41m | 18.1 | 22.7 | 14.3 | 15.9 |
| | *Only FP* (Stage 1 only) | 226 | 0 | 0 | 3h46m | 21.0 | 23.5 | 20.3 | 25.7 |
| | *Only LT* (Stage 2 only) | 0 | 185 | 0 | 3h5m | 16.4 | 19.8 | 11.4 | 12.7 |
| FCN (VGG19-BN), Cityscapes, mIoU, 50% | IDAP++ (ours): $FP \to LT \to FT$ | 92 | 68 | 650 | 13h30m | 24.3 | 69.1 | 121.4 | 176.3 |
| | Reverse order: $LT \to FP \to FT$ | 72 | 85 | 671 | 13h48m | 26.1 | 66.2 | 132.0 | 191.4 |
| | No Fine-Tuning: $FP \to LT$ | 93 | 78 | 0 | 2h51m | 24.8 | 63.0 | 121.4 | 176.3 |
| | *Only FP* (Stage 1 only) | 95 | 0 | 0 | 1h35m | 28.5 | 64.7 | 167.8 | 246.7 |
| | *Only LT* (Stage 2 only) | 0 | 71 | 0 | 1h11m | 22.9 | 61.8 | 97.0 | 139.1 |
| FCN (VGG19-BN), Cityscapes, mIoU, 70% | IDAP++ (ours): $FP \to LT \to FT$ | 110 | 82 | 850 | 17h22m | 18.2 | 68.9 | 47.1 | 82.9 |
| | Reverse order: $LT \to FP \to FT$ | 90 | 100 | 888 | 17h58m | 19.7 | 65.4 | 52.9 | 94.0 |
| | No Fine-Tuning: $FP \to LT$ | 119 | 81 | 0 | 3h20m | 18.8 | 62.1 | 47.1 | 82.9 |
| | *Only FP* (Stage 1 only) | 124 | 0 | 0 | 2h4m | 22.1 | 63.5 | 71.3 | 123.5 |
| | *Only LT* (Stage 2 only) | 0 | 86 | 0 | 1h26m | 17.1 | 60.8 | 37.9 | 60.4 |

Table 34: Pipeline Order in IDAP++ Ablation Study: Optimality of the Proposed Sequence, Part 5
*(FP – Filter Pruning, LT – Layer Truncation, FT – Fine-Tuning)*

| Model, Dataset, Quality Metric, Compression | Pipeline Description | FP Time (min) | LT Time (min) | FT Time (min) | Total Compres. Time (h, min) | Latency (ms) | Quality Metric | Params (M) | GFLOPs |
|---|---|---|---|---|---|---|---|---|---|
| FCN (VGG19-BN), Cityscapes, mIoU, 90% | IDAP++ (ours): $FP \rightarrow LT \rightarrow FT$ | 138 | 98 | 1103 | 22h19m | 16.8 | 61.2 | 28.3 | 41.8 |
| | Reverse order: $LT \rightarrow FP \rightarrow FT$ | 105 | 128 | 1154 | 23h7m | 18.4 | 57.9 | 31.5 | 46.9 |
| | No Fine-Tuning: $FP \rightarrow LT$ | 148 | 96 | 0 | 4h4m | 17.2 | 55.3 | 28.3 | 41.8 |
| | *Only FP* (Stage 1 only) | 147 | 0 | 0 | 2h27m | 20.6 | 54.6 | 42.0 | 62.4 |
| | *Only LT* (Stage 2 only) | 0 | 108 | 0 | 1h48m | 15.4 | 51.3 | 21.1 | 30.7 |
| U-Net (ResNet-50), Pascal VOC, mIoU, 50% | IDAP++ (ours): $FP \rightarrow LT \rightarrow FT$ | 57 | 40 | 420 | 8h37m | 13.4 | 76.1 | 82.4 | 121.6 |
| | Reverse order: $LT \rightarrow FP \rightarrow FT$ | 44 | 50 | 435 | 8h49m | 14.9 | 72.8 | 89.3 | 132.0 |
| | No Fine-Tuning: $FP \rightarrow LT$ | 58 | 39 | 0 | 1h37m | 13.8 | 70.2 | 82.4 | 121.6 |
| | *Only FP* (Stage 1 only) | 63 | 0 | 0 | 1h3m | 16.7 | 71.6 | 118.2 | 175.2 |
| | *Only LT* (Stage 2 only) | 0 | 42 | 0 | 0h42m | 12.6 | 68.4 | 65.4 | 94.7 |
| U-Net (ResNet-50), Pascal VOC, mIoU, 70% | IDAP++ (ours): $FP \rightarrow LT \rightarrow FT$ | 74 | 50 | 580 | 11h44m | 10.7 | 74.2 | 11.2 | 62.1 |
| | Reverse order: $LT \rightarrow FP \rightarrow FT$ | 55 | 64 | 609 | 12h8m | 12.0 | 70.5 | 12.6 | 68.4 |
| | No Fine-Tuning: $FP \rightarrow LT$ | 76 | 48 | 0 | 2h4m | 11.3 | 67.4 | 11.2 | 62.1 |
| | *Only FP* (Stage 1 only) | 77 | 0 | 0 | 1h17m | 13.9 | 68.1 | 16.9 | 82.3 |
| | *Only LT* (Stage 2 only) | 0 | 52 | 0 | 0h52m | 10.1 | 63.0 | 8.1 | 48.1 |
| U-Net (ResNet-50), Pascal VOC, mIoU, 90% | IDAP++ (ours): $FP \rightarrow LT \rightarrow FT$ | 95 | 62 | 809 | 16h6m | 9.3 | 61.7 | 5.4 | 31.1 |
| | Reverse order: $LT \rightarrow FP \rightarrow FT$ | 68 | 82 | 830 | 16h20m | 10.5 | 58.4 | 6.5 | 34.9 |
| | No Fine-Tuning: $FP \rightarrow LT$ | 97 | 62 | 0 | 2h39m | 9.6 | 55.1 | 5.4 | 31.1 |
| | *Only FP* (Stage 1 only) | 102 | 0 | 0 | 1h42m | 11.6 | 54.2 | 9.3 | 43.1 |
| | *Only LT* (Stage 2 only) | 0 | 68 | 0 | 1h8m | 8.7 | 50.8 | 4.0 | 23.9 |
| SegFormer (ViT-Base/16), COCO 2017, mIoU, 50% | IDAP++ (ours): $FP \rightarrow LT \rightarrow FT$ | 135 | 98 | 1097 | 22h10m | 21.5 | 46.0 | 102.4 | 133.8 |
| | Reverse order: $LT \rightarrow FP \rightarrow FT$ | 108 | 112 | 1143 | 22h43m | 23.1 | 42.6 | 111.7 | 147.9 |
| | No Fine-Tuning: $FP \rightarrow LT$ | 132 | 92 | 0 | 3h44m | 21.9 | 40.2 | 102.4 | 133.8 |
| | *Only FP* (Stage 1 only) | 134 | 0 | 0 | 2h14m | 25.4 | 41.7 | 144.9 | 190.5 |
| | *Only LT* (Stage 2 only) | 0 | 105 | 0 | 1h45m | 20.4 | 38.9 | 82.7 | 104.4 |

Table 35: Pipeline Order in IDAP++ Ablation Study: Optimality of the Proposed Sequence, Part 6
*(FP – Filter Pruning, LT – Layer Truncation, FT – Fine-Tuning)*

| Model, Dataset, Quality Metric, Compression | Pipeline Description | FP Time (min) | LT Time (min) | FT Time (min) | Total Compres. Time (h, min) | Latency (ms) | Quality Metric | Params (M) | GFLOPs |
|---|---|---|---|---|---|---|---|---|---|
| SegFormer (ViT-Base/16), COCO 2017, mIoU, 70% | IDAP++ (ours): $FP \to LT \to FT$ | 157 | 122 | 1342 | 27h1m | 17.3 | 45.1 | 32.5 | 62.9 |
| | Reverse order: $LT \to FP \to FT$ | 132 | 145 | 1417 | 28h14m | 19.1 | 41.8 | 35.4 | 68.1 |
| | No Fine-Tuning: $FP \to LT$ | 162 | 128 | 0 | 4h50m | 17.9 | 39.1 | 32.5 | 62.9 |
| | *Only FP* (Stage 1 only) | 163 | 0 | 0 | 2h43m | 21.7 | 38.7 | 47.3 | 89.4 |
| | *Only LT* (Stage 2 only) | 0 | 124 | 0 | 2h4m | 16.6 | 35.7 | 26.7 | 51.9 |
| SegFormer (ViT-Base/16), COCO 2017, mIoU, 90% | IDAP++ (ours): $FP \to LT \to FT$ | 188 | 141 | 1632 | 32h41m | 14.8 | 33.4 | 13.2 | 27.5 |
| | Reverse order: $LT \to FP \to FT$ | 155 | 174 | 1682 | 33h31m | 16.1 | 30.1 | 14.6 | 31.2 |
| | No Fine-Tuning: $FP \to LT$ | 199 | 138 | 0 | 5h37m | 15.0 | 28.5 | 13.2 | 27.5 |
| | *Only FP* (Stage 1 only) | 195 | 0 | 0 | 3h15m | 17.9 | 27.2 | 19.1 | 41.8 |
| | *Only LT* (Stage 2 only) | 0 | 143 | 0 | 2h23m | 13.9 | 25.6 | 10.1 | 21.9 |
| DCGAN, CIFAR-10, FID, 50% | IDAP++ (ours): $FP \to LT \to FT$ | 8 | 5 | 60 | 1h13m | 3.8 | 24.9 | 8.2 | 8.3 |
| | Reverse order: $LT \to FP \to FT$ | 6 | 7 | 65 | 1h18m | 4.2 | 26.8 | 9.1 | 9.4 |
| | No Fine-Tuning: $FP \to LT$ | 8 | 5 | 0 | 0h13m | 3.9 | 28.7 | 8.2 | 8.3 |
| | *Only FP* (Stage 1 only) | 9 | 0 | 0 | 0h9m | 4.9 | 29.4 | 12.1 | 12.5 |
| | *Only LT* (Stage 2 only) | 0 | 4 | 0 | 0h4m | 3.4 | 31.2 | 6.4 | 6.1 |
| DCGAN, CIFAR-10, FID, 70% | IDAP++ (ours): $FP \to LT \to FT$ | 10 | 7 | 99 | 1h56m | 3.1 | 25.9 | 4.1 | 4.8 |
| | Reverse order: $LT \to FP \to FT$ | 8 | 9 | 95 | 1h52m | 3.5 | 27.8 | 4.5 | 5.2 |
| | No Fine-Tuning: $FP \to LT$ | 10 | 7 | 0 | 0h17m | 3.3 | 29.9 | 4.1 | 4.8 |
| | *Only FP* (Stage 1 only) | 11 | 0 | 0 | 0h11m | 4.0 | 30.8 | 6.2 | 7.1 |
| | *Only LT* (Stage 2 only) | 0 | 6 | 0 | 0h6m | 2.8 | 33.1 | 3.1 | 3.6 |
| DCGAN, CIFAR-10, FID, 90% | IDAP++ (ours): $FP \to LT \to FT$ | 14 | 9 | 155 | 2h58m | 2.4 | 34.7 | 1.8 | 1.9 |
| | Reverse order: $LT \to FP \to FT$ | 10 | 13 | 158 | 3h1m | 2.7 | 38.1 | 2.1 | 2.3 |
| | No Fine-Tuning: $FP \to LT$ | 14 | 9 | 0 | 0h23m | 2.5 | 41.0 | 1.8 | 1.9 |
| | *Only FP* (Stage 1 only) | 15 | 0 | 0 | 0h15m | 3.1 | 39.7 | 3.0 | 3.4 |
| | *Only LT* (Stage 2 only) | 0 | 7 | 0 | 0h7m | 2.1 | 45.2 | 1.4 | 1.6 |

Table 36: Pipeline Order in IDAP++ Ablation Study: Optimality of the Proposed Sequence, Part 7
*(FP – Filter Pruning, LT – Layer Truncation, FT – Fine-Tuning)*

| Model, Dataset, Quality Metric, Compression | Pipeline Description | FP Time (min) | LT Time (min) | FT Time (min) | Total Compres. Time (h, min) | Latency (ms) | Quality Metric | Params (M) | GFLOPs |
|---|---|---|---|---|---|---|---|---|---|
| VQGAN, COCO-Stuff, FID, 50% | IDAP++ (ours): $FP \rightarrow LT \rightarrow FT$ | 20 | 14 | 150 | 3h4m | 12.8 | 19.4 | 14.1 | 15.4 |
| | Reverse order: $LT \rightarrow FP \rightarrow FT$ | 16 | 18 | 158 | 3h12m | 14.3 | 21.1 | 15.6 | 17.2 |
| | No Fine-Tuning: $FP \rightarrow LT$ | 20 | 14 | 0 | 0h34m | 13.1 | 22.8 | 14.1 | 15.4 |
| | *Only FP* (Stage 1 only) | 22 | 0 | 0 | 0h22m | 16.1 | 23.4 | 20.1 | 22.4 |
| | *Only LT* (Stage 2 only) | 0 | 12 | 0 | 0h12m | 11.9 | 25.8 | 11.2 | 12.9 |
| VQGAN, COCO-Stuff, FID, 70% | IDAP++ (ours): $FP \rightarrow LT \rightarrow FT$ | 28 | 20 | 234 | 4h42m | 3.1 | 20.1 | 6.1 | 7.5 |
| | Reverse order: $LT \rightarrow FP \rightarrow FT$ | 22 | 26 | 241 | 4h49m | 3.6 | 22.9 | 6.8 | 8.3 |
| | No Fine-Tuning: $FP \rightarrow LT$ | 28 | 20 | 0 | 0h48m | 3.3 | 24.7 | 6.1 | 7.5 |
| | *Only FP* (Stage 1 only) | 31 | 0 | 0 | 0h31m | 4.2 | 25.3 | 9.1 | 10.7 |
| | *Only LT* (Stage 2 only) | 0 | 16 | 0 | 0h16m | 2.9 | 27.8 | 4.5 | 5.8 |
| VQGAN, COCO-Stuff, FID, 90% | IDAP++ (ours): $FP \rightarrow LT \rightarrow FT$ | 38 | 27 | 320 | 6h25m | 9.1 | 32.6 | 2.2 | 2.7 |
| | Reverse order: $LT \rightarrow FP \rightarrow FT$ | 30 | 35 | 335 | 6h40m | 10.4 | 35.1 | 2.6 | 3.1 |
| | No Fine-Tuning: $FP \rightarrow LT$ | 38 | 27 | 0 | 1h5m | 9.4 | 37.4 | 2.2 | 2.7 |
| | *Only FP* (Stage 1 only) | 42 | 0 | 0 | 0h42m | 12.0 | 36.8 | 4.1 | 4.5 |
| | *Only LT* (Stage 2 only) | 0 | 21 | 0 | 0h21m | 8.3 | 39.1 | 1.6 | 2.0 |
| Stable Diffusion v1.5, MS COCO, FID, 50% | IDAP++ (ours): $FP \rightarrow LT \rightarrow FT$ | 140 | 95 | 1501 | 28h56m | 96.2 | 13.1 | 612.3 | 57.9 |
| | Reverse order: $LT \rightarrow FP \rightarrow FT$ | 110 | 125 | 1589 | 30h24m | 105.8 | 14.7 | 654.9 | 62.8 |
| | No Fine-Tuning: $FP \rightarrow LT$ | 145 | 98 | 0 | 4h3m | 98.7 | 16.9 | 612.3 | 57.9 |
| | *Only FP* (Stage 1 only) | 154 | 0 | 0 | 2h34m | 115.5 | 17.3 | 822.6 | 78.1 |
| | *Only LT* (Stage 2 only) | 0 | 87 | 0 | 1h27m | 90.1 | 19.4 | 488.4 | 43.6 |
| Stable Diffusion v1.5, MS COCO, FID, 70% | IDAP++ (ours): $FP \rightarrow LT \rightarrow FT$ | 171 | 115 | 2016 | 38h22m | 76.4 | 13.5 | 321.8 | 34.3 |
| | Reverse order: $LT \rightarrow FP \rightarrow FT$ | 135 | 150 | 2110 | 39h55m | 84.2 | 16.8 | 351.4 | 38.9 |
| | No Fine-Tuning: $FP \rightarrow LT$ | 170 | 115 | 0 | 4h45m | 79.1 | 18.4 | 321.8 | 34.3 |
| | *Only FP* (Stage 1 only) | 189 | 0 | 0 | 3h9m | 91.3 | 19.7 | 421.6 | 46.8 |
| | *Only LT* (Stage 2 only) | 0 | 95 | 0 | 1h35m | 71.8 | 22.9 | 281.3 | 30.1 |

Table 37: Pipeline Order in IDAP++ Ablation Study: Optimality of the Proposed Sequence, Part 8
*(FP – Filter Pruning, LT – Layer Truncation, FT – Fine-Tuning)*

| Model, Dataset, Quality Metric, Compression | Pipeline Description | FP Time (min) | LT Time (min) | FT Time (min) | Total Compres. Time (h, min) | Latency (ms) | Quality Metric | Params (M) | GFLOPs |
|---|---|---|---|---|---|---|---|---|---|
| Stable Diffusion v1.5, MS COCO, FID, 90% | IDAP++ (ours): $FP \to LT \to FT$ | 210 | 140 | 2658 | 50h8m | 58.1 | 25.7 | 72.3 | 10.8 |
| | Reverse order: $LT \to FP \to FT$ | 165 | 185 | 2789 | 52h19m | 64.9 | 29.1 | 81.0 | 12.1 |
| | No Fine-Tuning: $FP \to LT$ | 210 | 145 | 0 | 5h55m | 60.4 | 32.8 | 72.3 | 10.8 |
| | *Only FP* (Stage 1 only) | 226 | 0 | 0 | 3h46m | 71.2 | 31.1 | 113.2 | 16.0 |
| | *Only LT* (Stage 2 only) | 0 | 117 | 0 | 1h57m | 53.7 | 37.6 | 54.1 | 8.1 |
| BERT Base, MNLI-m, Acc@1, 50% | IDAP++ (ours): $FP \to LT \to FT$ | 20 | 14 | 240 | 4h34m | 8.1 | 83.1 | 52.8 | 19.4 |
| | Reverse order: $LT \to FP \to FT$ | 16 | 20 | 255 | 4h51m | 9.2 | 80.2 | 56.9 | 21.0 |
| | No Fine-Tuning: $FP \to LT$ | 21 | 14 | 0 | 0h35m | 8.4 | 78.0 | 52.8 | 19.4 |
| | *Only FP* (Stage 1 only) | 22 | 0 | 0 | 0h22m | 10.7 | 79.3 | 72.1 | 26.4 |
| | *Only LT* (Stage 2 only) | 0 | 15 | 0 | 0h15m | 7.5 | 76.4 | 40.4 | 14.2 |
| BERT Base, MNLI-m, Acc@1, 70% | IDAP++ (ours): $FP \to LT \to FT$ | 28 | 20 | 341 | 6h29m | 5.5 | 82.1 | 32.4 | 11.2 |
| | Reverse order: $LT \to FP \to FT$ | 22 | 26 | 362 | 6h50m | 6.2 | 78.9 | 35.1 | 12.4 |
| | No Fine-Tuning: $FP \to LT$ | 29 | 20 | 0 | 0h49m | 5.8 | 76.4 | 32.4 | 11.2 |
| | *Only FP* (Stage 1 only) | 33 | 0 | 0 | 0h33m | 7.1 | 77.2 | 44.8 | 15.9 |
| | *Only LT* (Stage 2 only) | 0 | 18 | 0 | 0h18m | 5.2 | 74.1 | 27.9 | 9.6 |
| BERT Base, MNLI-m, Acc@1, 90% | IDAP++ (ours): $FP \to LT \to FT$ | 44 | 33 | 520 | 9h57m | 4.8 | 72.9 | 10.1 | 3.7 |
| | Reverse order: $LT \to FP \to FT$ | 32 | 38 | 557 | 10h27m | 5.4 | 69.4 | 11.3 | 4.2 |
| | No Fine-Tuning: $FP \to LT$ | 41 | 30 | 0 | 1h11m | 5.0 | 67.1 | 10.1 | 3.7 |
| | *Only FP* (Stage 1 only) | 44 | 0 | 0 | 0h44m | 6.3 | 65.9 | 15.6 | 5.6 |
| | *Only LT* (Stage 2 only) | 0 | 28 | 0 | 0h28m | 4.3 | 63.2 | 7.5 | 2.9 |
| GPT-2 Base, SQuAD 1.1, F1, 50% | IDAP++ (ours): $FP \to LT \to FT$ | 24 | 17 | 278 | 5h19m | 9.2 | 86.8 | 48.2 | 12.8 |
| | Reverse order: $LT \to FP \to FT$ | 18 | 26 | 287 | 5h31m | 10.4 | 83.4 | 52.1 | 14.0 |
| | No Fine-Tuning: $FP \to LT$ | 25 | 16 | 0 | 0h41m | 9.6 | 81.1 | 48.2 | 12.8 |
| | *Only FP* (Stage 1 only) | 26 | 0 | 0 | 0h26m | 11.9 | 82.6 | 67.2 | 18.9 |
| | *Only LT* (Stage 2 only) | 0 | 19 | 0 | 0h19m | 8.7 | 79.3 | 36.7 | 9.3 |

Table 38: Pipeline Order in IDAP++ Ablation Study: Optimality of the Proposed Sequence, Part 9
*(FP – Filter Pruning, LT – Layer Truncation, FT – Fine-Tuning)*

| Model, Dataset, Quality Metric, Compression | Pipeline Description | FP Time (min) | LT Time (min) | FT Time (min) | Total Compres. Time (h, min) | Latency (ms) | Quality Metric | Params (M) | GFLOPs |
|---|---|---|---|---|---|---|---|---|---|
| GPT-2 Base, SQuAD 1.1, F1, 70% | IDAP++ (ours): *FP → LT → FT* | 32 | 20 | 389 | 7h21m | 7.9 | 86.1 | 55.0 | 14.5 |
| | Reverse order: *LT → FP → FT* | 25 | 30 | 412 | 7h47m | 8.8 | 82.7 | 59.3 | 16.1 |
| | No Fine-Tuning: *FP → LT* | 33 | 22 | 0 | 0h55m | 8.2 | 80.3 | 55.0 | 14.5 |
| | *Only FP* (Stage 1 only) | 36 | 0 | 0 | 0h36m | 10.1 | 81.4 | 71.2 | 19.8 |
| | *Only LT* (Stage 2 only) | 0 | 21 | 0 | 0h21m | 7.4 | 78.9 | 47.1 | 12.3 |
| GPT-2 Base, SQuAD 1.1, F1, 90% | IDAP++ (ours): *FP → LT → FT* | 46 | 30 | 561 | 10h37m | 6.8 | 70.3 | 9.3 | 2.7 |
| | Reverse order: *LT → FP → FT* | 34 | 45 | 586 | 11h5m | 7.5 | 67.1 | 10.4 | 3.1 |
| | No Fine-Tuning: *FP → LT* | 47 | 31 | 0 | 1h18m | 7.0 | 63.8 | 9.3 | 2.7 |
| | *Only FP* (Stage 1 only) | 55 | 0 | 0 | 0h55m | 8.6 | 62.4 | 14.8 | 4.4 |
| | *Only LT* (Stage 2 only) | 0 | 28 | 0 | 0h28m | 6.1 | 58.9 | 6.7 | 1.9 |
| T5 Base, MNLI-m, Acc@1, 50% | IDAP++ (ours): *FP → LT → FT* | 30 | 22 | 326 | 6h18m | 17.1 | 85.4 | 151.2 | 46.1 |
| | Reverse order: *LT → FP → FT* | 24 | 35 | 348 | 6h47m | 18.8 | 82.1 | 164.8 | 50.4 |
| | No Fine-Tuning: *FP → LT* | 31 | 22 | 0 | 0h53m | 17.4 | 79.8 | 151.2 | 46.1 |
| | *Only FP* (Stage 1 only) | 33 | 0 | 0 | 0h33m | 21.1 | 80.9 | 213.7 | 62.4 |
| | *Only LT* (Stage 2 only) | 0 | 20 | 0 | 0h20m | 16.2 | 78.5 | 121.0 | 36.3 |
| T5 Base, MNLI-m, Acc@1, 70% | IDAP++ (ours): *FP → LT → FT* | 40 | 30 | 468 | 8h58m | 13.9 | 84.0 | 97.8 | 30.9 |
| | Reverse order: *LT → FP → FT* | 32 | 41 | 486 | 9h19m | 15.6 | 80.1 | 105.4 | 34.2 |
| | No Fine-Tuning: *FP → LT* | 41 | 32 | 0 | 1h13m | 14.4 | 77.8 | 97.8 | 30.9 |
| | *Only FP* (Stage 1 only) | 44 | 0 | 0 | 0h44m | 17.8 | 78.9 | 131.6 | 42.7 |
| | *Only LT* (Stage 2 only) | 0 | 28 | 0 | 0h28m | 13.1 | 75.6 | 83.2 | 26.4 |
| T5 Base, MNLI-m, Acc@1, 90% | IDAP++ (ours): *FP → LT → FT* | 56 | 41 | 687 | 13h4m | 10.1 | 71.6 | 21.4 | 6.9 |
| | Reverse order: *LT → FP → FT* | 44 | 52 | 712 | 13h28m | 11.4 | 68.1 | 24.3 | 8.0 |
| | No Fine-Tuning: *FP → LT* | 57 | 40 | 0 | 1h37m | 10.5 | 64.8 | 21.4 | 6.9 |
| | *Only FP* (Stage 1 only) | 65 | 0 | 0 | 1h5m | 12.5 | 63.4 | 34.7 | 10.9 |
| | *Only LT* (Stage 2 only) | 0 | 36 | 0 | 0h36m | 9.2 | 59.7 | 16.0 | 5.1 |

Tables 30, 31, 32, 33, 34, 35, 36, 37, 38 present an extensive ablation of the IDAP++ pipeline design across vision, detection, segmentation, generative, and NLP models. For each architecture and for three compression regimes (50%, 70%, 90%), we compare five variants: (i) our full pipeline (Filter Pruning → Layer Truncation → Fine-Tuning), (ii) reversed order (Layer Truncation → Filter Pruning → Fine-Tuning), (iii) no fine-tuning, (iv) Stage 1 only (filter pruning only), and (v) Stage 2 only (layer truncation only). The results clearly show that both the order of stages and the presence of fine-tuning are crucial: the full IDAP++ pipeline consistently yields the best or near-best quality for a given compression level, while maintaining competitive compression time and delivering the strongest gains in latency, parameter count, and FLOPs.

First, the comparison between the standard and reversed orders highlights the importance of applying filter pruning before layer truncation. Across almost all models and compression ratios, reversing the order leads to a substantial drop in quality at similar or even slightly higher compression time. For example, on ResNet-50 / ImageNet at 70% compression, IDAP++ achieves 75.4% Acc@1 with 6.1M parameters and 1.0 GFLOPs in 23 h 38 min, whereas the reversed pipeline drops to 71.1% Acc@1 with 6.8M parameters and 1.1 GFLOPs in 24 h 7 min. Similar behavior appears for ViT-Base/16 on CIFAR-10 (97.5% vs. 94.8% Acc@1 at 70% compression) and for structured tasks such as Faster R-CNN and SegFormer on detection/segmentation benchmarks. This suggests that early removal of uninformative filters "cleans up" the internal representations, making the subsequent layer-level decisions more reliable and reducing the risk of removing structurally important blocks.

Second, the role of fine-tuning is clearly visible in the "No Fine-Tuning" rows. Without any adaptation after pruning, models suffer a sharp quality degradation even though parameters and FLOPs are identical to those of the fully fine-tuned IDAP++ variant. For instance, ResNet-50 / ImageNet at 90% compression falls from 69.3% Acc@1 with full IDAP++ to 60.1% without fine-tuning; BERT Base / MNLI-m at 70% compression drops from 82.1% to 76.4; Stable Diffusion v1.5 at 70% compression shows FID increasing from 13.5 to 18.4. Importantly, the wall-clock cost of fine-tuning dominates total compression time (hundreds to thousands of minutes depending on the model), but it is precisely this phase that recovers most of the performance lost during aggressive structural changes. The trade-off is therefore explicit: short, pruning-only schedules are cheap but produce clearly inferior models, while IDAP++ invests additional time to obtain compressed networks that remain competitive with their dense counterparts.

Third, comparing "Only Filter Pruning" and "Only Layer Truncation" demonstrates that the two stages are strongly complementary. Filter pruning alone typically preserves moderate quality but leaves a relatively heavy model; layer truncation alone yields more compact architectures but is significantly more destructive. For ResNet-50 / ImageNet at 70% compression, filter-only pruning achieves 69.4% Acc@1 with 8.9M parameters and 1.5 GFLOPs, whereas layer-only truncation achieves 66.2% Acc@1 with 5.4M parameters and 0.8 GFLOPs. The full IDAP++ pipeline, however, reaches 75.4% Acc@1 with 6.1M parameters and 1.0 GFLOPs — simultaneously surpassing both ablations in quality while maintaining a competitive resource profile. This pattern is repeated for EfficientNet-B4, ViT-Base/16, and all detection/segmentation models (Faster R-CNN, YOLOv4, FCN, U-Net, SegFormer), as well as for VQGAN and Stable Diffusion: the joint optimization in width and depth yields strictly better accuracy/FID–efficiency trade-offs than any single-stage strategy.

Finally, the NLP experiments confirm that these conclusions generalize beyond vision and generative models. On BERT Base, GPT-2 Base, and T5 Base, the full IDAP++ pipeline consistently outperforms all ablations for each compression level. For example, on GPT-2 Base / SQuAD 1.1 at 70% compression, IDAP++ attains 86.1 F1 with 55.0M parameters and 14.5 GFLOPs in 7 h 21 min, whereas the reversed order yields 82.7 F1; omitting fine-tuning reduces performance further to 80.3 F1; filter-only and layer-only variants drop to 81.4 and 78.9 F1, respectively, despite similar or smaller resource budgets. On T5 Base / MNLI-m at 70% compression, IDAP++ reaches 84.0% accuracy against 80.1–78.9% for the ablations, with lower latency and fewer parameters. Overall, Tables 30, 31, 32, 33, 34, 35, 36, 37, 38 show that (i) the ordering Filter Pruning → Layer Truncation is empirically optimal, (ii) fine-tuning is essential to unlock the benefits of aggressive structural pruning, and (iii) both stages of IDAP++ are necessary to achieve the best quality–efficiency–time trade-off across architectures and modalities.

