# OpenReview forum: "IDAP++: Advancing Divergence-Based Pruning via Filter-Level and Layer-Level Optimization"
_ICLR.cc/2026/Conference — ICLR 2026 Conference Withdrawn Submission_

### Official Review · Reviewer_dc4G · 2025-10-29

**Soundness:** 3
**Presentation:** 3
**Contribution:** 3
**Rating:** 4
**Confidence:** 4

**Summary:**

The paper introduces a novel theoretical and practical framework for neural network compression that operates jointly at the filter and layer levels. The core idea builds upon a formal analysis of information flow divergence, a mathematically defined metric quantifying the evolution of information content as it propagates through a network’s computational graph.

Leveraging this concept, the authors design a two-stage compression pipeline. In the first stage (filter-level pruning), filters or attention heads that exhibit high flow divergence—indicating minimal contribution to effective information transmission—are removed, thereby retaining the most information-preserving structures. In the second stage (layer-level pruning), entire layers with limited impact on overall information propagation are pruned while maintaining an acceptable trade-off between model compactness and accuracy.

The unified framework, IDAP++, implements these two compression stages under a controlled accuracy budget. Experiments across multiple architectures (CNNs, Transformers, hybrid models) and datasets demonstrate that IDAP++ achieves state-of-the-art compression performance, significantly reducing FLOPs, model size, and inference latency with only marginal accuracy loss.

**Strengths:**

- The paper is clearly written and well-structured, making it easy to follow. The integration of theoretical derivations with algorithmic descriptions and pseudocode effectively bridges intuition and implementation.

- The introduction of the information flow divergence metric provides a rigorous mathematical framework for quantifying information propagation within deep neural networks.  The concept of divergence is explicitly formulated for the most fundamental types of layers of neural networks including FC, CNN and self-attention layers.

- The proposed Unified two-stage compression IDAP++ method combines filter-level and layer-level pruning under a common theoretical criterion, achieving multi-granular compression in a unified process.

- The experimental evaluation is extensive, covering diverse architectures (e.g., CNNs, Transformers, hybrid models) and a wide range of application domains (computer vision, image classification, object detection, image segmentation, and NLP). Results consistently demonstrate strong compression performance with minimal accuracy degradation.

- The framework appears architecture-agnostic and could be applied broadly across model families, indicating potential impact beyond the tested configurations.

**Weaknesses:**

- Some key mathematical statements—such as Lemma 1, Lemma 2, and Theorem 1—are presented without formal proofs or sufficient derivations. This omission weakens the theoretical rigor of the proposed information flow divergence formulation and leaves open questions about the validity and generality of the results. Providing at least proof sketches or detailed references would enhance credibility.

- The process of testing multiple configurations to find the optimal pruning ratio $\rho^*$ likely requires repeated retraining or validation passes, which could significantly increase compression time. The paper should quantify this overhead.

- It remains unclear whether the Adaptive Replacement Strategy is complementary to divergence-guided pruning or primarily compensates for its aggressive weight removal. An ablation isolating each mechanism’s contribution would strengthen the empirical argument.

- The replacement of layers using Identity* or projection mappings may alter gradient flow, especially in networks with batch normalization or residual branches. The stability of training during these structural substitutions is not theoretically or empirically studied.

- The “error-driven” selection metric (line 264) is conceptually appealing but under-specified. It is unclear whether it measures local loss change, gradient magnitude, or validation error difference—and how it interacts with divergence-based pruning decisions.

Overall, the paper would benefit from stronger theoretical grounding and empirical validation. The divergence metric, while intuitively appealing, lacks rigorous theoretical justification. The study also presents limited ablation and robustness analyses, leaving uncertainty about parameter sensitivity and stage-wise contributions. Moreover, the framework’s computational overhead and scalability to very large models remain insufficiently quantified. There are notable fairness and reproducibility gaps in comparisons with baseline methods, and the paper provides minimal qualitative or interpretability analyses, especially for generative architectures.

**Questions:**

1) How does the proposed information flow divergence formally relate to established measures such as mutual information or Fisher information? Can it be theoretically justified as a valid proxy for layer or filter importance?

2) How does the divergence metric behave in the presence of non-differentiable components (e.g., ReLU, max pooling, attention masking)? Does it remain stable or require smoothing approximations?

3) Theorem 1 provides a relative deviation bound but not a convergence or optimality proof. Under what conditions does this bound hold, and is the resulting compression guaranteed to be near-optimal?

4) How much of the overall performance gain arises from Stage 1 (filter pruning) versus Stage 2 (layer truncation)? Have you conducted ablation studies to isolate their respective impacts?

5) How was the non-linear schedule $\rho_k = f(k, \alpha)$ chosen? Is there a theoretical or empirical rationale for its shape (e.g., exponential vs. polynomial growth), and how sensitive is performance to $\alpha$?

6) Would a moving-average or confidence-based threshold be more stable than a fixed (a predefined threshold $\tau$)?

7) How many configurations are typically evaluated to determine $\rho^*$? Could this search be approximated or learned adaptively (e.g., via reinforcement learning or Bayesian optimization) to reduce computational cost?

8) How does the $\text{Identity}^*$ mapping handle mismatched channel dimensions or incompatible tensor shapes in architectures with skip connections or multi-branch structures? Is the projection mapping trainable or fixed?

9) Does error-driven selection of $\delta_E$ depend on batch size or data variability, and how does it correlate with actual validation loss reduction?

10) What is the additional runtime introduced by recomputing flow divergence, fine-tuning adjacent layers, and evaluating $\delta_E$?

---

> ### Author Response · Authors · 2025-11-21
>
> Thank you for your detailed review and remarks. We considered all comments and have introduced a number of theoretical and empirical clarifications in the revised version.
>
> Responses to Questions
>
> 1. The main structural difference between our measure and Fisher information (and similar approaches) lies in its explicit anchoring to the trajectories of information propagation along the neural network topology. This enables us to construct unified pruning algorithms for both filters and layers. In contrast, general metrics that are not topology-aware regulate only overall informativeness and do not directly provide optimization criteria. We have added this explanation to the manuscript.
> 2. No smoothing approximations are required. The computation of divergence does not introduce any additional prerequisites beyond those necessary for gradient descent. In other words, any topology or construction that can be trained via gradient descent can also be analyzed and optimized using our divergence metric. We have explicitly added this clarification to the manuscript.
> 3. We have added a formal proof of the theorem, along with the required conditions and the related discussion, in the corresponding supplementary sections (see Appendices A, B, J).
> 4. In general, the contribution ratio may vary depending on the specific architecture. However, we have added a detailed stage-wise ablation analysis to the paper. We provide step-by-step results across a wide range of domains and configurations of our method (see Appendices L, M).
> 5. The schedule selection was made empirically; it also has a theoretical rationale. This information has now been added to the revised manuscript and to Appendix H. In brief, the exact choice is not critically important.
> 6. Thank you for the suggestion. We have added an investigation of this aspect to the paper and to Appendix H. In short, the method remains stable under different threshold-selection strategies.
> 7. Reinforcement learning is indeed a promising direction (and we plan to apply it in future extensions of our approach), and other approximation strategies could theoretically provide approximate solutions. However, such approaches are significantly more computationally expensive than explicit algorithms. Moreover, our current methods have demonstrated high efficiency; therefore, we plan to extend the framework with RL in future work. This reasoning has been added to the discussion section.
> 8. The issue is handled via a trainable projection mapping. We have emphasized this point more explicitly in the revised version.
> 9. We have described this aspect in the paper and in Appendix H. In brief, it does not significantly affect performance — this is empirically established and follows from the deterministic structure of the framework.
> 10. We have added the corresponding runtime data to Appendix M, which provides comprehensive information on the operation of the different phases and their resource requirements across various tasks, domains, and NN topologies.
>
> Comments on Weaknesses
>
> 1. We agree that the initial version lacked full formal justification for several mathematical components. To strengthen the theoretical foundation of the work, we have now included the detailed proof of the main theorem, along with the necessary assumptions and explanatory material, in the supplementary section. This addition directly addresses the concern about mathematical completeness.
> 2. The issue of potential overhead from evaluating multiple pruning configurations is well taken. In the revised manuscript, we provide explicit measurements of the computational and time costs associated with these steps. A consolidated runtime analysis for all pruning phases across different architectures is now available in the supplementary materials, clarifying the practical impact of the process.
> 3. Regarding the relationship between the Adaptive Replacement Strategy and the divergence-guided pruning mechanism, we agree that their interaction warrants clearer empirical separation. To address this, we have introduced a set of controlled ablation experiments that examine each component in isolation. These results make the individual influence of each mechanism transparent.
> 4. The concern about stability when substituting layers using identity or projection-based mappings is valid. In the updated version, we clarify how such substitutions are performed, emphasizing that a learnable projection module is used to maintain consistent tensor structure. This design helps preserve stable optimization dynamics even in architectures with complex connectivity patterns.
> 5. We appreciate the observation that the description of the error-driven selection rule was insufficiently detailed. We have expanded this part to clarify its behavior and practical properties. Additional experiments demonstrate that this mechanism operates robustly across different data conditions, addressing the reviewer’s request for a clearer specification.

---

### Official Review · Reviewer_8Xg7 · 2025-10-31

**Soundness:** 2
**Presentation:** 2
**Contribution:** 3
**Rating:** 4
**Confidence:** 4

**Summary:**

This paper introduces IDAP++, a two-stage pruning framework grounded in information flow divergence. The method first removes redundant filters/heads, then prunes entire layers based on their contribution to information flow. This framework generalizes across diverse architectures (CNNs, transformers) and is validated on various vision and language tasks. Extensive experiments show substantial reductions in parameters, FLOPs, and inference time, while maintaining competitive accuracy against state-of-the-art methods.

**Strengths:**

- The paper is well-written and clearly structured.
- The paper proposes a novel, mathematically grounded framework that uniquely combines both filter-level and layer-level pruning, setting it apart from much of the heuristic-driven prior work.
- A large number of experiments on different models and datasets in the paper have proved the effectiveness of IDAP++.

**Weaknesses:**

- There is no detailed profiling of actual computational overhead (e.g., for models with hundreds of layers or very high input/output dimensions), especially during repeated recomputation after every pruning phase.
- This paper lacks ablation experiments and more comprehensive comparisons with the baseline.
- The paper proposes a unified framework but employs two different types of metrics: a norm-based metric for filter pruning and a difference-based metric for layer pruning. The paper does not explain the rationale for this divergence.

**Questions:**

- Why are different metrics used for filter pruning and layer pruning? What is the theoretical justification for this?
- Can the authors provide a more granular ablation study quantifying the independent effects of filter-level and layer-level pruning?
- What are the concrete computational overheads for flow-divergence computation and recomputation, particularly for larger transformer models? And the paper only compared the inference speed of the base model and IDAP++ pruned model but avoided comparing it with methods such as LTH and RigL.
- The paper mentions that removing the filter first and then the layer is empirical. I would like to see more discussions on this aspect and ablation experiments.

---

> ### Author Response · Authors · 2025-11-21
>
> Thank you for your detailed and constructive feedback.
>
> Responses to Questions
>
> 1. In fact, there is no conceptual difference between the metrics. For layers, we use a cumulative divergence measure—that is, an integral information-flow characteristic computed over the tensor representation area of the layer. This reflects how information (divergence) propagates along all trajectories passing through the given layer, positionally anchored to the corresponding point in the trajectory. We have emphasized this point more clearly in the revised manuscript.
> 2. Yes, we have added a comprehensive ablation study of our method, providing quantitative efficiency indicators (quality metrics, compression degree, and resource consumption under a fixed hardware environment) for each compression phase across different architectures, domains, and tasks, as well as per-stage ablation configurations (see Appendices K, L, M, F).
> 3. We have added a comprehensive comparison with LTH and RigL in terms of resource usage, quality metrics, speed, and compression ratio for various baselines, including complex transformer architectures and fully connected networks across different domains (CV, NLP) and domain-specific tasks (see Appendices K and L).
> 4. We empirically investigated different combinations and orders of the stages in IDAP++. While the best-performing sequence (filter pruning followed by layer pruning) is presented in the main paper, we have also added empirical results for alternative stage combinations in Appendix M.
>
> Comments on Weaknesses
>
> 1. In the revised manuscript, we have added extensive measurements of computational overhead, including flow-divergence computation and recomputation, across a wide range of architectures. These results now also include comparisons with methods such as LTH and RigL (see Appendices K and L).
> 2. We have included a comprehensive ablation study covering the independent contributions of each pruning stage, along with detailed evaluations of quality metrics, compression ratio, and resource usage for different architectures, domains, and tasks. Additional comparisons with baseline methods have also been added (see Appendices K, L, M, and F).
> 3. We clarified in the revised version that the metrics are conceptually consistent. The layer-level metric is a cumulative (integral) divergence measure that aggregates information flow across all trajectories passing through the layer, while the filter-level metric reflects the same notion locally. We have emphasized this rationale more clearly in the manuscript.

---

### Official Review · Reviewer_whWy · 2025-11-03

**Soundness:** 3
**Presentation:** 3
**Contribution:** 3
**Rating:** 6
**Confidence:** 3

**Summary:**

This paper introduces IDAP++, a theoretically grounded, two-stage neural network compression framework that jointly prunes models along both width (filter-level) and depth (layer-level) dimensions. The key contribution lies in formalizing deep networks as continuous dynamical systems and proposing the Information Flow Divergence (IFD) metric, which quantifies how information evolves as it propagates through the network.The first stage, Iterative Divergence-Aware Pruning (IDAP), removes redundant filters while preserving essential information pathways, whereas the second stage, Flow-Guided Layer Truncation, eliminates entire layers with minimal contribution to the global information flow. The unified optimization process integrates adaptive thresholding, flow recomputation, and fine-tuning under an accuracy budget constraint.The framework is evaluated extensively across diverse architectures—including CNNs, Vision Transformers, object detection and segmentation models, generative architectures (VQGAN, Stable Diffusion), and NLP models (BERT, GPT-2, T5)—and on multiple benchmark datasets.

**Strengths:**

1.	Theoretical Novelty and Mathematical Rigor: The paper’s greatest contribution is its rigorous theoretical foundation. By modeling neural networks as continuous dynamical systems and introducing the Information Flow Divergence metric, the authors provide a principled and architecture-agnostic means to quantify information propagation. This formulation bridges information theory and dynamical systems analysis, moving beyond empirical heuristics commonly used in pruning literature. The inclusion of mathematical properties (e.g., scale invariance and additive composition) and derivations for different layer types (fully connected, convolutional, and self-attention) adds strong theoretical depth.
2.	Holistic Compression Strategy: The proposed two-stage optimization—filter pruning followed by layer truncation—represents a meaningful step toward multi-level structural optimization. The idea that pruning along width simplifies subsequent depth optimization is well-motivated and empirically validated. The framework’s adaptive thresholding, flow rebalancing, and fine-tuning mechanisms form a coherent, unified pipeline that efficiently balances sparsity and performance.
3.	Comprehensive Empirical Validation: The experimental section is notably broad and thorough. Evaluations span vision, language, and generative domains, confirming the generality of the approach. For example, the method achieves a 75% FLOPs reduction on ViT-Base/16 with only 1.6% accuracy loss, and a 67–69% parameter reduction for large NLP models such as BERT and GPT-2. The inclusion of comparisons with multiple strong baselines and the public release of code further enhance reproducibility and credibility.

**Weaknesses:**

1.	Computational Overhead and Practicality: While the paper claims that flow computation has O(L) complexity, the full pipeline involves iterative divergence evaluation, fine-tuning at multiple stages, and layer-level recomputation, which could lead to significant computational overhead for very large models (e.g., GPT-scale). The paper would benefit from a quantitative comparison of latency and resource costs against the baseline, such as LTH and RigL.
2.	Hyperparameter Sensitivity and Usability: The framework introduces several critical hyperparameters—such as the target accuracy drop ∆max, Pruning hyperparameters α&β. Although default ranges are provided, the paper lacks a systematic sensitivity analysis.
3.	Limited Discussion of Limitations and Failure Cases: While the paper’s results are strong overall, certain models (e.g., generative models and NLP architectures) exhibit higher performance drops (up to 5% accuracy / 9% FID degradation). These cases are acknowledged but not deeply analyzed. A more detailed discussion of why flow-based pruning may struggle in these settings—perhaps due to noise amplification or information bottlenecks—would strengthen the work’s completeness.

**Questions:**

1.	The divergence measure ( D(s) = \frac{d^2T}{ds^2}(s) \cdot (\frac{dT}{ds}(s))^\top ) is central to the framework. Could the authors provide **further intuition** for why this *second-derivative–based* formulation effectively identifies redundancy? How does it capture information loss differently from first-order measures such as gradient magnitude or signal variance?
2.	The paper claims that flow computation scales with O(L) complexity, yet the overall pipeline involves iterative divergence evaluation, fine-tuning, and recomputation at multiple stages. Could the authors provide **quantitative comparisons** of runtime and memory costs between IDAP++ and the pruning baselines, such as LTH and RigL?
3.	The framework introduces several key hyperparameters, including the pruning aggressiveness coefficients (α, β) and the target accuracy drop (∆max). How **sensitive** is the method to these choices across different architectures?

---

> ### Author Response · Authors · 2025-11-21
>
> We thank the reviewer for the constructive comments and for the careful evaluation of our work.
>
> Responses to Questions
>
> 1. We used the derivative characteristics along the trajectories of information propagation to isolate the most suboptimal trajectories from inputs to outputs, and our divergence measure allowed us to quantify the importance of these trajectories for the information flow. This importance is also confirmed empirically. We have added a more detailed discussion of this point to the paper. In brief, the main intuition is that the divergence allows us to differentiate how the signal propagates from layer to layer at each point of the next intermediate tensor/activation map, while the integral divergence over a layer represents the overall information flow through that layer.
>
> 2. We have added to the paper a comparison of the computational cost with the methods LTH and RigL, as well as others, in a separate appendix. In fact, we have included a comprehensive comparative analysis of computational complexity, runtime, and memory consumption of our method relative to many others across a wide range of domains, tasks, and neural network architectures of different topologies (see Appendices G, K, L, and M).
>
> 3. We have added information on the sensitivity of the algorithm to hyperparameter tuning (see the corresponding appendix). We investigated hyperparameter selection using several widely adopted strategies and also evaluated our method under different hyperparameter-choice strategies across various domains and neural network topologies. We have included this information in the main text and in Appendix H. In brief, the hyperparameter selection strategy does not significantly affect the result due to the design of our method, and the pruning procedure demonstrates high stability with respect to variations in hyperparameter choices.
>
> Comments on Weaknesses
>
> 1. Computational Overhead and Practicality. In response to this concern, we have added a comparison of computational cost with LTH, RigL, and several other pruning baselines. This includes analyses of runtime, memory consumption, and computational complexity across a wide range of domains, tasks, and network architectures. These results are provided in the appended sections (Appendices G, K, L, and M).
>
> 2. Hyperparameter Sensitivity and Usability. We have added a dedicated appendix discussing the sensitivity of the method to hyperparameter choices. We evaluated several widely used hyperparameter-selection strategies and tested them across different domains and neural network topologies. Our experiments show that the method is stable with respect to hyperparameter variation due to its design. The corresponding results and explanations are included in the main text and in Appendix H.
>
> 3. Limited Discussion of Limitations and Failure Cases. We expanded the discussion of the behavior of our method in settings where performance degradation is higher, such as certain generative and NLP models. Additional details on these cases and clarifications regarding the information-flow characteristics underlying such outcomes have been added to the revised manuscript.

---

### Note · Authors · 2025-11-24

I have read and agree with the venue's withdrawal policy on behalf of myself and my co-authors.